# Glacial lake outburst flood hazard under current and future conditions: worst-case scenarios in a transboundary Himalayan basin

Simon K. Allen[1,2], Ashim Sattar[1], Owen King[3], Guoqing Zhang[4], Atanu Bhattacharya[3,5], Tandong Yao[4], Tobias Bolch[3]

[1]Department of Geography, University of Zurich, CH-8057, Zurich, Switzerland
[2]Institute for Environmental Science, University of Geneva, CH-1205, Geneva, Switzerland
[3]School of Geography and Sustainable Development, University of St Andrews, KY16 9AL, St Andrews, UK
[4] State Key Laboratory of Tibetan Plateau Earth System, Environment and Resources (TPESER), Institute of Tibetan Plateau Research, Chinese Academy of Sciences, Beijing 100101, China
[5] Department of Earth Sciences and Remote Sensing, JIS University, Kolkata, India

*Correspondence to*: Simon Allen (skallenz@gmail.com)

**Abstract**

Glacial lake outburst floods (GLOFs) are a major concern throughout High Mountain Asia, where societal impacts can extend far downstream. This is particularly true for transboundary Himalayan basins, where risks are expected to further increase as new lakes develop. Given the need for anticipatory approaches to disaster risk reduction, this study aims to demonstrate how the threat from a future lake can be feasibly assessed along-side that of worst-case scenarios from current lakes, and how this information is relevant for disaster risk management. We have focused on two previously identified dangerous lakes (Galongco

and Jialongco), comparing the timing and magnitude of simulated worst-case outburst events from these lakes both in the Tibetan town of Nyalam and downstream at the border with Nepal. In addition, a future scenario has been assessed, whereby an avalanche-triggered GLOF was simulated for a potential large new lake forming upstream of Nyalam. Results show that large (> 20 mil m$^3$) rock and/or ice avalanches could generate GLOF discharges at the border with Nepal that are more than 15 times larger than what have been observed previously, or anticipated based on more gradual breach simulations. For all

assessed lakes, warning times in Nyalam would be only 5 – 11 minutes, and 30 minutes at the border. Recent remedial measures undertaken to lower the water level at Jialongco would have little influence on downstream impacts resulting from a very large magnitude GLOF, particularly in Nyalam where there has been significant development of infrastructure directly within the high-intensity flood zone. Based on these findings, a comprehensive approach to disaster risk management is called for, combining early warning systems with effective land use zoning and programs to build local response capacities. Such

approaches would address the current drivers of GLOF risk in the basin, while remaining robust in the face of worst-case, catastrophic outburst events that become more likely under a warming climate.

**Keywords**

Glacial lake outburst flood, process chain, hazard, risk, future, Himalaya

**1 Introduction**

Widespread retreat of glaciers has accelerated over recent decades in the Himalaya as in most other mountain regions worldwide as a consequence of global warming (Bolch et al., 2019; King et al., 2019; Maurer et al., 2019; Zemp et al., 2019). A main consequence has been the rapid expansion and new formation of glacial lakes (Gardelle et al., 2011; Nie et al., 2017; Shugar et al., 2020), which has large implications for both water resources and hazards (Haeberli et al., 2016). When water is

suddenly and catastrophically released, Glacial Lake Outburst Floods (GLOFs) can devastate lives and livelihoods up to hundreds of kilometres downstream (Carrivick and Tweed, 2016; Lliboutry et al., 1977). This threat is most apparent in the Himalaya, where glacial lakes have been increasing rapidly in both size and number (Chen et al. 2021; Wang et al. 2020; Zhang et al., 2015), and where a frequency of 1.3 GLOFs per year has been recorded since the 1980s (Veh et al., 2019). The fact that GLOFs can extend across national boundaries exacerbates the challenges for early warning or other risk reduction

strategies, particularly in politically sensitive regions (Allen et al., 2019; Khanal et al., 2015a).

Lakes can develop either underneath (subglacial), at the side, in front (proglacial), within (englacial), or on the surface of a glacier (supraglacial), with the dam being composed of ice, moraine, or bedrock. In Asia, most scientific attention has focussed upon the hazard associated with the catastrophic failure of moraine-dammed lakes, and particularly those trapped behind

proglacial moraines (e.g., Fujita et al., 2013; Westoby et al., 2014; Worni et al., 2012). Such lakes can be very large, with volumes larger than 100 million m$^3$, and depths exceeding 200 m (Cook and Quincey, 2015), and are susceptible to a range of failure mechanisms owing to the low material strength of the dam structure (Clague and Evans, 2000; Korup and Tweed, 2007). In Asia, as elsewhere in the world, displacement waves generated from large impacts of ice or rock have contributed to

the majority of moraine dam failures, occurring predominantly over the warm summer months (Emmer and Cochachin, 2013; Liu et al., 2013; Richardson and Reynolds, 2000). At least 17 GLOF disasters (causing loss of life or infrastructure) have been documented in Tibet since 1935, mostly originating in the central-eastern section of the Himalaya (Nie et al., 2018). Coupled with rapidly increasing population and infrastructural development in the region, an urgent need for authorities to take action and implement timely risk reduction measures has been acknowledged (Wang and Zhou, 2017), considering the best available knowledge on existing threats (e.g., Allen et al., 2019; Wang et al., 2015a, 2018), but also with a view to the future (Furian et al., 2021; Zheng et al., 2021a).

Despite no clear trend observed in GLOF activity over recent decades in the Himalaya (Veh et al., 2019), the ongoing expansion of lakes towards steep and potentially destabilised mountain flanks is expected to lead to new challenges in the future with implications for hazards and risk (Haeberli et al., 2017). Based on approaches to model the possible future expansion and development of new lakes (Linsbauer et al., 2016) several studies have aimed to quantify the possible implications for GLOF frequency and/or magnitude for different regions (Allen et al., 2016; Emmer et al., 2020; Magnin et al., 2020). For example, in the Indian Himalayan state of Himachal Pradesh, Allen et al. (2016) demonstrated a 7-fold increase in the probability of GLOF triggering and a 3-fold increase in the downstream area affected by potential GLOF paths under future deglaciated conditions. Meanwhile, Zheng et al. (2021a) have elaborated such analyses for the entire High Mountain Asia, revealing that the number of lakes posing a transboundary threat within border areas of China and Nepal could double in the future. While such large-scale, first-order studies are important for raising general awareness of the future challenges that mountain regions will face (Hock et al., 2019), there are limitations in the extent to which these studies can directly inform planning and response actions at the ground level.

The need for forward-looking, anticipatory approaches to hazard and risk modelling, including attention to possible worst-case scenarios is clearly recognised within international guidelines on glacier and permafrost hazard assessment (GAPHAZ, 2017). However, practical examples on how to account for worst-case scenarios and future lake development in local GLOF hazard assessment and risk management have been rarely demonstrated. International best practice is framed by both a first-order assessment undertaken at large scales (to identify potentially critical lakes), followed by a detailed assessment for these lakes using numerical models to simulate downstream flood intensities as a basis for hazard mapping (GAPHAZ, 2017). This is a common approach for existing threats, where the time, data, and expertise needed to invest in comprehensive hazard modelling and mapping can be well justified for a lake that is known to be critical, yet, worst-case scenarios are often neglected and may far exceed historical precedence. For future lakes, where the timing of lake formation is typically highly uncertain, there remains a methodological gap in the hazard assessment process, as authorities are unlikely to undertake sophisticated hazard mapping for a threat that may not even eventuate. In this study we aim to address these gaps, by providing an illustrative example of how a worst-case outburst scenarios from a potential future lake can be systematically assessed along-side the threat posed by current lakes, before discussing the relevance of such an assessment for disaster risk management in a transboundary context.

Focusing on the transboundary Poiqu river basin in the central Himalaya, the specific objectives of the study are to 1) apply systematic criteria to establish worst-case outburst scenarios and assess the magnitude of downstream impacts from two potentially critical lakes, considering also the effect of recent remedial measures at one of the lakes, 2) compare the results with a potential outburst from a large lake that is anticipated to develop in the future, and 3) discuss the implications for early warning or other risk reduction strategies.

## 2 Study area

This analysis focuses on a ca. 40 km stretch of the lower Poiqu river basin originating from Galongco glacial lake, considering potential GLOF impacts in Nyalam town (capital of Nyalam county, Tibetan Autonomous Region), and downstream to the border with Nepal at Zhangmu (Fig. 1). The elevation range of the study area extends over 6000 metres, from the summit of Shishapangma at 8,027 m a.s.l, whose glacierised slopes feed Galongco, to 2000 m a.s.l in the river valley at Zhangmu. According to Wang and Jiao (2015), mean annual air temperature and mean annual precipitation in Nyalam (3810 m asl) are 3.8°C and 650.3 mm respectively, with sub-zero temperatures lasting from November – March each year. Temperatures peak in July (10.8°C), while highest average precipitation totals are recorded in September (87.9 mm). In total, 60% of the annual rainfall falls during the monsoon months of July – September (Wang et al., 2015b)

The Poiqu basin is the Tibetan portion of the large transboundary Poiqu/Bhote Koshi/Sun Koshi River Basin, along which the economically important Friendship Highway links China to Nepal, and where significant hydropower resources are located (Khanal et al., 2015b). Based on a larger study across Tibet, the Poiqu basin has been identified as a clear hotpot of transboundary GLOF danger (Allen et al. 2019 – Fig. 1), where at least 6 major GLOF events were reported over the past century, including repetitive events from Jialongco in 2002 (Chen et al., 2013), and Cirenmaco in 1964, 1981 and 1983 (Wang et al., 2018). The 1981 event resulted in numerous fatalities, and estimated losses of up to US\$4 million (currency value as of 2015) as a result of damage to houses, roads, hydropower, and disruption to trade and transportation services (Khanal et al., 2015a). Meanwhile, an outburst of $1.1 \times 10^5 \, m^3$ from Gongbatongshacuo (adjacent to Cirenmaco) in July 2016, resulted in significant damage to hydropower and roads, exacerbating losses inflicted one year earlier by the Gorkha earthquake (Cook et al., 2018). Whereas Gongbatongshacuo has completely drained, Cirenmaco remains a persistent threat, identified by multiple studies as being one of the most dangerous lakes in Tibet (Allen et al., 2019; Wang et al. 2015a; Wang et al., 2018).

In the current study, we focus not on Cirenmaco, which has already been the subject of comprehensive investigations (Wang et al., 2018), but rather on two other well-documented threats of Jialongco and Galongco, owing to their potential to cause damage to the Tibetan county capital of Nyalam, and downstream in Nepal (Allen et al. 2019; Shresta et al. 2010). In fact, after Cirenmaco, Galongco and Jialongco were ranked 2[nd] and 3[rd] respectively in a recent assessment of most dangerous glacial lakes across Tibet, owing to both the physical characteristics of the lakes and their surroundings (see section 4.1), and high levels of exposure in downstream areas (Allen et al. 2019). Both moraine-dammed proglacial lakes have expanded rapidly over the past decades, with Galongco, the largest lake in the basin, increasing its area by 450% from 1.00 to 5.46 km$^2$ in the period 1964-2017 (Wang et al., 2015b; Zhang et al., 2019). The potential future lake is located around 6 km further upstream from Jialongco (Fig. 1 – see 3.1 for further description).

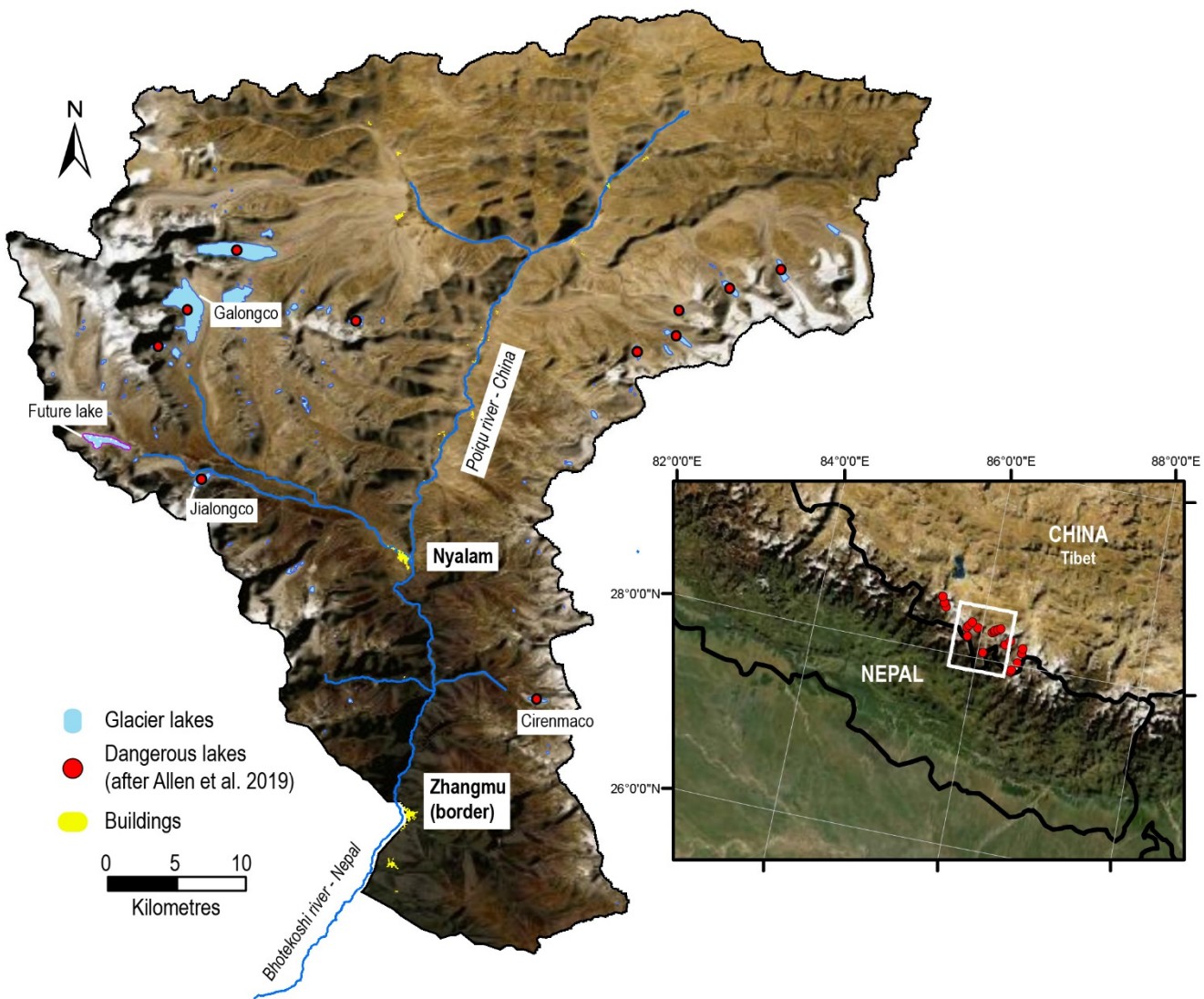

**Figure 1: Location of the Poiqu River basin within a hotspot of GLOF risk, as determined on the basis of 30 potentially most dangerous lakes identified across Tibet (after Allen et al., 2019). The current lakes focussed on in this study of Galongco and**
**Jialongco are indicated, as is the modelled future lake, the county capital town of Nyalam, and the town of Zhangmu, through which the border between China and Nepal passes. Cirenmaco, from which several outburst floods have been reported, is also indicated; Background image: ESRI Basemap Imagery.**

## 3 Methodological approach

In line with recent international guidance in GLOF hazard assessment (GAPHAZ 2017), in this study we consider lake
susceptibility, which determines the likelihood of a given outburst scenario to occur, and use the GIS-based open-source numerical simulation tool r.avaflow to model the GLOF process chain and determine downstream impacts. In order to compare the threat posed by the two current lakes with an anticipated future lake, we focus on worst-case scenario modelling – that is to say, very large avalanche-triggered outburst events from Jialongco, Galongco, and the anticipated future lake.

3.1 Lake susceptibility and scenario development
The assessment follows a systematic approach that considers wide-ranging atmospheric, cryospheric and geotechnical factors that can influence lake susceptibility, and thereby the likelihood of a GLOF event occurring (after GAPHAZ 2017). We draw

on remotely sensed data to the extent possible, complimented with field observations to enable a semi-quantitative assessment and comparison of susceptibility factors across the three lakes. Topographic characteristics (dam geometry, slope angles etc)

and geological structures of the surrounding slopes were precisely measured using high resolution 1m Pleiades orthoimagery and Digital Elevation Model (DEM), generated from 0.5 m resolution tri-stereo Pleiades imagery acquired in October 2018, covering the whole Poiqu basin. Potentially unstable zones of glacial ice were identified in the imagery and Google Earth, based on orientation and density of crevassing, with a subsequent estimate of the ice thickness and volume provided from GlabTop model output (Table 1). Furthermore, the time series of Google Earth imagery was examined to identify any evidence

of historical mass movements, that could indicate an enhanced threat to the lakes below. Factors assessed, their primary attributes, and sources used are further described in Section 4.1. Based on this assessment, and the recognition of a large ice and/or rock avalanche triggered GLOF process-chain being the most significant threat to all 3 lakes, avalanche source areas were identified as input to the process chain modelling (Table 1).

**Table 1: Input scenarios for rock/ice avalanche starting zones (see Fig. 2) threatening Jialongco (JC), Galongco (GC), and the future lake (FL). Source areas are defined based on high-resolution satellite imagery. Mean ice thickness and resulting ice volume is based on GlabTop. Note that the JC-L scenario is defined for a lowered Jialongco lake, as the lake level was lowered since 2018; See Section 4.1 for further details.**

| | Mean slope (°) | Area (m²) | Type | Mean ice thickness (m) | Ice volume (10⁶ m³) | Mean rock thickness (m) | Rock volume (10⁶ m³) | Total volume (10⁶ m³) |
|---|---|---|---|---|---|---|---|---|
| JC | 35 | 600,000 | Ice avalanche | 30 | 18 (100%) | – | – | 18 |
| JC-L | 35 | 600,000 | Ice avalanche | 30 | 18 (100%) | – | – | 18 |
| GC | 50 | 460,000 | Rock-ice avalanche | 10 | 4.6 (20%) | 40 | 18.4 (80%) | 23 |
| FL | 55 | 516,000 | Rock avalanche | – | – | 40 | 20.6 (100%) | 20.6 |


3.2 Avalanche and GLOF Modelling

The GLOF process chain was simulated with r.avaflow (Mergili et al., 2017; Pudasaini and Mergili, 2019), a GIS-based open-source simulation framework for multi-phase mass flows, which has the capacity to dynamically compute the interaction

between triggering landslides (in this case rock/ice avalanches) and lakes. The model is also capable of computing debris flow hydraulics. Major model inputs include the initial avalanche source characteristics (Table 1), terrain data, friction parameters and erosion parameters (see below), lake bathymetry and volume.

Bathymetry surveys of Jialongco and Galongco were undertaken in 2019, using an unmanned vessel. The onboard GPS system

achieves ~2.5 m horizontal positioning accuracy, while the single-beam sonar sounder has a vertical accuracy of 1 cm ± 0.1% of depth measured. Contour maps of lake depths were interpolated by using Kriging geo-statistics. Maximum depths of 134 and 200 metres were recorded for Jialongco and Galongco respectively, while volumes based on the interpolated bathymetry were 40 and 590 x 10⁶ m³. Following the construction of an artificial channel and associated lowering of the water level in

Jialongco, bathymetry was remeasured in 2021, giving a post-lowering maximum depth of 113 m, and volume of 23.5 x $10^6$ m³.

For GLOF modelling from the future lake, the location, bathymetry, and volume of the potential lake upstream from Jialongco is based on a modelled overdeepening in the glacier bed topography using GlabTop (Linsbauer et al., 2012). The model is now well established for providing a first-order indication of where lakes may develop in the future (e.g., Allen et al., 2016; Haeberli et al., 2016; Linsbauer et al., 2016; Magnin et al., 2020). The ice thickness distribution from GlabTop is subtracted from a surface DEM to obtain the bed topography, i.e. a DEM without glaciers, from which overdeepenings in the glacier bed can be detected and volumes estimated. Inputs to the model include manually edited glacier branch lines, and a DEM – in this case the NASA Shuttle Radar Topography Mission (SRTM) Version 3.0 (void filled) was used, at 30 m resolution. While the model predicts several possible locations in the Poiqu basin where large future lakes can develop, we focussed on the largest of these lakes that threaten the town of Nyalam. Based on the modelled geometry of the overdeepening, a maximum future lake depth of 168 m, and volume of 70 x $10^6$ m³ is estimated. The modelled bedrock topography forms the lake dam, i.e., the possible deposition of moraine on top of the bedrock, creating a higher dam structure, is not considered. Likewise, in keeping with a worst-case approach, we do not consider sediment deposition into the lake, that will potentially reduce the volume and longevity of the lake (Steffen et al., 2022). Beyond its potential size, this overdeepening was selected owing to its position in an area of low surface gradient behind a pronounced terminal moraine, beneath a tongue where supraglacial ponds are already developing, and at an elevation that is lower than other overdeepenings in the area. All factors provide favourable preconditioning for the formation of a large proglacial lake (Frey et al., 2010; Linsbauer et al., 2016).

Depending on the defined GLOF process-chain scenarios (Table 1), we assume the mixture of one or two solid phases in the initial avalanche (rock component; ρ=2700 kg/m³ and ice component; ρ=900 kg/m³) and one fluid phase; ρ=1000 kg/m³ (lake water), where the ice-rock volume ratios are calculated based on assessment in Section 4.1. We define the damming moraine of the lakes as entrainment zones composed of the rock phase (representing glacier deposits) with a grain density of 2700 kg/m³. A simplified entrainment model is applied, which is a product of the flow momentum and the empirical entrainment coefficient (Mergili et al., 2017). However, the final erosion depths are dependent on the momentum of the particular process and are controlled by the entrainment coefficient. Other input parameters include basal friction angle ($\varphi$) and internal friction angle ($\delta$) that govern the rheology of the flow. Here we set $\varphi = 25°$, $\delta = 10°$ for the initial stage of the process chain dominated mostly by the solid phase, i.e., avalanche, lake impact, and moraine erosion. For the downstream process from the moraine, we set $\varphi = 25°$, $\delta = 1°$ to model the flow as a water-saturated debris flow. The domain of the model is constructed such that it completely encompasses the avalanche source areas down to the China-Nepal border. All the simulations are executed for a total duration set to 1 hour 15 minutes (4500 s) providing enough time to evaluate the GLOF propagation downstream to the border. Finally, to evaluate the flow hydraulics obtained in terms of flow depth and discharge; we define three cross-sections along the flow channel located (i) immediately downstream of the damming moraine (ii) at Nyalam (nearest settlement), and (iii) at Zhangmu (China - Nepal border).

It is to be noted that we assumed no entrainment of the frontal moraine in the Jailongco Lowered Scenario (JC-L), as the damming moraine was lowered by up to 15 – 20 m, and armoured with concrete as a part of the engineering works performed for GLOF mitigation since 2018. For GLOFs originating from the future lake we evaluate the cascading impact of the flow impacting into Jialongco, located ~6 km downstream (see Fig. 1). While several freely available DEMs were tested (e.g., ALOS PALSAR at 12. 5 m or HMA at 8 m), topographic artefacts led to modelling errors. As such, the 1-m Pleiades DEM

was finally used for all simulations (based on imagery from 2018, with exception of the JC-L simulation which used an updated DEM from 2021 for the dam area).

3.3 Future lake development

Previous studies (e.g., King et al., 2018; Quincey et al., 2007) have identified glacier surface attributes which may precondition
the surface of debris-covered glaciers for supraglacial lake development. Glaciers bounded by large lateral and terminal moraines which have a flat or gently sloping (<~2°), slowly flowing (<~10 m a$^{-1}$) main tongue are hotspots of supraglacial pond development as surface meltwater cannot drain from the glacier surface (e.g., King et al., 2018; Quincey et al., 2007). Such pond networks expand when the mass balance of the glacier is negative and coalesce to eventually form a supraglacial lake at the hydrological base level of the glacier- the lowest point where the glacier surface intersects the terminal moraine
(Figures 3 & 19 in Benn et al., 2012). Large supraglacial lakes located close to the termini of debris-covered glaciers can persist for decades, over which period they expand, deepen and eventually transition to become proglacial lakes, such as Galongco and Jialongco. By examining contemporary and historical glacier surface velocity and elevation changes it is therefore, possible to identify glacier surfaces suited for surface meltwater ponding, which represent current and future sites of supraglacial lake development. To establish the possibility of lake development and the likely future trajectory of lake area
growth on the parent glacier up-valley from Jialongco (RGI60-15.09475), we examined the surface velocity, rate of thinning, and the evolution of the geometry (surface slope) of the glacier in recent decades.

We used the Pleiades DEM and glacier surface elevation change data generated by King et al., (2019) to examine the evolution of the geometry of glacier RGI60-15.09475 since the 1970s. Glacier surface slope estimates were derived by the fitting of
linear regression models through 'average' (mean of 5 evenly spaced) elevation profiles of the glacier surface split into 750 m long segments (King et al., 2018). We also assessed the current flow regime of the glacier using surface velocity data, which was generated through the tracking of glacier surface features visible in Sentinel 2 imagery over the period 2017-2019 (Pronk et al., 2021). Examination of these parameters established that the conditions at the surface of the glacier (Fig. 7) are well suited to imminent glacial lake development considering the factors outlined by Quincey et al., (2007), namely low (<2°)
surface slope, negligible ice flow (<10 m a$^{-1}$) and sustained glacier thinning.

To investigate the likely size of such a lake in the coming decades we consider two different scenarios of glacier thinning between 2015 and 2100 and follow a similar method to that of Linsbauer et al., (2013) to simulate glacier thickness into the future, but employ different criteria to determine future lake area. Our first scenario is based on the assumption that the
acceleration in glacier thinning in the Poiqu basin measured by King et al., (2019) is replicated by the year 2100. Such an increase in thinning will be driven by a further 1°C increase in temperature by 2100 (Kraaijenbrink et al., 2017), further to the ~1°C increase in temperature which has occurred in the central Himalaya (Maurer et al., 2019) since the 1970s. The second scenario is based on the premise that the increase in thinning which has occurred between 1974 and 2015 will be replicated over subsequent equivalent time periods (by 2056, 2097, etc). We extrapolated the thinning rates from King et al., (2019) and
integrated the resulting elevation changes between 2015 and 2100. We then assumed that once the glacier surface had lowered to a height below the hydrological base level of the glacier (4890 m a.s.l.), meltwater ponding would occur and that DEM pixels with an elevation of less than this threshold represented lake area at that point in time.

## 4 Results

For Jialongco, Galongco and the potential future lake, we focus below on results relating to the susceptibility of the lakes to
produce an outburst event, and the potential magnitude of downstream impacts, as simulated under worst-case scenarios. A
full hazard and risk assessment, including a complete range of outburst scenarios and vulnerability mapping, is beyond the
scope of this study.

### 4.1 Lake susceptibility and scenario development

The susceptibility component of GLOF hazard assessment establishes the likelihood of an event from a given lake, considering
the wide-ranging factors that can condition or trigger an outburst. The likelihood (which can be both qualitative or quantitative
for some hazards) is always specific to a given magnitude and valid for a given time frame, recognising that susceptibility can
evolve over time (Allen et al., 2022). Based on this assessment, scenarios for hazard modelling can be established, including
worst-case outburst scenarios as we focus on here. Taking a systematic approach (after GAPHAZ 2017), we compare the
relative susceptibility of the three lakes considered in this study, considering also how this susceptibility might evolve in the
future (Table 2). The table distinguishes those factors that condition and/or trigger an outburst event, while also linking to
those factors that inform about possible outburst magnitudes.

Located in a transitional zone to the north of the main Himalayan divide, the upper Poiqu basin is subject to heavy rainfall
during the Asian summer monsoon. With a significantly larger watershed area, Galongco is considered more susceptible to
heavy rain and/or snow melt leading to high lake water levels, and under future deglaciated conditions the lake may become
fed by a well-developed paraglacial stream network. However, even under these conditions, the relatively favourable dam
geometry (low width to height ratio and 15 m dam freeboard) suggests that the likelihood and magnitude associated with an
outburst via this triggering mechanism is low. Similarly, self-destruction via warm temperatures and melting of ground ice
within the moraine dam is extremely unlikely. Creeping permafrost features visible in the vicinity of Galongco, modelled mean
annual ground surface temperature (MAGST) (after Obu et al., 2019) and a partially hummocky appearance of the lake dam,
suggests a strong likelihood of a partially ice-cored moraine, but the huge width (> 200 m) and gentle downstream slope of the
dam would make a catastrophic failure in the case of thawing extremely unlikely.

As with the majority of large glacial lakes across the Himalaya (Liu et al., 2013; Richardson and Reynolds, 2000; Sattar et al.,
2021), the main triggering threat is considered to come from large slope instabilities, impacting into the lake. Under current
conditions, Jialongco is assessed to be most susceptible to ice avalanches, given the presence of a steep, highly crevassed
tongue positioned directly behind the lake (Fig. 2a). With an average slope of 35°, large transverse crevasses marking a sharp
break in topography, and likely temperate conditions at the bed, full collapse of the glacier tongue (~18 x $10^6$ m$^3$) is considered
a feasible worst-case scenario (Table 1). The mass would impact the lake in a direction parallel to the longitudinal axis of the
lake, leading to maximum overtopping wave heights and swashing effect. Smaller ice avalanches from this glacier have
triggered GLOFs from Jialongco in 2002, at a time when the lake was less than half of its current size (Chen et al., 2013).
While climate warming is expected to increase temperatures and meltwater at the glacier bed (Kääb et al., 2021), potentially
reducing the stability of the glacier, warming-driven retreat of the tongue will see a reduction in the potential avalanche volume
over time, and eventually, this threat will be eliminated completely as the ice retreats to a flatter plateau. In comparison, the
partially debris-covered parent glacier tongue of Galongco has a gentle mean slope (18°) and uniform gradient. Potential
unstable ice masses threatening Galongco, from steep ice cliffs and hanging glaciers, are found higher up on the mountain
(Fig. 2b), with estimated maximum volumes in the range of 0.1 – 1 x $10^6$ m$^3$. Avalanches from the larger of these starting

zones would strike the lake perpendicular to the longitudinal axis of the lake (from the west) meaning most of the energy from a displacement wave would be dissipated on the opposing side of the lake. It is a similar situation above the future lake, where small, and comparatively thin hanging glaciers are restricted to the slopes southwest of the potential lake (Fig. 2c).

Hence, a large rock or combined ice-rock avalanche is considered to be the most feasible mechanism capable of triggering a worst-case GLOF from either Galongco or the potential future lake. The northeast-facing slopes of Shishapangma rise nearly 3000 m above Galongco, and are likely to be mostly underlain by cold permafrost conditions. This is inferred both from the distribution of rock glaciers in the region, extending down as low as 4000 m a.s.l (Bolch et al., 2022), and modelled MAGST (Obu et al., 2019) (Table 2). However, the presence of ice cliffs and hanging glaciers can lead to thermal perturbations, and even melt conditions in otherwise very cold environment (Shugar et al., 2021). Based on close examination with high-resolution imagery, a large potential starting zone extending from 6550 – 7340 m a.s.l was identified on a heavily fractured slope beneath the south ridge of Shishapangma (Fig. 2b). Here, as in the surrounding peaks, layered leucogranite sits above sillimanite gneisses with a gentle northerly dipping schistosity (Searle et al., 1997). The slope has been eroded and potentially oversteepened by the glacier below. Based on structures outcropping on the face, a 40 m maximum bedrock depth was assumed, while steep ice cliffs and firn covering the slope is estimated to not exceed 10 m, resulting in a combined starting volume of 23 x $10^6$ m$^3$ (20% ice and 80% rock). The potential future lake is positioned directly beneath the ice-free ~ 2000 m high eastern face of Ramthang Karpo Ri (Fig. 2c), where MAGST is in the range of -3°C – -6°C. The face is dissected by numerous vertical structures and there is evidence of several scarps from previous instabilities. A large potential source area was identified, comparable to the Galongco scenario, with scarps on the face suggesting similar maximum depths of up to 40 m, leading to a total rock avalanche volume of 20.6 x $10^6$ m$^3$ (Table 1).

Even on a global scale, ice and/or rock avalanche volumes of the magnitude included in the scenarios here are rare (Kääb et al., 2021; Schneider et al., 2011), although have occurred recently (Shugar et al., 2021) and prehistorically (Stolle et al., 2017) in the Himalaya. While Poiqu basin is located within a high seismic hazard zone (Shedlock et al., 2000), it is notable that the 2015 Gorkha earthquake did not cause any large ice/rock avalanches in the Poiqu basin, despite significant damage in Nyalam and along the highway to Nepal (Kargel et al., 2016). Hence, given a lack of historical large instabilities in the basin, ice/rock avalanches of the magnitude included in this study are assessed to be low to very low likelihood events (see also Section 5 - discussion). Geologically there is little basis for distinguishing the likelihood of bedrock failures above the three lakes, and permafrost conditions are comparable (Table 2). Owing to the position of Jialongco directly beneath a steep glacier tongue, history of ice-avalanche triggered outburst events, and more unfavourable dam conditions (low freeboard, narrow width), we assess a worst-case outburst from this lake to be more likely than from Galongco under current conditions. Finally, all three lakes are or will be susceptible to instantaneous or progressive landslides occurring from the adjacent lateral moraines, most notably for Jialongco where active instabilities are clearly evident (Fig. 2a). Recent studies have shown that large lateral failures, either instantaneous or progressive, can be sufficient to initiate catastrophic process chains where dam geometries are sufficiently prone to erosion ( Zheng et al., 2021b).

**Table 2: First-order assessment of wide-ranging factors determining the susceptibility of glacial lakes (based on GAPHAZ 2017). An expert assessment of high (\*\*\*), moderate (\*\*), and low (\*) susceptibility for each of the factors is indicated. Factors not considered relevant for these lakes are indicated with (--). Factors can be relevant for conditioning (con.) and/or triggering (trig.) a GLOF, and can also have an influence on outburst magnitude (mag.).**


| Susceptibility factors for GLOFS | Relevance | | | Relevant Attributes | Susceptibility | | | Assessment methods and sources |
|---|---|---|---|---|---|---|---|---|
| | Con. | Trig. | Mag. | | Jialong Co | Galong Co | Future lake | |
| **a) Atmospheric** | | | | | | | | |
| Temperature | + | + | | *Mean temperature* | Increasing \*\*\* | Increasing \*\*\* | Increasing \*\*\* | Climate observations and projections (Ren et al., 2017; Sanjay et al., 2017) |
| | | | | *Intensity and frequency of extreme temperatures* | Increasing \*\*\* | Increasing \*\*\* | Increasing \*\*\* | |
| Precipitation | + | + | + | *Intensity and frequency of extreme precipitation events.* | Increasing \*\*\* | Increasing \*\*\* | Increasing \*\*\* | |
| **b) Cryospheric** | | | | | | | | |
| Permafrost (pf) conditions | + | + | + | *State of pf distribution and persistence within lake dam area and bedrock surrounding slopes* | No pf in dam area (MAGST > 1°C). Degrading pf in surrounding headwalls (< -3°C) \* | Likely ice-cored moraine dam (MAGST -1°C. Degrading pf in surrounding headwalls (< -4°C) \*\* | Possible pf in dam (MAGST -0.5 - -1°C). Degrading pf in surrounding headwalls (-3°C - -6°C) \* | Model-based results (Obu et al., 2019; Schmid et al., 2015); Google Earth |
| Glacier retreat and downwasting | + | | + | *Enlargement of proglacial lakes, enhanced supraglacial lake formation, dam removal or subsidence* | Lake currently at maximum extent. Glacier not in contact with lake -- | Minimal potential for further expansion (+1%) \* | Lake will be actively expanding over several decades, as overdeepening emerges \*\*\* | GlabTop; Landsat archive (Zhang et al. 2019); Google Earth; DEM differencing (King et al., 2019) |
| Advancing glacier (incl. surging) | + | | | *Formation of ice-dammed lakes* | Not relevant -- | Not relevant -- | Not relevant -- | Google Earth |
| Ice avalanche potential | | + | + | *Steep glacier tongue or ice cliffs, crevasse density and orientation, ice geometry* | High potential Steep heavily crevassed glacier tongue. Likely past events triggering a GLOF \*\*\* | Moderate potential Considerable steep cliff ice and small hanging glaciers \*\* | Low potential. A few small hanging glaciers \* | GlabTop; DEM slope analyses; Google Earth |
| Calving potential | | + | + | *Width of glacier calving front, activity, crevasse density* | Glacier not in contact with lake -- | Minimal potential (calving front = 300 m) \* | High potential (calving front = > 1km) \*\*\* | Google Earth |
| Lake size | + | | + | *Area, volume, and/or depth* | Mean depth: 64 m (lowered to 48 m) Volume: 40 (reduced to 23.5) x $10^6$ m$^3$ \*\* | Mean depth: 108 m Volume: 590 x $10^6$ m$^3$ \*\*\* | Mean depth: 46 m Volume: 70 x $10^6$ m$^3$ \*\*\* | Field based bathymetry; GlabTop for future lake |
| **c) Geotechnical and Geomorphic** | | | | | | | | |

| | | | | | | | | |
|---|---|---|---|---|---|---|---|---|
| Dam type | + | | + | *Bedrock, moraine, ice* | Moraine, now partially armoured *** | Moraine *** | Moraine *** | Google Earth |
| Dam width to height ratio | + | | + | *Width across the dam crest relative to the dam height* | 4:1 (engineered now to 8:1) *** | 9:1 * | 8:1 (large uncertainty) * | Google Earth; High resolution DEM analyses (Pleiades) |
| Freeboard (measured from the crest of the dam to the lake water level, irrespective of any outflow channel) | + | | + | *Elevation difference between lake surface and lowest point of moraine.* | ~ 20 m (engineered now to ~ 10) * | ~ 15 m ** | ~ 10 m (large uncertainty) ** | Google Earth; High resolution DEM analyses (Pleiades) |
| Downstream slope of dam | + | | | *Mean slope on downstream side of lake dam.* | Artificially armoured channel * | 10° * | 20° (large uncertainty) ** | Google Earth; High resolution DEM analyses (Pleiades) |
| Vegetation on dam | + | | | *Density and type of vegetation (grass, shrubs, trees).* | Partially armoured. Grass/scrub in other areas * | Absent *** | Absent *** | Google Earth |
| Catchment area | + | | | *Total size of drainage area upstream of catchment* | 9 km² ** | 35 km² *** | 10 km² ** | DEM analyses |
| Catchment mean slope | + | | | *Steepness of catchment area* | 32° ** | 28° ** | 29° ** | DEM analyses |
| Catchment drainage density | + | | | *Density of the stream network in catchment area* | Low density stream network to develop under deglaciated conditions * | Moderate density stream network to develop under deglaciated conditions ** | Low density stream network to develop under deglaciated conditions * | GIS based hydrological modelling |
| Catchment stream order | + | | | *Presence of large fluvial streams, facilitating rapid drainage into lake* | Low order streams to develop in future * | Moderate order streams to develop in future ** | Low order streams to develop in future * | GIS based hydrological modelling |
| Upstream lakes | | + | | *Presence and susceptibility of upstream lakes.* | None currently. Two small lakes (~0.01 km²) anticipated in future * | None currently or anticipated in future. -- | None currently or anticipated in future. -- | GlabTop; Google Earth |
| Rock avalanche potential | | + | + | *Steep, structurally unstable bedrock slopes with potential to runout into the lakes.* | Steep, heavily fractured slopes. Recent instabilities not evident. Scarps indicative of prehistoric failures ** | Steep, extensively glaciated slopes. Recent instabilities not evident. Scarps indicative of prehistoric failures *** | Steep, heavily fractured slopes. Recent instabilities not evident. Scarps indicative of prehistoric failures ** | GIS-based topographic potential modelling; Google Earth and high resolution imagery. |
| Moraine instabilities | | + | + | *Potential for landslides from moraine slopes into the lake* | Steep moraine and talus slopes > 400 m high. Large instabilities evident *** | Steep moraine slopes 100 – 200 m high. Minor instabilities evident ** | Steep moraine slopes in the order of 100 – 200 m anticipated ** | Google Earth |
| Seismicity | | + | | *Peak ground acceleration* | Very High 5.1 m/s² *** | High 4.1 m/s² *** | Very High 4.6 m/s² *** | Global Seismic Hazard Map (Shedlock et al., 2000) |

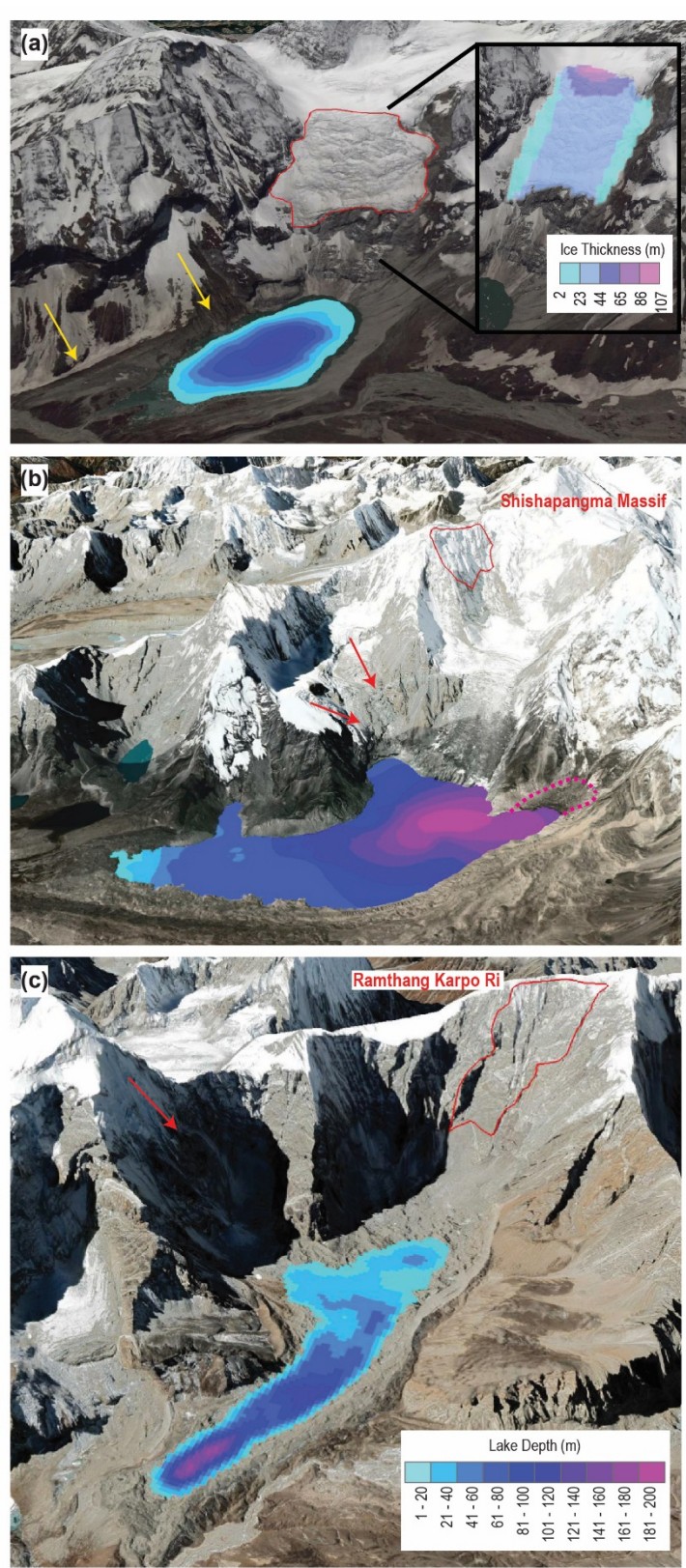

**Figure 2: Rock/ice avalanche starting zones (in red) used as input scenarios for the modelling of outburst flood process chains from the 3 lakes (see Table 1 for details). (a) Jialongco: The inset shows the GlabTop modelled ice thickness of the ice avalanche source area, and yellow lines indicate the steep lateral moraine walls also threatening the lake (see also Fig. 9). (b) Galongco: Large rock/ice avalanche source area outlined in red, while arrows indicate smaller sources areas of unstable ice, with possible future expansion of the lake shown by the dashed line. (c) Projected future lake: Large rock avalanche source area outlined in red, while arrow indicates**


**possible source area of smaller ice avalanches. Measured (and interpolated) lake bathymetry is shown in (a) and (b), with modelled**
**bathymetry of the future lake (c) derived from GlabTop. Background imagery from © Google Earth.**

4.2 GLOF modelling

Worst-case outburst scenarios for the three lakes were simulated until the border between China and Nepal (town of Zhangmu). The modelled flow does not extend beyond the border owing to the limited coverage of the required high-resolution Pleiades
DEM. Of the two current lakes assessed, the modelled peak discharge from Galongco is more than 5 times larger than that from Jialongco, leading to flow depths up to 14 m higher impacting the town of Nyalam (Table 3, Figs. 3 and 4). At the border, 20 km downstream, inundation depths are up to 17 m higher for the Galongco simulation as the large volume of water becomes constricted in the narrow topography of the valley, with discharge values remaining above 100,000 $m^3$ $s^{-1}$ even after 1 hour (Fig. 4). The simulated worst-case outburst from the potential future lake has a calculated peak discharge at the dam of 359,628
$m^3$ $s^{-1}$, resulting in flow depths (27 m) and discharge (163,667 $m^3$ $s^{-1}$) in Nyalam that would exceed that of Jialongco, but are an order of magnitude lower than from Galongco. Differences in the shape of the outflow hydrographs at the dam (Fig. 4a), and travel distance, lead to minor variations in the arrival of the modelled flood waves in Nyalam and further downstream at the border with Nepal. The flood wave from Jialongco first registers after 6 minutes in Nyalam, with the maximum flow heights arriving 2 minutes later (all times relative to the initial avalanche release). In contrast, the flood wave from Galongco
first registers after 10 minutes, with maximum flow heights arriving 4 minutes later. An outburst from the potential future lake has a similar arrival time of only 11 minutes in Nyalam, while all simulated outbursts reach the Nepalese border within a range of 28 - 32 minutes after the avalanche release. Notably, the remedial measures undertaken at Jialongco, which have lowered and armoured the lake dam (erosion set to zero in the model – see section 3.2), result in a slightly larger initial peak discharge (Table 3, Fig 4) because there is a greater splashing effect and larger overtopping volume owing to the reduced freeboard.
Downstream, the simulated GLOF then attenuates at a slower rate (54% decrease in discharge between Nyalam and Zhangmu) compared to the simulation for the original lake (81% decrease in discharge between Nyalam and Zhangmu) (Fig 4).

Potential processes that could further enhance the GLOF magnitude include entrainment of large volumes of sediment along the flow path leading to additional bulking of the flow volume, blockages of a river by GLOF deposits leading to secondary
outburst events, and a process chain involving more than one lake. Significant erosion of sediment from within the main river channels is considered unlikely for any of the three outburst scenarios, given that average trajectory slope angles measured along the flow paths are well below those needed to entrain sediment from within a channel (Huggel et al., 2004). However, undercutting, erosion and destabilisation of the river banks as a result of the GLOF means that such secondary hazards cannot be excluded, particularly in the steep sided gorge downstream of Nyalam. Immediately below Nyalam, the valley narrows,
leading to pooling of water in the simulations, and a backwash effect is produced that extends 2 km up the Poiqu river, with maximum flow depths of >60 m under the Galongco scenario (Fig. 3a). Significant deposition of sediment can be anticipated within this backwash zone, with the potential to block the Poiqu river and form a major secondary hazard, in line with processes observed and modelled during the 2021 catastrophic mass flow in Chamoli, northern India (Shugar et al., 2021).


**Table 3: Measured and modelled lake and outburst flood parameters for Galongco (GL), Jialongco pre-lowering (JC), Jialongco post-lowering (JC-L), and the future lake (FL). All timings are relative to the start of the initial rock and/or ice avalanche.**

| | GL | JC | JC-L | FL |
|---|---|---|---|---|
| Lake area (km$^2$) | 5.46 | 0.62 | 0.49 | 1.54 |
| Mean lake depth (m) | 108 | 64 | 48 | 46 |
| Lake volume ($10^6$ m$^3$) | 590 | 40 | 23.5 | 70 |
| Dam breach width (m) | 850 | 650 | -- | 420 |
| GLOF peak at dam (m$^3$ s$^{-1}$) | 585,686 | 92,421 | 101,919 | 359,628 |
| Time of arrival at Nyalam | 10 min | 5 min | 6 min | 11 min |
| Flow depth at Nyalam (m) | 37 | 23 | 23 | 27 |
| Flow discharge at Nyalam (m$^3$ s$^{-1}$) | 221,655 | 64,124 | 77,695 | 163,667 |
| Time of arrival at Zhangmu | 28 min | 32 min | 28 min | 30 min |
| Flow depth at Zhangmu (m) | 29 | 7 | 12 | 14 |
| Flow discharge at Zhangmu (m$^3$ s$^{-1}$) | 170,404 | 12,251 | 35,389 | 54,656 |

In contrast to previous modelling results for Galongco (Shrestha et al., 2010; Zhang et al., 2021), the worst-case avalanche triggered GLOF path is not confined to the existing river channel, overtopping the orographic-right side of the valley (bounded by old moraines) and spilling over into Jialongco to form a second, larger flow path towards Nyalam (Fig. 3a). The two paths converge again about 6 km upstream from Nyalam. The hyper-elevation of the flow that enables this overtopping is consistent with observations of catastrophic mass flows of comparable magnitudes (Shugar et al., 2021). Results further indicate that an outburst event from the potential future lake could slam into, pool up, and eventually overtop the lateral moraine of Jialongco, producing a potential chain reaction where Jialongco also breaches (Fig. 5). Despite adding volume to the flow, the presence of Jialongco with its prominent lateral moraine acts as a topographic obstruction that slows and reduces the energy of the outburst event, with a 50% reduction in discharge values measured immediately upstream and downstream of Jialongco. Although only one specific cascading lake interaction, this example highlights that lakes positioned downstream of another lake do not necessarily increase GLOF hazard, depending upon the downstream lake geometry and its orientation relative to the incoming GLOF path.

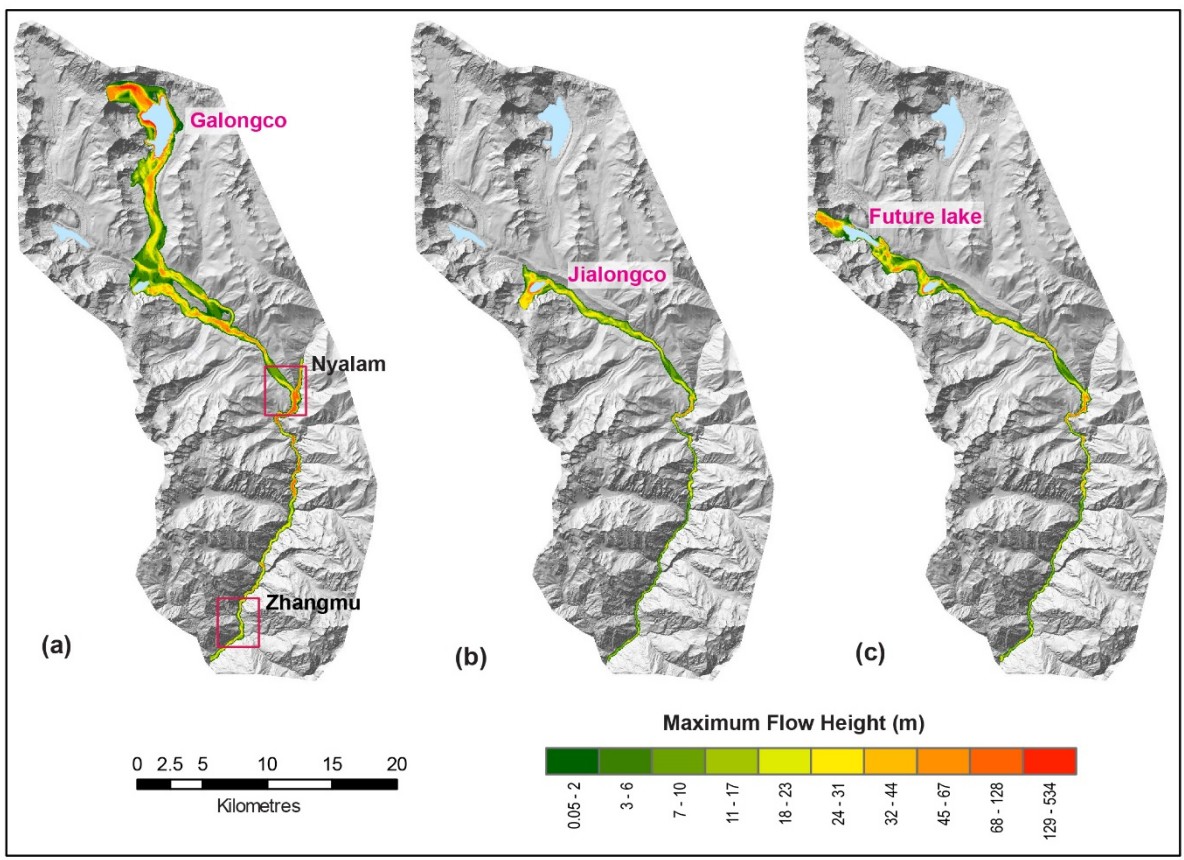

**Figure 3: Modelled GLOF flow heights for worst-case scenarios from (a) Galongco, (b) Jialongco (JC-L), and (c) the potential future lake. The location of Nyalam and Zhangmu towns are indicated by the red boxes in (a).**

405

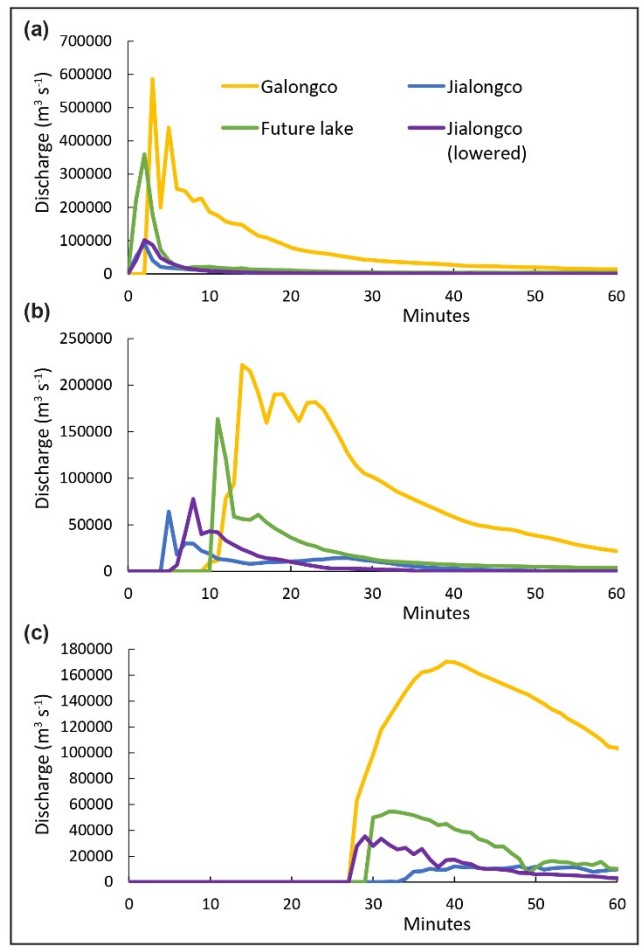

**Figure 4: Modelled GLOF discharge for three assessed lakes taken at (a) the lake dam, (b) Nyalam and (c) Zhangmu.**

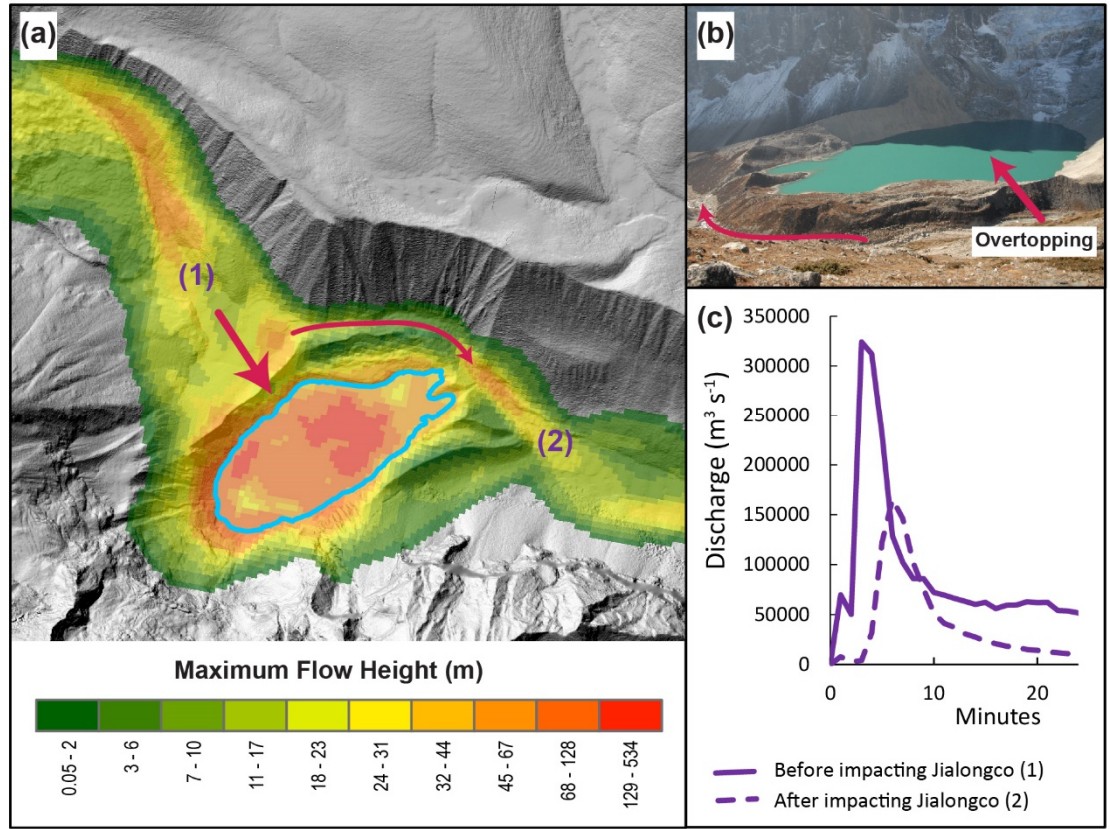

**Figure 5: (a) Modelled GLOF flow heights for an outburst event from the potential new lake, showing area of pooling and overtopping into Jialongco. Background DEM generated based on Pleiades data 15 Oct 2018 © CNES and Airbus DS. The moraine height at the point where overtopping is illustrated in the photo (b) is around 40 m (Photo: O. King, October 2018). (c) Flow hydrographs immediately upstream (1) and downstream (2) of Jialongco are simulated with r.avaflow. Note that the simulation is based on the post-lowering lake bathymetry and dam geometry of Jialongco (JC-L).**

4.3 GLOF impact and exposure

We identify from Open Street Map and Google Earth imagery, the buildings in Nyalam exposed to different GLOF intensity levels according to simulated debris flow intensities (after GAPHAZ 2017). While classification schemes vary across countries, land areas potentially affected by high flood or debris flow intensities (calculated on the basis of flow heights and/or flow velocities), are typically considered as high hazard zones even for low probability events (GAPHAZ, 2017). In Nyalam, lower flow heights associated with an outburst from Jialongco result in marginally lower levels of exposure compared to simulated events from Galongco or the potential future lake (Fig. 6). Despite the majority of buildings in Nyalam being located 10 – 20 metres above the river channel, where they have been unaffected by past outburst events from Jialongco (Chen et al., 2013), there is clearly significant exposure within the high intensity zone of a worst-case outburst. Furthermore, the rapid expansion of infrastructure along the river banks north of the main settlement over the past several years has significantly increased the built area exposed to potential GLOF events, with many new buildings located in the high intensity flood zone. Overall, levels of exposure are comparable for simulated outbursts from both Galongco and the potential future lake, with both worst-case events also likely to disrupt the main highway and bridges linking to the town.

Downstream from Nyalam in the reach to the border with Nepal there are few buildings located along the river bank, and the main threat is to the 38 km stretch of the transnational highway, of which the proportion affected by high-intensity flood levels

is 27% and 40%, for modelled outbursts from Jialongco and Galongco respectively (and 28% for the future lake scenario). While we did not simulate beyond the border, previous events (e.g., Cook et al., 2018; Wang et al., 2018), and assessment

435    studies (Khanal et al., 2015a; Shrestha et al., 2010) have highlighted the significant risk to Nepalese communities, hydropower stations, and other infrastructure located along the banks of the Bhotekoshi river.

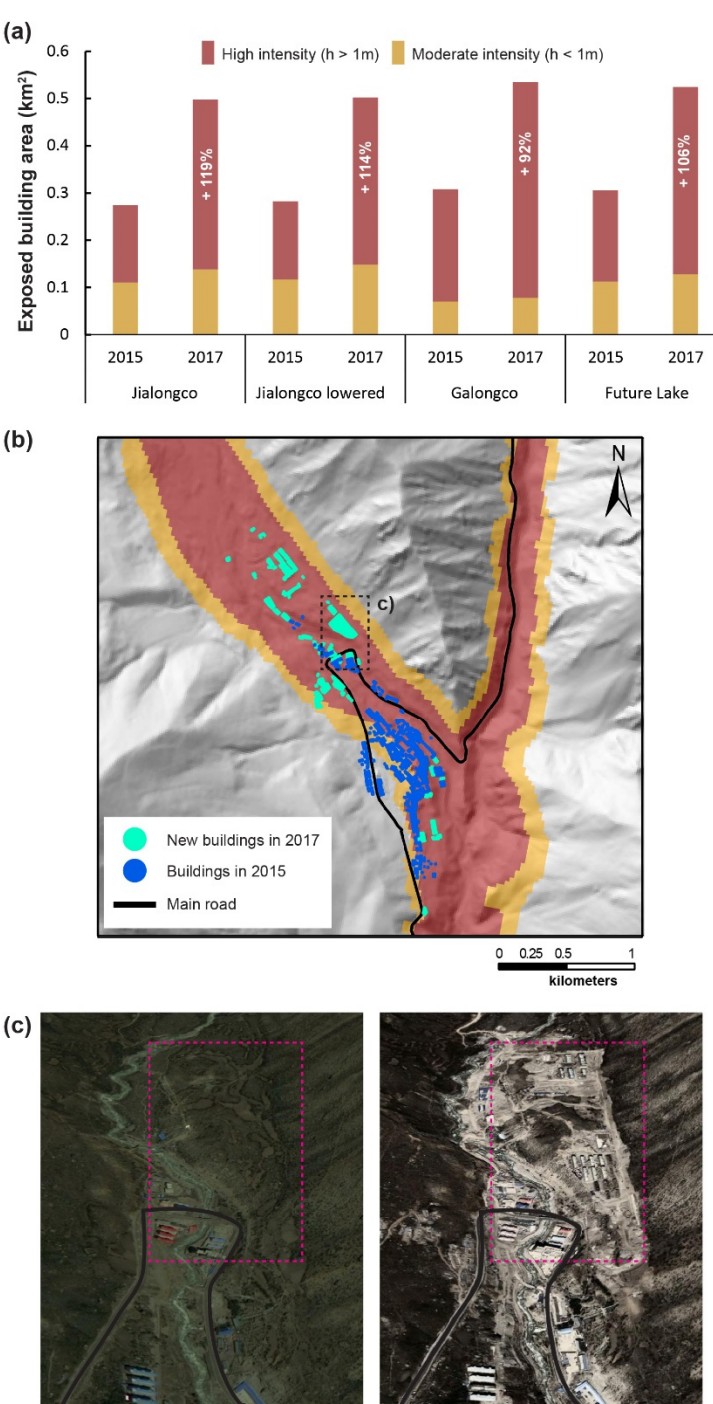

**Figure 6: (a) Built area in Nyalam exposed to modelled GLOF intensity levels for the three assessed lakes, showing the effect of rapid**
440    **infrastructural development between 2015 and 2017. The percentage indicates the increase in built area within the high intensity zone. (b) Modelled intensities for the Galongco outburst scenario showing the recent expansion of infrastructure, as mapped from © Google Earth imagery from June 2015 (c - left) and October 2017 (c - right). A notable area of infrastructure development just upstream of the main bridge is highlighted in the dashed rectangle.**

## 4.4 Trajectory of future lake development

The thinning of glacier RGI60-15.09475 over at least the last four decades has caused the development of a glacier surface that is well suited for supraglacial lake development (Fig. 7). The central 2.5 km of the glacier's ablation zone, where supraglacial ponds are already forming, is effectively stagnant, very gently sloping and has become heavily pitted due to differential ablation in response to spatially variable debris thickness. These conditions will enable the further expansion of the supraglacial pond network, which is unlikely to drain quickly.

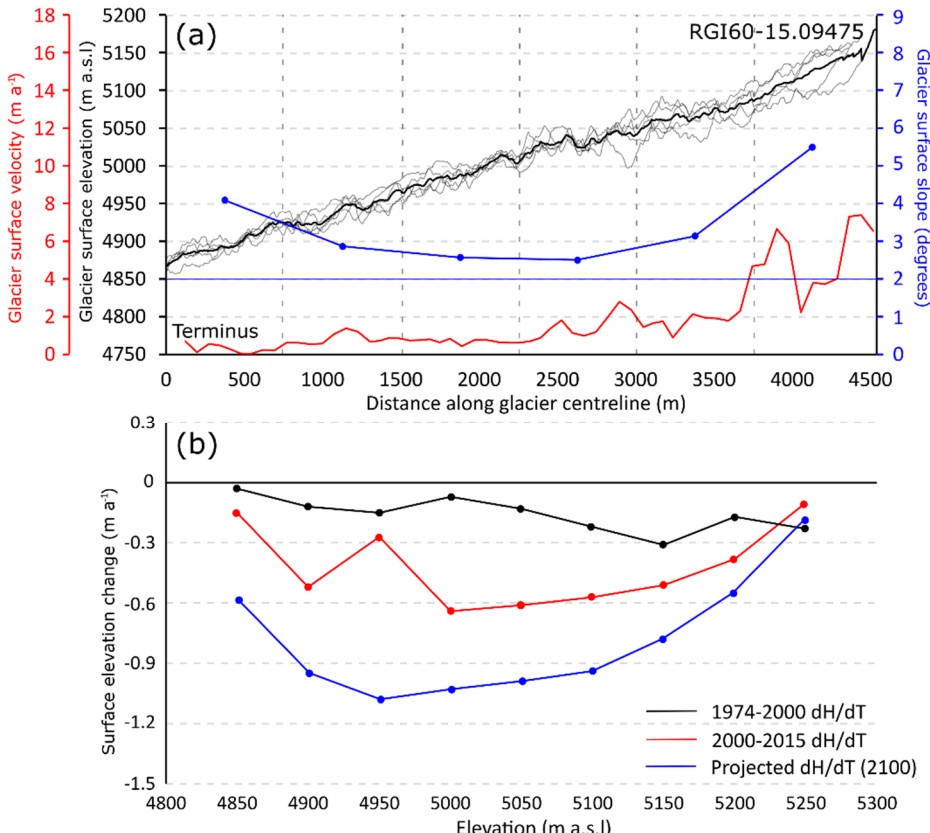

**Figure 7: (a) Surface topography, slope, and velocity regime of glacier RGI60-15.09475 in 2017/18. Widespread meltwater ponding is expected once glacier surface slope declines to ~2° and little flow is evident to allow for crevasse formation and meltwater drainage. (b) Surface elevation change over the glacier from DEM differencing over the period 1974-2000 and 2000-2015 and the rate of elevation change projected to occur by 2100 (Scenario 1). The same gradient of thinning is assumed to occur by 2056 and be replicated again by 2097 in Scenario 2.**

The extrapolation of thinning measured over the last four decades over glacier RGI60-15.09475 suggests that a large portion of the glaciers surface will soon sit below an elevation where supraglacial meltwater would normally drain from the glacier surface, allowing for the development of a supraglacial lake. Under scenario 1 (1974-2015 thinning replicated by 2100), 0.6 km² of the glaciers surface will be below the hydrological base level of the glacier by 2100 (Fig. 8). The majority of this area will be located within 1 km of the glacier's terminal moraine, although some small areas further up-glacier will also sit below the hydrological base level by 2100 due to the glacier's inverse ablation gradient (Fig. 8). Under scenario 2 (1974-2015 thinning replicated by 2056, 2097), up to 1.33 km² of the surface of glacier RGI60-15.09475 will sit below the hydrological base level of the glacier by 2100. Hence, a large portion of the glacier surface above the 1.54 km² overdeepening identified by GlabTop (Table 3) will have become susceptible to supraglacial lake expansion and proglacial lake formation by 2100 (Fig. 8d).

Projected thinning exceeds the ice thickness estimated by GlabTop in current ablation hotspots, most notably towards the terminus of the glacier, where the future ice surface elevation is similar to the simulated bedrock elevation by 2070 under scenario 1 and 2045 under scenario 2. Extrapolated thinning does not match the estimated ice thickness over the majority of the area of the proposed overdeepening further up glacier, where GlabTop suggests ice could be up to 230 m thick.

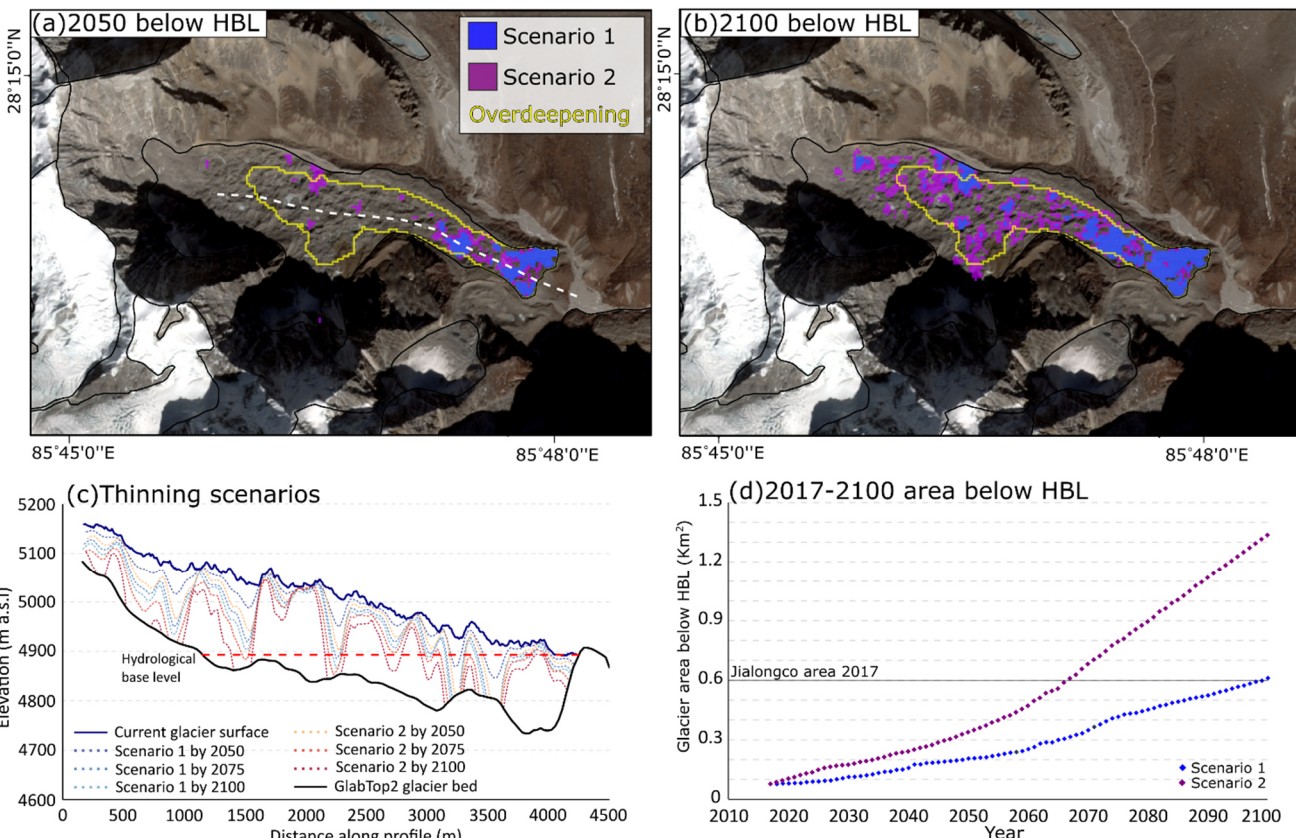

**Figure 8: Meltwater ponding (if elevation < the hydrological base level of the glacier) by 2050 (a) and 2100 (b) under different scenarios of thinning for glacier RGI60-15.09475. Background imagery is modified Copernicus Sentinel data (14-10-2021), processed by the European Space Agency. Glacier surface elevation profiles taken along the dashed profile on panel (a) under each scenario of thinning are also shown in panel (c). The full timeline of supraglacial lake area expansion is shown in (d). The area within the yellow polygon shows the location of a bed overdeepening (1.54 km$^2$) predicted by GlabTop. Ice flow is from left to right in a-c.**

## 5 Discussion

The results from this study demonstrate how GLOF hazard assessment at the basin-scale can be expanded to consider new threats that may develop in the future. In doing so, this study has taken established approaches for lake susceptibility assessment (GAPHAZ 2017) and GLOF modelling (Mergili et al. 2017) and applied these approaches to consider also an outburst scenario from a potential future lake. To the extent possible, the assessment was based on freely available data and imagery. However, in steep, mountain topography such data can have limitations, and a high-resolution DEM derived from Pleiades imagery was required to achieve accurate GLOF modelling results for Poiqu River basin. While not intended to substitute the comprehensive multi-scenario modeling and field-based hazard mapping that needs to support decision-making (e.g., Frey et al., 2018), the results from this study provide an intermediary step for disaster risk management planning. Using the tools and approaches demonstrated here, authorities can effectively bridge the knowledge gap between the known threats to which they may already

be responding, and those potentially much larger, yet poorly constrained threats that are anticipated to emerge or become more likely in the future.

### 5.1 Comprehensive approach to disaster risk management

For the Poiqu basin, these results come at an opportune time, given that local authorities over the past years have initiated major engineering work at Jialongco (Fig. 9). In principle, the focus of authorities on Jialongco is supported by the results of this study, which indicate that the lake has the greatest likelihood of producing a large GLOF that threatens the village of Nyalam, and, under a worst-case scenario, will lead to significant flood heights and discharges downstream in Nepal. While assessed to be less likely, a large rock/ice-avalanche triggered outburst from Galongco would result in a higher intensity flood event, with discharge values in Nyalam almost 3 times larger than those simulated for Jialongco. At the border with Nepal (Zhangmu), our simulations reveal potential peak discharges in the range of $35,000 - 170,000 \, \text{m}^3 \, \text{s}^{-1}$ under worst-case scenarios, which is more than 15 times larger than indicated by earlier modelling studies (Shrestha et al., 2010), suggesting that previously estimated potential property losses of up to US\$197 million in downstream communities of Nepal are far lower than what could feasibly occur. In comparison with past events, the 1981 outburst from Cirenmaco, resulting in around 200 fatalities and up to US\$4 million damage, had an estimated peak discharge of around $10,000 \, \text{m}^3 \, \text{s}^{-1}$ in Zhangmu (Cook et al., 2018; Wang et al., 2018), while the 2016 event from Gongbatongsha lake was about half this magnitude again, but resulted in economic losses of > US\$ 70 million, but no loss of life (Sattar et al., 2022).

Despite the threat the lake poses, the focus at Jialongco on hard engineering strategies to reduce GLOF risk could prove both costly and inefficient, if not complimented by a more comprehensive and forward-looking strategy that considers large process chains and appropriate response actions. The removal and armouring of much of the frontal moraine and construction of a stable outlet channel (Fig. 9) would have only a minimal effect on the potential downstream GLOF magnitudes resulting from a catastrophic ice avalanche into the lake (Figs. 4 and 6). On the one hand, the engineering work has reduced the amount of moraine material available for initial erosion (leading to a more rapid and slowly attenuating water-dominated flow), while on the other hand, the reduction in freeboard has left the lake more susceptible to overtopping, resulting in a larger volume GLOF event (Fig 4a). The simulations also reveal the limited potential for early warning in the case of large process chains, with catastrophic GLOF discharges reaching Nyalam in only 5 – 11 minutes following an ice and/or rock avalanche detaching. For downstream communities in Nepal, warning times under worst-case scenarios could be as little as 30 minutes, which is a significant reduction on previous estimates of up to 2 hours in the case of Galongco, whereby a more gradual lake breaching mechanism was modelled (Zhang et al., 2021). Particularly in transboundary regions requiring communication and collaboration between countries before any alert is acted upon, minutes lost or gained can be critical for effective early warning and evacuation.

Given the demonstrated minimal effect that lake lowering would have on a potentially devastating, worst-case GLOF from Jialongco, and the fact that warning times for all 3 assessed process chains would be minimal in Nyalam, we argue that a focus on engineering measures and early warning systems needs to be coupled with effective land use zoning and programs to strengthen local response capacities (e.g., Huggel et al., 2020). Such a comprehensive strategy would reduce the risk not only from an outburst from Jialongco, but also provide future-proofing against larger outburst scenarios from Galongco or potential new lakes that develop over the next century. In general, increasing exposure of people and assets is seen as a main driver of disaster risk in mountain regions (Hock et al., 2019), and this is clearly evidenced through the rapid increase in built infrastructure upstream of Nyalam, directly within the high-intensity zone of potential worst-case GLOF events (Fig. 6), but

also within the path of more moderate events (Zhang et al., 2021). Lowering of the water level in Jialongco has likely reduced the threat to these buildings from a smaller, higher probability outburst event, but similar action would need to be repeated at Galongco and as new lakes emerge in the future, in order to maintain this minimum level of protection, while doing little to reduce the risk of a larger worst-case event. Land use zoning is therefore urgently required, in order to regulate the future development of infrastructure occurring within high hazard zones, also considering worst-case scenarios. Furthermore, framing

any EWS within a broader catchment-scale monitoring program could enable a degree of forecasting, allowing alert levels to be raised and evacuation preparations then initiated within high hazard zones, prior to a warning system being activated. For example, precursory movement associated with recent large high mountain slope failures has been detected with optical or InSAR satellite data (Bhardwaj and Sam 2021; Carla et al., 2019), and through dense seismic monitoring networks (Tiwari et al., 2022), although real-time operational monitoring systems are rare and remain an important research priority.


5.2 Worst-case scenarios and future climate change

While GlabTop and other similar modelling approaches (see Farinotti et al., 2019a) have been widely used to anticipate future glacial lake locations and assess related risks and opportunities (e.g., Farinotti et al., 2019b; Haeberli et al., 2016; Magnin et al., 2015), large uncertainties remain as to if and when specific overdeepenings will transition into lakes. In this study, we have

focussed on a very large overdeepening positioned beneath a flat, heavily debris-covered glacier tongue – a classic geomorphological setting in which large proglacial lakes typically develop (Benn et al., 2012; Haritashya et al., 2018), and analogous to the setting of Galongco. Coupled with the fact that conditions at the surface of the glacier have already allowed supraglacial lakes to form in the ablation zone of the glacier, there can be a high degree of confidence that a future proglacial lake will develop in this location, trapped behind the prominent terminal moraine. Under the two thinning scenarios employed

in this study, supraglacial lake area equivalent to the current area of Jialongco will be replicated on glacier RGI60-15.09475 by ~2070 to 2100 (Fig. 8). These estimates may still represent a conservatively slower trajectory of lake development on this glacier. Both the development of extensive supraglacial ponds and ice cliff networks and the transition of a supraglacial lake to a full depth proglacial lake can increase the overall thinning rate in the ablation zone of debris-covered glaciers (King et al., 2020; Mölg et al., 2020; Thompson et al., 2016). Our simple extrapolation of current thinning rates and patterns does not

account for the initiation or expansion of these ablative processes. Therefore, we would rather expect greater thinning than our results predict in the lowermost ~1.5 km of the glacier over coming decades once a substantial amount of meltwater has ponded at the glaciers surface.

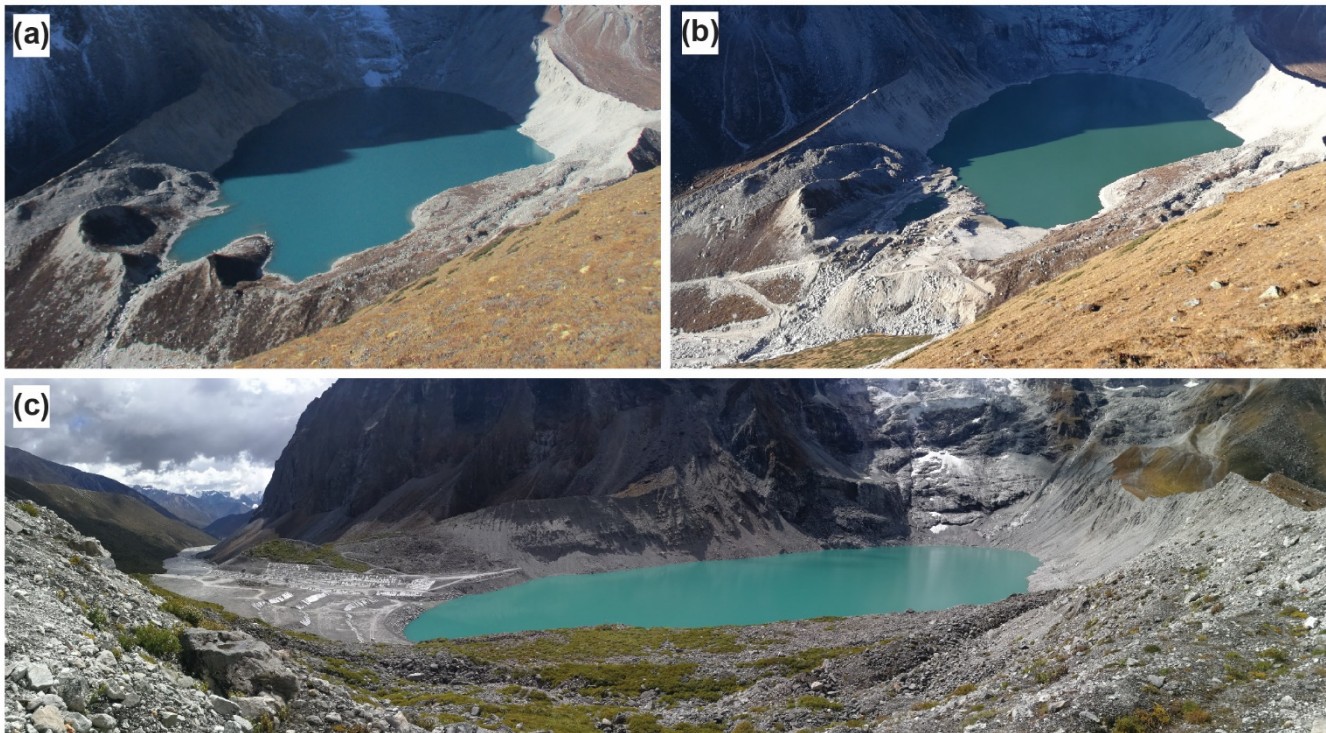

**Figure 9: Images taken of Jialongco in (a) October 2018 showing the natural state of the lake, (b) October 2020 and (c) September 2021, showing the engineering work that has been undertaken in the outlet area, lowering the lake level, removing much of the frontal moraine, and establishing a stable, armoured outlet channel. Photos: T. Bolch (a) and G. Zhang (b, c).**

Regardless of uncertainties in the timing of future lake development, the results from this study suggest that hazard mapping
and associated response planning that accounts for existing worst-case outburst threats from Jialongco, and particularly Galongco, would largely remain valid for the future lake scenario. In other words, the potential magnitude of a worst-case GLOF from Galongco far exceeds anything the future lake could produce, while a worst-case event from Jialongco has the fastest arrival time in Nyalam. However, the formation of the new lake, and others, will undoubtedly increase the likelihood of a large magnitude event occurring within the basin, and hence, risk levels to people and infrastructure will increase if
response strategies are not adequate. One of the key challenges in glacial hazard research is assigning a likelihood or probability to outburst scenarios, particularly for such very large scenarios for which there may be no historical precedence in a given basin (Allen et al., 2022). The worst-case scenarios modelled here are an order of magnitude larger than observed or assessed under previous studies (Shrestha et al., 2010; Zhang et al., 2021), but for the first time consider potential process chains involving large rock/ice avalanches > 20 million m$^3$ striking glacial lakes. The resulting GLOF discharges and flow heights
produced by such catastrophic process chains modelled here are certainly extreme, with a return period exceeding 200 years (Carrivick et al., 2016) or possibly more (Veh et al., 2020) relative to documented discharge values from past GLOFs in Asia. However, the recent Chamoli disaster, and earlier events from Seti River, remind that large avalanches capable of triggering such a process chain in the Himalaya do occur (Shugar et al., 2021), and their frequency is expected to be increasing as permafrost warms and slopes destabilise (Haeberli et al., 2017). Combined with larger and more numerous lakes (Zheng et al.,
2021a), the likelihood of large-magnitude process chains occurring must be increasing over time, and therefore these more extreme scenarios, even if beyond historical precedence, need to be considered under a comprehensive approach to risk management.

GAPHAZ (2017) draws on the example of Switzerland, where very low probability but large magnitude hazard events are typically included within a zone of "residual danger", that extends to include events with a return period beyond 300 years. Ultimately, the probability threshold used to define such a zone, and the regulations or response strategies applied within that zone, need to be well-aligned to local societal values and risk tolerance levels. Some desirable response strategies may have a low opportunity cost, such as ensuring evacuation centres and other critical infrastructure (e.g. schools, police, medical facilities) are positioned well out of the zone of residual danger, while other strategies may come with higher social, environmental and economic costs. At the very least, there needs to be awareness and communication of the residual risk those living in such a zone face, ensuring that strategies such as lake lowering do not lead to a moral hazard and maladaptation, particularly in view of future climate change and emerging threats.

## 5 Conclusions

The Poiqu basin in the central Himalaya has been well established as a hotspot from which transboundary GLOF threats can originate. In the current study, we have focused on two lakes that directly threaten the Tibetan town of Nyalam and areas downstream, comparing the likelihood, potential magnitude, and impacts of very large outburst events from these lakes. In addition, a future scenario has been modelled, whereby an outburst was simulated for a potential new lake, anticipated to form upstream of Jialongco. For all lakes, worst-case scenarios were assessed, with large rock and/or ice avalanches striking the lakes to trigger GLOF process chains. The study has recognised that:

- Jialongco, although smaller in size, poses the most immediate threat to Nyalam and downstream communities, owing to its position beneath a steep, heavily crevassed glacier tongue, and history of outburst events. Even though recent engineering work has lowered the lake level by an average of 16 metres and stabilised the dam area, this has minimal effect on the magnitude and arrival time of a simulated worst-case GLOF triggered by a large ice avalanche.

- The likelihood of a large rock/ice avalanche >20 mil m$^3$ striking Galongco is considered very low, but increasing as permafrost slopes warm. The process chain would generate extreme GLOF discharges up to 5 times larger than simulated for Jialongco, resulting in flow heights up to 14 and 17 metres higher in Nyalam and at the border with Nepal (Zhangmu) respectively.

- The assessed future lake could obtain a size comparable to Jialongco by 2070, but possibly earlier as a result of ablative processes around supraglacial ponds and ice cliffs on the debris-covered tongue. Even once the lake obtains its full potential area and volume, a worst-case rock avalanche-triggered outburst will have peak discharges and flow heights that are an order of magnitude lower than what Galongco can produce, but larger than for Jialongco.

- For all three assessed lakes, worst-case outburst events would impact Nyalam within 5-11 minutes of the process chain initiating, while reaching Zhangmu in around 30 minutes, posing severe challenges for early warning and evacuation. While previous studies have focused on rapid lake expansion in the region, for the town of Nyalam, it is rather the expansion of infrastructure directly within the high-intensity flood zone from both current and future lakes that has significantly increased GLOF exposure levels.

On the basis of these findings, a comprehensive and forward-looking approach to disaster risk reduction is called for, including early warning systems, effective land use zoning that is in line with local risk tolerance levels, and programs to build local response capacities. Relying only on hard engineering strategies at the lake source could prove insufficient, as such strategies

do not address underlying exposure and vulnerability to GLOFS and other geohazards, and are demonstrated to be ineffective in the face of worst-case, catastrophic outburst events.


**Author contribution**

SA and AS designed the study and undertook the GLOF modelling, and hazard assessment. OK performed the modelling of future lake development. AB produced the high resolution Pleiades DEM. SA, OK, TB, and GZ provided insights, images and
bathymetry data from field visits. SA, TB, and TY acquired the project funding. All authors contributed to the drafting and revision of the manuscript.

**Acknowledgement**

This work was supported by the Swiss National Science Foundation (IZLCZ2_169979/1) and by the Strategic Priority
Research Program of the Chinese Academy of Sciences (Grant No. XDA20100300). We thank Adam Emmer, Fabian Walter, and a third anonymous reviewer for their extremely comprehensive and constructive comments.

**Competing interests**

The authors declare that they have no conflict of interest.

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
