# Peer review of "Glacial lake outburst flood hazard under current and future conditions: worst-case scenarios in a transboundary Himalayan basin"

_Natural Hazards and Earth System Sciences, 2021_

## Referee Comment (RC1)

Dear Editors,

Dear Authors,

thank you for giving me the opportunity to review the manuscript nhess-2021-167 entitled "Glacial lake outburst flood hazard under current and future conditions: first insights from a transboundary Himalayan basin" by Simon K. Allen and co-authors. In this study, Allen et al. considered worst-case scenarios to model outbursts from three glacier lakes (among those one that might appear in the future) in the Poiqu basin, Chinese Himalaya. These simulations are thought to give guidance in decision and designing early warning systems to mitigate some of the impacts that GLOFs have repeatedly brought to the Poiqu basin in the past. Using the modelling software Hec-RAS, the authors found large differences in maximum flow depths and arrival times of the simulated flood waves at Nyalam, a rapidly growing settlement in close vicinity to the Nepalese border. These findings demonstrate the transboundary hazard that GLOFs may pose to mountain communities. The manuscript is well-written, offers good illustrations, and is (with few exceptions) clearly understandable, also to a broad audience.

**Major comments**

Allen et al. used a suite of well-established tools to assess possible trigger mechanisms of GLOFs, and to model hydro-mechanical characteristics and runouts of the flood waves. Unfortunately, the authors missed the opportunity to couple such triggers with the subsequent failure of moraine dams. Recent software such as r.avaflow have recently improved the understanding of GLOF processes[1–3], allowing for pseudo-probabilistic assessments of GLOF triggers and impacts under a range of adjustable and testable boundary conditions. However, given that the authors considered only three worst-case scenarios of moraine-dam failure, largely decoupled from the initial triggers, I see only limited progress in calculating the likelihood of observing a GLOF of a given magnitude, though this is repeatedly stated in the manuscript. Clearly, the authors present a thorough expert-based assessment of potential conditioning or triggering factors. Yet it remains unclear how this traditional, rather subjective procedure brings GLOF hazard and risk assessment forward with regards to the many previous case studies that we saw in the past years[4–13]. Thus, the authors' goal to estimate "the core hazard dimensions of GLOF magnitude and likelihood (or probability)" (L245-246) only remain partly fulfilled in my opinion. If this manuscript is intended to improve early warning and risk management, then the reader (at best a person involved in such tasks) might expect a more objective use of the terms hazard or probability, that is a numeric value between zero and one. A probability or return period might provide a robust baseline for decision makers compared to the current distinction between a 'high' or 'low' level of susceptibility or impact. Fortunately, most of the data (including high-resolution DEMs and satellite images) for such an extended appraisal already come with this manuscript. I thus would like to encourage the authors to revise their manuscript accordingly, considering a much wider range of scenarios in order to proceed towards a more objective assessment of current and future GLOF hazard in their study region.

**More detailed and technical comments**

In the following line-by-line comments, I wish to add more details to these issues. I also point at locations that could either deserve more attention or that could be shortened.

- L6: 'far reaching': in terms of run-out? Social consequences? Media coverage?

- L10: 'well-known dangerous': please consider avoiding subjective terms.
- L26-27: 'Based on … capacity building programs': Hardly mentioned in the text, consider revising or deleting.
- L39: 'rapidly in both size and number': how much? Consider using recently multi-temporal glacier lake inventories[14,15].
- L40: What is the frequency of GLOFs in the Himalaya? If I am not mistaken, most researchers deem GLOFs a low rate of occurrence.
- L47f: 'Such lakes can have volumes >100 m³', according to ref. [16]
- L69: 'within the eastern Himalayan region': why there?
- L75: 'are lacking': suggest to tone this a bit down, as it undermines previous appraisals[17–19].
- L79-82: Is this statement valid for all glaciated high mountains? Or only the Himalayas?
- L89-91: The last sentence could be deleted, if shortening is needed. The introduction ends well with a strong presentation of the research goals.
- L93: 'considering potential GLOF impacts'
- L97: 'mean annual air temperature'
- L99: are these precipitation rates (I think these are monthly?) the maximum or the average recorded in a given period?
- L105: 'GLOF danger': replace with 'GLOF hazard'?
- L107: 'losses of up to US$4 million': is this in today's currency?
- L111-112: 'a large and persistent threat, considered as one of the most dangerous lakes in Tibet': suggest to tone this down again. Given that previous appraisals used different criteria and thresholds, we may wish to avoid confusion what is now the 'most dangerous lake' in a given region.
- Figure 1: Please highlight the border between Nepal and China.
- L139: So estimating lake depth comes with zero uncertainty?
- L150: To my knowledge GlabTop models the glacier thickness in a bedrock topography. Did the authors add the moraine on top of this bedrock depression? How high / wide is this moraine? If not, how susceptible is a bedrock depression to overtopping? Did the authors consider sediment input from supraglacial debris into the lake?
- L155: Please consistently use Fig. / Fig / Figure (or any other option) throughout the manuscript. Sometimes the abbreviation is written with a dot, sometimes not.
- L154-164: the PFV approach may underestimate the amount of water that is generated from mass flows entering the lake and causing a splash wave. Given that the authors consider ice avalanches and landslides entering the lakes, how useful is it to use the PFV approach?
- L156: The PFV concept further seems to confuse mean and maximum depth. Why should the drain empty completely if the depth of the breach is equal to the mean depth of the lake? Again, the assumption of a fixed breach depth might simplify (or even underestimate) the worst-case scenario of GLOF volumes.
- L162-163: 'In comparison, … (Xu, 1988)': what is this comparison good for?
- L165: 'the DEM accuracy is unknown': what is the accuracy of the 1-m DEM then?
- L171-172: Again no uncertainties in these equations? Propagating estimates through Equations 1-4 might have already generated a substantial amount error that remains not completely untouched in the remainder of the manuscript. Also, why did the authors not choose a physical dam break model such as BASEMENT[20]?
- L180: What is the mechanism that causes overtopping? A splash wave? Or overflow by a (gradually) growing lake volume?
- L189: 'Uniformity of land cover and lack of vegetation': strong statements - please show this. Source for the chosen value of Manning's n?

- L201-2014: I found it somehow confusing to see first the paragraph on flood modelling, followed by a paragraph that describes potential causes for this flood. Again, it may sound a bit harsh, but considering these triggers decoupled from the flood routing models seems to represent not the state of the art in GLOF modelling. Determining the likelihood (again without a probability) for a rock or ice avalanche from a fixed angle of reach might be suitable for a large-scale assessment of GLOF susceptibility that pursue a rapid screening for sites that need more attention. Here, the authors have identified such sites, so there is (from my perspective) no need to rely still on the more traditional side of GLOF susceptibility appraisals. In other words, if there is a potential for rock to detach, what could be its size? How rapidly may this rock avalanche enter a lake? What would the displacement wave look like? It also remains unclear why the authors had chosen exactly those susceptibility factors in Table 3 (or 2). Suggest to add references to this table that showed if these susceptibility factors in deed had positive (+) relevance for triggering a GLOF.
- L205: There seems to be a referencing issue: The item Table 3 must be Table 2.
- L222: 'cannot': 'may not'?
- L217-231: It's good to know about historic changes in glacier velocity and elevation, but these insights seem to be decoupled from the future lake development. Please try to make this link clearer or consider deleting.
- L241: Please elaborate more on what you mean with 'replicate'.
- L237-243: Do these scenarios also consider accelerated lake growth and glacier retreat, once the lake has formed?
- L246: 'probability': again, there is no probability involved in this study. 'exposure of buildings' has not been mentioned in the Methods. 'A full hazard … assessment … is beyond the scope of this study': why then calling this paper then a hazard assessment in the title?
- L250: Why did the simulations stop the border? Isn't it one of the goals of this paper to show the transboundary hazard from GLOFs?
- L266: Why are the flow depths of 25m more realistic? Calls for some sensitivity analysis.
- L274-276: Again, very simplified assumption on sediment entrainment and aggradation.
- L285: 'significant': how much?
- L287: 'high-impact low-probability': How high is this impact and how low is this probability?
- L288: 'requires more sophisticated modeling': unclear why this has not been considered in this specific case study that offers the data to do this sort of modeling.
- L301: 'likelihood or probability': replace with 'possibility'?
- L310: 'low': how low?
- L311: 'can be effectively discounted': why? And how efficient?
- L316: 'large slope instabilities': In-SAR might help to detect those?
- L326-347: Most of this section could be redundant or offer much more potential for discussion, if the authors considered a suite of lake impacts in this study.
- L343: 'making this a high magnitude, but very low likelihood process chain': how high is this magnitude and how low is the likelihood?
- L344-345: 'active instabilities are clearly evident': so, why not show these?
- L349-355: Similar to the comments above, it remains unclear, how the authors define the likelihood (or magnitude) of a GLOF.
- L360: 'Con.', 'Trig.', 'Mag.': please write out fully. 'Catchment drainage density': why is this important. 'TP = 3280': Unclear what this means, please explain. 'Steep moraine slopes >500 m high': really? A 500 m high moraine dam?
- L372: How complete is OSM in this region?

- L387: 'While we did not simulate beyond the border owing to the limited coverage … ': needs to come earlier.
- L435: 'applied these approaches for the first time': really?
- L436: What is 'complex' in this topography?
- L444-447, and again L455ff: The artificial drainage of Jialongco somehow undermines parts of this paper, given that the simulations use the lake volume before the drainage, right? It would have been good to show how simulated flood magnitudes (flow depths, extent of the inundation, etc.) change because of the reduced flood volume, and if this in turn changes GLOF risk. In essence, if the probability of a given flood magnitude decreases ( = hazard), then risk must also decrease, assuming constant values of exposure and vulnerability. It thus remains unclear why this insight has been held out from the introduction and all subsequent analysis.
- L458: 'had only a minimal effect on the overall lake size': Unclear conclusion with regards to Fig. 9.
- L463: 'would reduce the threat to these buildings': contradicts to what is written some sentences above?
- L470-483: largely repetitive from the Results, consider substantial shortening or deleting.
- L495-499: Content of these sentences unclear, even if formulated 'in other words'. Please rephrase or shorten.
- L501: When talking about early warning, why not using the calculated flood arrival times to provide a solid basis for discussion? What do the authors suggest to implement in such an early warning system? How can people response to the warning?
- L503-504: 'under the philosophy of preparing for the worst, while hoping for the best': please avoid jargon.
- L504: 'complex transboundary regions': what is complex here and how can this study help to understand this complexity?
- L523-525: Not sure whether the increase in GLOF risk has been quantified in this study?
- L528-529: 'Hard engineering strategies that address only the hazard source are a socially and environmentally less desirable option': Not sure whether I can agree with this statement: If Hazard = 0 (because there is no lake), then Risk = 0. Consider revising or deleting this statement.

**References**

1. Mergili, M. *et al.* How well can we simulate complex hydro-geomorphic process chains? The 2012 multi-lake outburst flood in the Santa Cruz Valley (Cordillera Blanca, Perú): How well can we simulate complex hydro-geomorphic process chains? *Earth Surf. Process. Landf.* **43**, 1373–1389 (2018).
2. Mergili, M., Frank, B., Fischer, J.-T., Huggel, C. & Pudasaini, S. P. Computational experiments on the 1962 and 1970 landslide events at Huascarán (Peru) with r.avaflow: Lessons learned for predictive mass flow simulations. *Geomorphology* **322**, 15–28 (2018).
3. Zheng, G. *et al.* The 2020 glacial lake outburst flood at Jinwuco, Tibet: causes, impacts, and implications for hazard and risk assessment. *The Cryosphere* **15**, 3159–3180 (2021).
4. Worni, R. Analysis and dynamic modeling of a moraine failure and glacier lake outburst flood at Ventisquero Negro, Patagonian Andes (Argentina). *J. Hydrol.* 12 (2012).
5. Sattar, A., Goswami, A., Kulkarni, Anil. V. & Emmer, A. Lake Evolution, Hydrodynamic Outburst Flood Modeling and Sensitivity Analysis in the Central Himalaya: A Case Study. *Water* **12**, 237 (2020).

6. Allen, S. K., Rastner, P., Arora, M., Huggel, C. & Stoffel, M. Lake outburst and debris flow disaster at Kedarnath, June 2013: hydrometeorological triggering and topographic predisposition. *Landslides* **13**, 1479–1491 (2016).

7. Frey, H., Haeberli, W., Linsbauer, A., Huggel, C. & Paul, F. A multi-level strategy for anticipating future glacier lake formation and associated hazard potentials. *Nat. Hazards Earth Syst. Sci.* **10**, 339–352 (2010).

8. Sattar, A., Goswami, A. & Kulkarni, A. V. Hydrodynamic moraine-breach modeling and outburst flood routing-A hazard assessment of the South Lhonak lake, Sikkim. *Sci. Total Environ.* **668**, 362–378 (2019).

9. Sattar, A. *et al.* Future Glacial Lake Outburst Flood (GLOF) hazard of the South Lhonak Lake, Sikkim Himalaya. *Geomorphology* **388**, 107783 (2021).

10. Nie, Y., Liu, W., Liu, Q., Hu, X. & Westoby, M. J. Reconstructing the Chongbaxia Tsho glacial lake outburst flood in the Eastern Himalaya: Evolution, process and impacts. *Geomorphology* **370**, 107393 (2020).

11. Schneider, D., Huggel, C., Cochachin, A., Guillén, S. & García, J. Mapping hazards from glacier lake outburst floods based on modelling of process cascades at Lake 513, Carhuaz, Peru. *Adv. Geosci.* **35**, 145–155 (2014).

12. Lala, J. M., Rounce, D. R. & McKinney, D. C. Modeling the glacial lake outburst flood process chain in the Nepal Himalaya: reassessing Imja Tsho's hazard. *Hydrol. Earth Syst. Sci.* **22**, 3721–3737 (2018).

13. Rounce, D. R., McKinney, D. C., Lala, J. M., Byers, A. C. & Watson, C. S. A new remote hazard and risk assessment framework for glacial lakes in theNepal Himalaya. *Hydrol. Earth Syst. Sci.* **20**, 3455–3475 (2016).

14. Wang, X. *et al.* Glacial lake inventory of high-mountain Asia in 1990 and 2018 derived from Landsat images. *Earth Syst. Sci. Data* **12**, 2169–2182 (2020).

15. Chen, F. *et al.* Annual 30 m dataset for glacial lakes in High Mountain Asia from 2008 to 2017. 26 (2021).

16. Haritashya, U. *et al.* Evolution and Controls of Large Glacial Lakes in the Nepal Himalaya. *Remote Sens.* **10**, 798 (2018).

17. Drenkhan, F., Huggel, C., Guardamino, L. & Haeberli, W. Managing risks and future options from new lakes in the deglaciating Andes of Peru: The example of the Vilcanota-Urubamba basin. *Sci. Total Environ.* **665**, 465–483 (2019).

18. Nussbaumer, S., Schaub, Y., Huggel, C. & Walz, A. Risk estimation for future glacier lake outburst floods based on local land-use changes. *Nat. Hazards Earth Syst. Sci.* **14**, 1611–1624 (2014).

19. Zheng, G. *et al.* Increasing risk of glacial lake outburst floods from future Third Pole deglaciation. *Nat. Clim. Change* **11**, 411–417 (2021).

20. Worni, R., Huggel, C. & Stoffel, M. Glacial lakes in the Indian Himalayas — From an area-wide glacial lake inventory to on-site and modeling based risk assessment of critical glacial lakes. *Sci. Total Environ.* **468–469**, S71–S84 (2013).

---

## Referee Comment (RC3)

In their submitted manuscript, Allen et al. propose a hazard assessment around two existing and one potentially forming proglacial lake in the Puiqu River basin, Himalaya. They document the glacial environment as well as its projected development under future climate conditions, potential outburst flood triggers and modeled discharges in vulnerable downstream communities. The study gives insights into different aspects of glacial lake outburst floods highlighting particularly interesting features of the investigated cases. However, it is difficult to grasp how relevant the scientific insights from this investigation are given that the authors' judgement is too often limited to qualitative assessment. For a scientific journal submission, I was expecting more substance in view of reproducibility and representativeness of the results (see major comments below).

Besides my main criticism, this manuscript is well written and easy to follow. The figures will benefit from annotations and some other modifications.

Fabian Walter.

MAJOR COMMENTS

My main point of criticism is that the reader of this manuscript is left with little information on how to assess validity or accuracy of the findings. In its current state, the study appears more like a presentation of important facts and qualitative judgements, which are typical for technical reports. For a scientific paper I would have expected some critical assessment of the flood risk, e.g., a benchmarking of the presented methods against previous occurrences of outburst floods. The authors cite accounts of previous outbursts in the area (Line 106). Could they be used for this?

The first part of the paper presents some motivation on why to study the chosen three lake basins. However, it is not possible to verify that this corresponds to a worst-case analysis. In this case, it would be necessary to show that no lake basins could produce more serious outburst floods. As the authors argue, this depends on moraine dam geometry, water volume and trigger potential. Under these aspects it cannot be argued that the presented set of lakes is representative for worst case scenarios.

At too many parts of the manuscript, the authors' qualitative judgement is presented as a scientific result. In particular, in the Section "GLOF likelihood", various factors influencing or triggering outburst floods are presented, but I could hardly find any objective arguments. It seems that the only one is the estimate of a dam-overtopping wave volume, which can be 10 times as high as the "incoming mass". Here and elsewhere in the manuscript, it has to be made clear that the conclusions are based on solid scientific grounds. Otherwise, a "low probability" could indicate one catastrophic event every 5 years as opposed to several ones per year. This is not what the authors imply. In a similar sense, it is not clear what the demanded "comprehensive and forward-looking approach to disaster risk reduction" is. To me, such an approach should always be taken and I see little connection to the present study or any finding, which made the suggested strategy particularly pertinent to the Poiqu River basin compared to any other place in the world.

SPECIFIC COMMENTS

I suggest including a cartoon explaining different lake formation scenarios and specifically the hydrological base line. To be honest, I had to stare for some time at Figure 3 of Benn et al. (2012) to understand this concept. On the other hand, I never grasped the meaning of the "the lowest point where the glacier surface intersects the terminal moraine" (it seems that by definition, the glacier and the terminal moraine should not intersect). Similarly, when the future evolution of the lakes is described, longitudinal profiles would be extremely helpful. This would help the reader to understand Figures 7 and 8 and appreciate the shown information.

The flood model is a key ingredient to this investigation. Although I agree that too many technical and mathematical details are not appropriate for this study, I was wondering what the main parameters and boundary conditions of this model are. Apparently, the flood volume, some time scale of drainage initiation and dam geometry play a role and it would be interesting to hear how these parameters drive the model.

Lines 45-47: "… most scientific attention has focused upon …" I do not agree with this statement. In the jökulhlaup literature, ice-marginal and subglacial lake drainages have also received a lot of attention. Whereas I cannot say which scenario has been most prominent, I would refrain from an absolute statement on scientific attention.

Lines 175-176: "$B_w$ and $h_b$ are fully obtained" measured?

Line 178: Reference for HEC-RAS is needed.

Line 185: Reference for DEM's is needed.

Line 189: Reference for Manning roughness value is needed.

3.2 Lake susceptibility assessment: Presenting the likelihood calculations seems appropriate here.

Line 231: "considering the factors outlined by …" these factors should be specified.

Line 257: I suggest a paragraph break here.

Line 457: "recent removal of much of the frontal moraine …" this needs a reference.

Lines 466: What are "capacity building programs"?

FIGURES

Figure 1: The lakes at Cirenmaco and Jialongco are difficult to discern.

Figure 2: The font sizes are a bit small, but $h_b$ defined in Panel A seems to disagree with Panel B.

Figure 4A: Where is the lake? The blue outline or the light blue polygon? A different color scale for maximum flow height would help.

Figure 5: It is difficult to tell where the lakes are. I do not see any blue patches.

Figure 6: The two images in Panels C need some annotation. What does the reader see in these images? Why is one so dark and the other one bright? What happened between the two?

Figure 8: This is the future lake site, right? Where will the lake form? The colored outlines make little sense and are hard to sea. Which direction does the ice flow?

Figure 9: Some arrows and annotation as well as scale bars are needed.

---

## Author Comment (AC1)

**Reviewer 1**

Major Comments:

Allen et al. used a suite of well-established tools to assess possible trigger mechanisms of GLOFs, and to model hydro-mechanical characteristics and runouts of the flood waves. Unfortunately, the authors missed the opportunity to couple such triggers with the subsequent failure of moraine dams. Recent software such as r.avaflow have recently improved the understanding of GLOF processes, allowing for pseudo-probabilistic assessments of GLOF triggers and impacts under a range of adjustable and testable boundary conditions. However, given that the authors considered only three worst-case scenarios of moraine-dam failure, largely decoupled from the initial triggers, I see only limited progress in calculating the likelihood of observing a GLOF of a given magnitude, though this is repeatedly stated in the manuscript. Clearly, the authors present a thorough expert-based assessment of potential conditioning or triggering factors. Yet it remains unclear how this traditional, rather subjective procedure brings GLOF hazard and risk assessment forward with regards to the many previous case studies that we saw in the past years. Thus, the authors' goal to estimate "the core hazard dimensions of GLOF magnitude and likelihood (or probability)" (L245-246) only remain partly fulfilled in my opinion. If this manuscript is intended to improve early warning and risk management, then the reader (at best a person involved in such tasks) might expect a more objective use of the terms hazard or probability, that is a numeric value between zero and one. A probability or return period might provide a robust baseline for decision makers compared to the current distinction between a 'high' or 'low' level of susceptibility or impact. Fortunately, most of the data (including high-resolution DEMs and satellite images) for such an extended appraisal already come with this manuscript. I thus would like to encourage the authors to revise their manuscript accordingly, considering a much wider range of scenarios in order to proceed towards a more objective assessment of current and future GLOF hazard in their study region.

We thank the reviewer for this critical feedback on the manuscript, and apologise for the time taken to respond. The comment from the reviewer gets right to the fundamental contribution our study is trying to make, and unfortunately this has perhaps not been adequately introduced or fulfilled. We are aware of the value of r.avaflow and other approaches for undertaking detailed, scenario-based hazard assessment, and are using exactly these models in other studies, where comprehensive hazard mapping is the desired outcome. However, what we are aiming to achieve in this particular paper is to address a gap in approaches/scales, fitting between large scale first-order approaches and scenario based hazard mapping, to provide an illustrated case-study showing how potential new lakes can begin to be considered in DRR planning. Is a given lake really going to present an unprecedented threat to downstream areas (relative to existing lakes)? Does such a potential lake need to be considered in the design of response strategies such as EWS? Such questions are not answered by existing studies, which to our knowledge, have not gone beyond first-order assessment of future lakes (e.g. GIS-based approaches).

On a practical level, we see this gap as an important niche to fill. There has now been several years' worth of first-order modelling studies showing that glacial lake area and number will increase in the future, yet in our interactions with authorities across several countries, we see no indication yet that these future threats are being considered in DRR planning. While it would be academically possible to generate detailed scenario-based hazard maps for a potential future lake, we consider a more practical intermediate step is needed to first demonstrate that a given future lake warrants such attention and further monitoring.

We agree that ultimately, DRR planning can be best informed by more quantitative assessment of outburst probabilities. However, this remains the holy grail for the assessment of existing lakes, let alone future lakes for which conditions of both the lake and surroundings are highly uncertain. Particularly for DRR strategies such as an EWS, we would therefore argue that there is merit in simulating single worst-case scenarios, as these can be a first basis for planning response strategies that remain robust under an uncertain future. For example, by planning community awareness programmes around a worst-case of 70 minutes vs. 130 minute warning time, or ensuring critical infrastructure is located well away from a future GLOF path.

The decision to use HEC-RAS for modelling downstream flood impacts was initially driven by a desire to keep the approach as simple as possible, with a view that methods could be attractive for upscaling to any number of potential future lakes. However, in view of the comments from all 3 reviewers, we accept that this was scientifically weak, as the triggering processes were not linked to the outburst scenario, leading to reliance on assumptions (e.g., concerning PFV) and empirical relationships. Therefore, in the revised manuscript we will use r.avaflow to simulate worst-case ice-rock avalanche triggered GLOFs process chains from all 3 lakes, including a new scenario for the artificially lowered lake volume of Jialongco. The selection of worst-case avalanche volumes will be comprehensively described, and based on a more nuanced assessment of triggering factors given in Table 3, particularly relating to glacial and geological conditions.

In conclusion, we would summarise the novelty and contribution of the revised manuscript as being **a first study to demonstrate the effect of future lake development on downstream flood magnitudes, with implications for DRR planning.** In addition, we will now be able to **demonstrate the effect of lake lowering on downstream flood magnitudes**.

Given the valid concerns of the reviewer, we also propose to modify the title to more accurately capture the scope and rationale of the paper:

*"Glacial lake outburst flood magnitudes under current and future conditions: implications for disaster risk reduction in a transboundary Himalayan basin"*

**Detailed comments:**

-L6: 'far reaching': in terms of run-out? Social consequences? Media coverage?
Revised to *"where societal impacts can occur far downstream"*

-L10: 'well-known dangerous': please consider avoiding subjective terms.
Revised to *"previously identified dangerous lakes"*. This is in line with the numerous studies cited in the text that assess these lakes as being dangerous.

- L26-27: 'Based on … capacity building programs': Hardly mentioned in the text, consider revising or deleting.
Revised to *"education and awareness raising programs"*. We feel it's an important point to make, even if not a central component of this study, because authorities (and funders) often overlook such important social dimensions of DRR.

- L39: 'rapidly in both size and number': how much? Consider using recently multi-temporal glacier lake inventories[14,15].
The more recent studies have been cited, and we report, by way of example, the increase of 27 km$^2$ in glacial lake area for the Central Himalaya reported over the past decade (Chen et al. 2021).

- L40: What is the frequency of GLOFs in the Himalaya? If I am not mistaken, most researchers deem GLOFs a low rate of occurrence.
We now report the GLOF frequency of 1.3 per year for the Himalaya since the 1980s (after Vey et al 2019).

- L47f: 'Such lakes can have volumes >100 m³', according to ref. [16]
Sentence revised to *"volumes exceeding 100m$^3$"* and the new reference added.

- L69: 'within the eastern Himalayan region': why there?
Have removed this part of the sentence – it was stating the obvious as the largest border area between Nepal and China is in the eastern Himalayan region.

- L75: 'are lacking': suggest to tone this a bit down, as it undermines previous appraisals

We don't intend to undermine the previous studies which have considered future lakes – many of which we were involved in or led – but these are studies aiming to identify district or regional-scale changes in GLOF risk. Other than a general indication of how hazard or risk could change, they don't provide information for specific lakes, that is required for DRR planning.

In view of the concern of the reviewer, wording of this sentence will be revised to *"yet practical examples on how to account for future lake development in local GLOF risk assessment and risk management have been rarely demonstrated"*.

- L79-82: Is this statement valid for all glaciated high mountains? Or only the Himalayas?
To our knowledge the statement is valid globally. We know of no cases where local authorities have begun to consider future lakes in the their hazard and risk assessment. Hence, we leave as a general, globally applicable statement.

- L89-91: The last sentence could be deleted, if shortening is needed. The introduction ends well with a strong presentation of the research goals.
Agree, sentence removed.

- L93: 'considering potential GLOF impacts'
Agree, wording revised

- L97: 'mean annual air temperature'
Wording revised

- L99: are these precipitation rates (I think these are monthly?) the maximum or the average recorded in a given period?
Thanks for pointing this out. There are average monthly rainfall totals (not rates). Text will be corrected.

- L105: 'GLOF danger': replace with 'GLOF hazard'?
Disagree – "danger" was the term deliberately used in Allen et al. 2019 as the study went beyond hazard to consider also exposure of infrastructure (but fell short of a full risk study).

- L107: 'losses of up to US$4 million': is this in today's currency?
These were values of 2015 – we will clarify this in the text.

- L111-112: 'a large and persistent threat, considered as one of the most dangerous lakes in Tibet': suggest to tone this down again. Given that previous appraisals used different criteria and thresholds, we may wish to avoid confusion what is now the 'most dangerous lake' in a given region.
The point we want to make here is that irrespective of different methods and criteria used, multiple studies point to his lake as being one of the most dangerous/hazardous in the region. We will add a couple of further papers to support this. Also we've removed the term "large" as this is subjective. Text revised to *"identified by multiple studies as being one of the most critical lakes in Tibet"*

- Figure 1: Please highlight the border between Nepal and China.
Border will be added.

- L139: So estimating lake depth comes with zero uncertainty?
Over the past 2 field seasons we have been able to measure bathymetry of both Jialongco and Galongco lakes. We will now compare the measured volumes to the estimated volumes from several empirical relationships to come up with a range of uncertainty. For GLOF modeling, the measured volumes will now be used for Jialongco and Galongco, while for the future lake, upper and lower volumes will be simulated, based on the uncertainty range established above.

- L150: To my knowledge GlabTop models the glacier thickness in a bedrock topography. Did the authors add the moraine on top of this bedrock depression? How high / wide is this moraine? If not, how susceptible is a bedrock depression to overtopping? Did the authors consider sediment input from supraglacial debris into the lake?

Reviewer is correct in their interpretation. In keeping with a philosophy of a worst-case simulation from the future lake, we do not consider additional height (and freeboard) from a moraine dam, the height of which can only be guessed. We take the bedrock topography as simulated from Glabtop. Likewise sediment deposition into the lake, thereby reducing the potential lake volume is not considered. These aspects/limitations will be noted in the methodology.

- L155: Please consistently use Fig. / Fig / Figure (or any other option) throughout the manuscript. Sometimes the abbreviation is written with a dot, sometimes not.

Corrected.

- L154-164: the PFV approach may underestimate the amount of water that is generated from mass flows entering the lake and causing a splash wave. Given that the authors consider ice avalanches and landslides entering the lakes, how useful is it to use the PFV approach?

The revised modelling approach using r.avaflow will generate the flood volume based on the actual simulated avalanche scenario into the lake.

- L156: The PFV concept further seems to confuse mean and maximum depth. Why should the drain empty completely if the depth of the breach is equal to the mean depth of the lake? Again, the assumption of a fixed breach depth might simplify (or even underestimate) the worst-case scenario of GLOF volumes.

This was a typo, and should have referred to maximum depth. In any case, the PFV approach will be replaced, and flood volumes will be dynamically generated with r.avaflow.

- L162-163: 'In comparison, … (Xu, 1988)': what is this comparison good for?

The comparison was provided to indicate that the estimated breach depth of 40 m is not completely unrealistic. This paragraph will be revised now based on the new breach depth simulated using r.avaflow.

- L165: 'the DEM accuracy is unknown': what is the accuracy of the 1-m DEM then?

No other high-resolution DEMs of Poiqu are publicly available to assess the vertical and horizontal accuracy in the Pleiades DEM. However, Berthier et al. (2014) computed vertical accuracy of Pleiades DEMs over the Agua Negra study site and reported mean vertical biases ranging from 0.99 to 1.33 m without GCPs. Similar accuracy level (0.3 m) was also reported by Zhou et al. (2015) from the comparison of a Pleiades-1 DEM with an airborne LiDAR DEM. Similarly, without ground control points (GCPs), the horizontal location accuracy of the images was estimated as 8.5 m (CE90, Circular Error at a confidence level of 90 %) for Pléiades-1A and 4.5 m for Pléiades-1B (Lebègue et al., 2013; Oh and Lee, 2014).

References
Berthier, E., Vincent, C., Magnusson, E. et al. (2014). Glacier topography and elevation changes derived from Pleiades sub-meter stereo images. The Cryosphere, 8, 2275-2291.
Zhou, Y., Parsons, B., Elliott, J.R., Barisin, I and Walker, R.T. (2015), Assessing the ability of Pleiades stereo imagery to determine height changes in earthquakes: A case study for the El Mayor-Cucapah epicentral area. Journal of Geophysical Research- Solid Earth, 120, 8793–8808.
Oh, J. and Lee. C. (2014), Automated bias-compensation of rational polynomial coefficients of high-resolution satellite imagery based on topographic maps. ISPRS Journal of Photogrammetry and Remote Sens., 100, 12–22.
Lebègue, L., Greslou, D., Blanchet, G., De Lussy, F., Fourest, S., Martin, V., Latry, C., Kubik, P., Delvit, J.-M., Dechoz, C., and Amberg, V. (2013). PLEIADES satellites image quality commissioning, Proc. SPIE 8866, Earth Observing Systems XVIII, 88660Z (23 September 2013), doi:10.1117/12.2023288, 2013.

- L171-172: Again no uncertainties in these equations? Propagating estimates through Equations 1-4 might have already generated a substantial amount error that remains not completely untouched in the remainder of the manuscript. Also, why did the authors not choose a physical dam break model such as BASEMENT?

The Froehlich equations will not be used in the revised manuscript. The simulation of the entire process chain will be undertaken using r.avaflow. Testing of different parameters in the simulations, and related uncertainties, will be reported in supplementary material.

- L180: What is the mechanism that causes overtopping? A splash wave? Or overflow by a (gradually) growing lake volume?

In the revised manuscript, the overtopping mechanism will be simulated using r.avaflow. Given that ice-rock avalanching is identified as the most feasible GLOF triggering mechanism for all 3 lakes, we expect a splash wave to be the primary mechanism. Gradual collapse of a lateral moraine wall could feasibly lead to gradual overflow in the case of Jialongco.

- L189: 'Uniformity of land cover and lack of vegetation': strong statements - please show this. Source for the chosen value of Manning's n?

Manning's n will not be required for the updated modelling with r.avaflow. Key model parameters (including the internal and basal friction angles of the solid material, the fluid friction number, and the coefficient of erosion) will be empirically defined, with references added.

- L201-2014: I found it somehow confusing to see first the paragraph on flood modelling, followed by a paragraph that describes potential causes for this flood. Again, it may sound a bit harsh, but considering these triggers decoupled from the flood routing models seems to represent not the state of the art in GLOF modelling. Determining the likelihood (again without a probability) for a rock or ice avalanche from a fixed angle of reach might be suitable for a large-scale assessment of GLOF susceptibility that pursue a rapid screening for sites that need more attention. Here, the authors have identified such sites, so there is (from my perspective) no need to rely still on the more traditional side of GLOF susceptibility appraisals. In other words, if there is a potential for rock to detach, what could be its size? How rapidly may this rock avalanche enter a lake? What would the displacement wave look like? It also remains unclear why the authors had chosen exactly those susceptibility factors in Table 3 (or 2). Suggest to add references to this table that showed if these susceptibility factors in deed had positive (+) relevance for triggering a GLOF.

Thanks for these comments. We agree, the current structure is not logical or in keeping with best practice. In the revised manuscript, the methodological description on lake susceptibility will come before the section on GLOF modelling (and likewise in the results). The link between triggering processes and outburst process will be described, and used as justification for the choice of modelling approach (r.avaflow). Use of topographic potential (angle of reach) for the rock/ice avalanche likelihood will be removed. This will be replaced with a more comprehensive assessment of local geological and glacial conditions to estimate possible upper limits to an initial avalanche volume. Characteristics of the avalanche flow and its interaction with the lake will then be simulated with r.avaflow. The susceptibility factors in Table 3 are taken from the GAPHAZ international guidance document. As suggested, we will add references (from the region) to support the relevance of these factors in this particular study.

- L205: There seems to be a referencing issue: The item Table 3 must be Table 2.
Thanks – will be corrected.

- L222: 'cannot': 'may not'?
Will be revised to "may not"

- L217-231: It's good to know about historic changes in glacier velocity and elevation, but these insights seem to be decoupled from the future lake development. Please try to make this link clearer or consider deleting.

As suggested above, we will improve on the text in section 3.3 to ensure that the relation between glacier dynamics and geometry, and their impact on future lake development, are more clearly set out in this

section. Contrary to the reviewers statement here, historic changes in glacier surface velocity and elevation are key to supraglacial meltwater ponding (e.g. Quincey et al., 2007; King et al., 2018) and initial lake development, so we will ensure this is clear to the reader.

- L241: Please elaborate more on what you mean with 'replicate'.
Will be revised to *"seen again"*

- L237-243: Do these scenarios also consider accelerated lake growth and glacier retreat, once the lake has formed?
They do not, which we are careful to acknowledge in the discussion of the results on lines L484-489. We will add to this text that lake expansion rates will likely be much higher when processes such as ice front calving begin to occur in response to lake expansion. Unfortunately, inclusion of ablative processes such as calving are beyond the scope of the approaches we employ in this work.

- L246: 'probability': again, there is no probability involved in this study. 'exposure of buildings' has not been mentioned in the Methods. 'A full hazard … assessment … is beyond the scope of this study': why then calling this paper then a hazard assessment in the title?
We accept the reviewers point. The opening sentence will be revised to *"We focus below on results relating to susceptibility of the 3 lakes, and the potential magnitude of downstream impacts, as simulated under worst-case outburst scenarios".*
Furthermore, the results section currently titled "GLOF likelihood" will be revised to "lake susceptibility" (consistent with the methodology section). This section will come first, before the section on downstream impacts.
Please note also the suggested revision to the paper title: *"Glacial lake outburst flood magnitudes under current and future conditions: implications for disaster risk reduction in a transboundary Himalayan basin".*
The approach and data used to assess exposure of buildings, will be noted in the methods.

- L250: Why did the simulations stop the border? Isn't it one of the goals of this paper to show the transboundary hazard from GLOFs?
We chose to only simulate until the border as that was the extent of the high resolution DEM we created, and further extension would have been at significant financial cost. This is noted later in the manuscript, but will be moved earlier. By simulating the arrival time and magnitude of the flood at the border, we believe this provides the intended basis for discussing transboundary hazard. Of course, detailed EWS planning will require modelling further downstream.

- L266: Why are the flow depths of 25m more realistic? Calls for some sensitivity analysis.
The problems with the lower resolution DEMs were clear and could be seen with simple profiles taken along the stream (stream path flowing uphill for example with obvious steps/blockages in the topography). So it was very clear that flow pooling and huge depths was artificial and not realistic. However, we agree that some sensitivity analyses around the parameters used in the modelling need to be included, and will do so in supplementary material.

- L274-276: Again, very simplified assumption on sediment entrainment and aggradation.
Using r.avaflow for the revised modelling will now allow us to make more quantitative statements around sediment entrainment and possible flow transformation.

- L285: 'significant': how much?
In the revised manuscript we will estimate this overtopping volume into Jialongco (assuming this overtopping still occurs in the new model results with r.avaflow).

- L287: 'high-impact low-probability': How high is this impact and how low is this probability?
As we are dealing with worst-case scenarios, we are assuming low (or very low) probabilities. As the reviewer no doubt knows, it's extremely difficult in the field of glacial hazards to assign quantitative

probabilities, unless dealing with reoccurring events (ice dammed lakes for example). However, we understand it would be important in our study to determine if such overflow into Jialongco might be possible with lower volume triggering events. We will therefore perform some sensitivity analyses on this, using initial volumes 75% and 50% less than the worst-case scenario to determine if runnup and overflow into Jialongco still occurs. The multiple simulations will not be continued downstream as this goes beyond the scope of the study. Results will be included in supplementary material.

- L288: 'requires more sophisticated modeling': unclear why this has not been considered in this specific case study that offers the data to do this sort of modeling.
In the revised manuscript we will attempt to simulate this chain reaction event using r.avaflow. This could be quite an innovative addition, given we are unaware of a model being used to simulate such a domino effect from one lake into the next.

- L301: 'likelihood or probability': replace with 'possibility'?
Here and throughout, we will speak only of likelihood, which can be qualitative (eg. high, med, low).

- L310: 'low': how low?
We don't believe given the scope of this study it is feasible to go beyond a qualitative probability level.

- L311: 'can be effectively discounted': why? And how efficient?
As above, we find it unfeasible to be more quantitative with such statements. As pointed out be other reviewers, the huge width of the dam make catastrophic erosion of the dam very unlikely. How the dam behaves in response to the avalanche triggered overflow in the r.avaflow simulations will give us some further basis for this statement.

- L316: 'large slope instabilities': In-SAR might help to detect those?
Noted, but outside the scope of this study. We will add to the discussion a point on the potential for In-SAR to be included in a monitoring/EWS strategy.

- L326-347: Most of this section could be redundant or offer much more potential for discussion, if the authors considered a suite of lake impacts in this study.
As noted in the response to the general comment, we prefer to keep the scope of this study on single worst-case scenarios, and believe there is justification for this. Using r.avaflow will now allow us to determine the outburst volume and downstream flood magnitudes linked to an assessed worst-case avalanche volume, and the section will be rewritten accordingly. We understand the challenges involved in estimating a worst-case avalanche volume, and will carefully discuss this aspect.

- L343: 'making this a high magnitude, but very low likelihood process chain': how high is this magnitude and how low is the likelihood?
Section will be rewritten based on new simulations, which will directly link the worst-case avalanche volume with the outburst volume. The estimation of likelihoods will remain an evidence-based qualitative estimate.

- L344-345: 'active instabilities are clearly evident': so, why not show these?
We will add a photo from the field showing this.

- L349-355: Similar to the comments above, it remains unclear, how the authors define the likelihood (or magnitude) of a GLOF.
Section will be rewritten based on new simulations, which will directly link the worst-case avalanche volume with the outburst volume. The estimation of likelihoods will remain an evidence-based qualitative estimate.

- L360: 'Con.', 'Trig.', 'Mag.': please write out fully. 'Catchment drainage density': why is this important. 'TP = 3280': Unclear what this means, please explain. 'Steep moraine slopes >500 m high': really? A 500 m high moraine dam?

Further explanation will be provided in the table, in line with the reviewer comments. The moraine slope >500m refers to the lateral moraine walls.

- L372: How complete is OSM in this region?

As noted in the text, both OSM and latest google earth imagery were used to ensure all exposed buildings were identified.

- L387: 'While we did not simulate beyond the border owing to the limited coverage … ': needs to come earlier.

Noted, and will be moved earlier.

- L435: 'applied these approaches for the first time': really?

To our knowledge, yes, this is the first time a study has undertaken any sort of GLOF modelling for an overdeepening/future lake to give downstream flow heights, velocities etc. Previous studies have identified overdeepenings and downstream infrastructure that are within a GIS routed flow trajectory. However, to avoid a discussion around what is or is not considered GLOF modelling, we will remove the term "first time".

- L436: What is 'complex' in this topography?

Will be revised to *"steep, mountain topography"*

- L444-447, and again L455ff: The artificial drainage of Jialongco somehow undermines parts of this paper, given that the simulations use the lake volume before the drainage, right? It would have been good to show how simulated flood magnitudes (flow depths, extent of the inundation, etc.) change because of the reduced flood volume, and if this in turn changes GLOF risk. In essence, if the probability of a given flood magnitude decreases ( = hazard), then risk must also decrease, assuming constant values of exposure and vulnerability. It thus remains unclear why this insight has been held out from the introduction and all subsequent analysis.

We agree, and learned only of the artificial drainage late in the process of working on this paper (highlighting the disconnect between local authorities and scientists in this region). For the revised manuscript, we will now include a new scenario with the reduced volume.

- L458: 'had only a minimal effect on the overall lake size': Unclear conclusion with regards to Fig. 9.

Now that we have measured bathymetry for this lake (pre volume reduction) we'll include an estimation of how much the volume has been reduced.

- L463: 'would reduce the threat to these buildings': contradicts to what is written some sentences above?

The sentence will be revised according to the results from the new scenario with the reduced lake volume.

- L470-483: largely repetitive from the Results, consider substantial shortening or deleting.

Repetition to be removed.

- L495-499: Content of these sentences unclear, even if formulated 'in other words'. Please rephrase or shorten.

The sentence will be revised for clarity and in line with the revised modelling results.

- L501: When talking about early warning, why not using the calculated flood arrival times to provide a solid basis for discussion? What do the authors suggest to implement in such an early warning system? How can people response to the warning?

This part of the discussion will be expanded as suggested, based on the flood arrival times modelled with r.avaflow. We'll also refer to the UNDP guidelines on the essential components of an EWS, including how people can respond, highlighting that a functional EWS goes far beyond the technical components.

- L503-504: 'under the philosophy of preparing for the worst, while hoping for the best': please avoid jargon.

Agree, rather more language for use when presenting to authorities and decision-makers and will be removed here.

- L504: 'complex transboundary regions': what is complex here and how can this study help to understand this complexity?

Wording of "complex" will be removed. The sentence is intended to simply highlight why warning times are so critical in a transboundary region, because we know from past experiences that communication between national authorities can lead to delays in alerting communities.

- L523-525: Not sure whether the increase in GLOF risk has been quantified in this study?

Fair point. We will revise to *"increased GLOF exposure"* which is justified based on Figure 6.

- L528-529: 'Hard engineering strategies that address only the hazard source are a socially and environmentally less desirable option': Not sure whether I can agree with this statement: If Hazard = 0

We want to say here that hazard will never = 0 in such a context, unless you drain every lake completely, and do so for every new lake that develops. In addition, even if you arguably could reduce GLOF hazard to zero, the community remains vulnerable and exposed to other geohazards. Hence, a more comprehensive approach, involving also EWS and related social interventions, are far more desirable in almost all contexts in our view. This does not mean that remedial measures do not have a place within such a comprehensive approach. In view of the reviewers concern, we will revise the wording to: *"Hard engineering strategies, in isolation, do little to address underlying risk drivers of exposure and vulnerability to GLOFS and other geohazards, and are likely unsustainable in the face of ongoing environmental changes and lake growth"*

---

## Author Comment (AC2)

**Reviewer 2**

**General Comments:**

1) The overall framework and structure – in the current version of this study, the authors first do the 'worst case' modelling and then search for possible GLOF triggers to justify modelled results (which is actually done not very convincingly when admitting that modelled GLOFs would need very unlikely occurrence of high magnitude (X0 Mm3) ice-rock avalanche into the lake as a trigger); logical framework would start with: (i) search for possible / likely GLOF triggers for existing lakes, (ii) feeding them into definition of outburst parameters and scenarios, and (iii) leading to GLOF modelling + (iv) future lake and GLOF. I suggest to consider re-structuring the manuscript accordingly

We thank the reviewer for his careful and comprehensive comments, and apologise for the time taken to respond. We agree that the logic of the paper structure was not ideal or in line with best practice. Both the methods and results will be restructured to first address lake susceptibility (incl. triggering), before proceeding to GLOF modelling, and finally consideration of the future lake. The fundamental methodological change we will be making, based on comments from all reviewers, will be the use of r.avaflow to simulate the GLOF process chain, thereby directly linking the triggering events (large ice-rock avalanche) to the outburst event.

2) Uncertainties in input data: as the future is uncertain, I'm quite reluctant to using any single value 'worst case scenario' concept and I call for using a range of values (and scenarios) instead. Below I comment on (some of the) major sources of uncertainties which are cumulating throughout the process and are not properly treated:

We understand the reviewers concern, and ourselves are involved in several studies where the goal necessitates a full range of scenarios are modelled. However, in this paper we are aiming to address what we see as a gap in approaches/scales, fitting between large scale first-order approaches and scenario based hazard mapping, to provide an illustrated case-study showing how potential new lakes can begin to be considered in DRR planning. Is a given lake really going to present an unprecedented threat to downstream areas (relative to existing lakes)? Does such a potential lake need to be considered in the design of response strategies such as EWS? Such questions are not answered by existing studies, which to our knowledge, have not gone beyond first-order assessment of future lakes (e.g. GIS-based approaches). Likewise, we believe these questions can be answered without going to the level of detailed scenario-based hazard modelling. On a practical level, we see this gap as an important niche to fill. There has now been several years' worth of first-order modelling studies showing that glacial lake area and number will increase in the future, yet in our interactions with authorities across several countries, we see no indication yet that these future threats are being considered in DRR planning. While it would be academically possible to generate detailed scenario-based hazard maps for a potential future lake, we consider a more practical intermediate step is needed to first demonstrate that a given future lake warrants such attention and further monitoring.

We therefore argue that there is merit in simulating single worst-case scenarios, as these can be a first basis for planning response strategies that remain robust under an uncertain future. For example, by planning community awareness programmes around a worst-case of 70 minutes vs. 130 minute warning time, or ensuring critical infrastructure is located well away from a future GLOF path. We therefore will maintain the focus on worst-case scenarios, but will revise the way in which these scenarios are developed. The revised approach will focus on defining an upper limit to an expected ice/rock avalanche starting volume, and use r.avaflow to simulate the resulting GLOF cascade. For the case of the GLOF from the new lake potentially overflowing into Jialongco and causing a

simultaneous outburst from that lake, additional sensitivity analyses will be undertaken to determine if this potentially critical process chain could also occur under smaller initial scenarios.

a) the essential value at the very beginning is the estimation of breach depth (in this study referred as breach height hb). The authors provide neither details on how this
value is estimated nor what the uncertainty of this estimation is; another issue is
whether flat (<7° (rough Google-Earth-based measurement)) and pretty wide (> 450
m) moraine dam (e.g. Galong co) could ever be breached; and if it is breached, the
crucial question is how deep (longitudinal profile of the breach is typically far from flat –
I mean, if you have a vertical difference between the lake level and the toe of moraine
dam 40 m (this is how you define breach depth, right?), lake level decrease in case of
breach will be less than that (it is not going to be breached to 0° slope), depending on
longitudinal width of dam body; this is actually seen in Fig. 2b: if you define breach
depth in this way, you should not use the same value in calculating released volume,
because it differs to the lake level decrease (and in turn it leads to substantial overestimation of
released volume))
We agree that these parameters and assumptions on breach depth were not well substantiated. The revised modelling approach using r.avaflow will reduce the need for these assumptions around the dam breaching and/or overtopping mechanisms. The erodible area of the dam will still need to be defined, and will be done based on careful consideration of the dam characteristics including any possible ground ice (relevant for Galongco only). In addition, we've since obtained bathymetry measurements from both Jialongco and Galongco that will be used for the modelling.

b) in the next step, the authors use this pretty uncertain value to estimate released
volume (which is not correct in my opinion, see above) and breach parameters, using
Froehlich (1995) empirical relationships; but it is important to realize that: (i) Froehlich
(1995) is based on compiled information of man-made earthen dam failures, not
natural dams; (ii) failure mechanism of most of these cases in the database was piping,
not overtopping; (iii) released volumes in the dataset was mainly <1Mm3; and >100
Mm3 in only two cases (Oros, Teton); with expected released volumes 25, 70 and 262
Mm3, you are extrapolating far beyond observed data of Froehlich (1995) and the
uncertainty is unknown (Froehlich, 2004 should be checked).
Based on this, and other comments, we will no longer use the Froehlich approach. The GLOF process chains will be simulated using r.avaflow, with breach parameters dynamically calculated. Important model parameters (e.g. internal and basal friction angles of the solid material, the fluid friction number, and the coefficient of erosion) will be empirically defined, with references provided.

3) Timing - Using Eq. 4, calculated time for breach formation of Galong co is 153 min, but
you expect peak discharge in Nyalam in 82 min -> please explain what times are you
referring to (82 min from breach initiation, from peak discharge at the dam (when from
breach initiation?) or from development of the breach?); being as clear as possible is
especially important when talking about EWSs, presenting hydrographs at the dam would
be beneficial.
A new figure will be included giving the hydrographs at the dam – thank you for this suggestion! For clarity, all timing will be revised to be relative to the impact of the initial mass movement into the lake.

4) GLOF likelihood – this section gives some largely general statements and qualitative RSbased
observations and looks more like a discussion rather than result to me; Tab. 3
summarises first order GLOF susceptibility factors, but this study is not a first order

assessment – it is a detailed study of two existing and one potential future lake; what is shown in Tab. 3 is perhaps true for most of the lakes in the region (warming climate, steep slopes and crevassed glaciers upstream, …) and leaves the question of GLOF likelihood open; the use of >30° threshold for initiation of mass movements seems too simplifying and not really helpful for the scale you are working on

In line with other reviewer comments, this section will be renamed to "lake susceptibility" recognising that it is outside the scope of this study, and probably not feasible, to provide quantitative likelihoods linked to specific scenarios. Table 3 will be revised, and where possible, quantitative details will be provided. Please note however, many observations (e.g. permafrost characteristics) will remain RS based and to some extent qualitative. The slope threshold-based avalanche calculations will be removed, and replaced with a more detailed assessment of local geological and glacial conditions. It's true that some statements are general and could be applied to other lakes in the region, e.g, "No permafrost in dam. Degrading permafrost in surrounding slopes", but there is no basis for a more detailed quantified statement as there are no high resolution permafrost models for this area. Nonetheless, we prefer to keep the table in full, even if some entries are general, because this is the first study to directly utilise the comprehensive "check-list" table of susceptibility factors coming out of the GAPHAZ international guidance document, so an important opportunity to demonstrate both the applicability and limitations of the GAPHAZ approach.

We would note that there are few, if any, examples of quantitative likelihoods being applied in the hazard assessment of moraine dammed outburst floods, and related scenario-based studies have typically applied qualitative high, medium and low likelihoods to large, medium and small magnitude events respectively. To better reflect the scope of our study, and acknowledge that the assessment of GLOF likelihood is rather limited here to a large worst-case scenario (and therefore can't be considered a full hazard assessment), we also propose to modify the title to *"Glacial lake outburst flood magnitudes under current and future conditions: implications for disaster risk reduction in a transboundary Himalayan basin".*

5) Practical implications – the authors mention the importance of such studies for local authorities, which is in principal true and also a rationale of many similar studies. My experience is that practical utilization, however, often lacks behind. As documented by the authors, local authorities meanwhile started remedial works by themselves, meaning that they have some kind of GLOF hazard assessments and management procedures in hands. I expect these documents may not be publicly available, but attempting getting in touch with authorities in charge of these measures would be highly appreciated (and could also help to bridge the gap between what scientists and authorities are doing).

This is a great comment on the practical implications of this project. We have struggled to make progress here throughout the 3 years of the project, and despite several efforts in the field, we've been unable to enter into exchange with the local authorities and learn details from them about what is planned in terms of DRR measures. In Tibet, and close to a military controlled border region, exchange with authorities is even more challenging than in other regions we have worked. It seems almost everything is explained as being "for military purposes". Hence, the onset of remedial measures, has come as a surprise, and there is no evidence that these measures are underpinned by GLOF modelling. A desire to bridge this gap between scientists and authorities is exactly why we focussed this study on the methodological space between first-order assessment approaches, and detailed scenario-based hazard assessment. In our view, it is within this space that we can most reasonably demonstrate the importance of considering future lakes as part of DRR planning. Going to the level of comprehensive scenario-based GLOF modelling and hazard mapping only makes sense in our view once there is buy-in from local authorities, particularly with regards to the threat of a future lake. Otherwise it remains an academic exercise only.

**Specific Comments:**

L19-20: please comment on what can be done to reach this ambitious aim (not a part of the study)
We agree that our study rather does not feed directly into "decision making" and we will remove reference to this in the abstract. Nonetheless, as described in the responses above, we do believe our study provides fundamental insights for future-proofing DRR planning. For example, demonstrating that land zoning or EWS planning based on the threat from Jialongco and Galongco will possibly be insufficient to deal with the threat from a large new lake forming some decades later in the same basin. Anecdotally we know there are plans from international donors to fund the design and implementation of a transboundary EWS in this basin, so such messages are important of feed into the scoping of such projects.

L40: high magnitude
As per comments from other reviewers, we will avoid subjective wording here, and rather refer to the reported GLOF frequency of 1.3 per year for the Himalaya since the 1980s (after Vey et al 2019).

L52: I would not call 17 GLOFs overt the Tibet since 1935 'particularly common'
Agree – sloppy wording. We'll revise factually to state "At least 17 GLOF disasters (causing loss of life or infrastructure) have been documented in Tibet since 1935, mostly....."

L68: these numbers are confusing; you mentioned 3-fold increase, does it mean that future doubling in border areas of China – Nepal is thus below average?
Agree – taken out of context of the original paper these numbers are confusing and cannot be directly compared. To avoid confusion we'll remove the reference to a 3-fold increase in risk, and focus just on the doubly of transboundary lakes, which is the most relevant information for the present study.

L82-85: not sure this is met
We agree that our study rather does not feed directly into "decision making" and we will remove reference to this. Nonetheless, as described in the responses above, we do believe our study provides fundamental insights for future-proofing disaster response planning. For example, demonstrating that land zoning or EWS planning based on the threat from Jialongco and Galongco will possibly be insufficient to deal with the threat from a large new lake forming some decades later in the same basin..

L92: please consider adding description of 2(3) studied lakes in this section
Thanks for this excellent suggestion. We'll add a physical description of the 3 lakes, including key physical parameters and for the two current lakes, details on assessed hazard/danger levels provided in previous studies. This will also help better justify the focus on these lakes.

Fig. 1: please consider adding topography info; there are many dangerous lakes in the region – the authors are asked to justify why they focus on these two existing and one potential future lake (while there are other lakes forming currently)
We propose to add a series of inset photos showing details on the three lakes. Linked to the physical description added to the text (previous comment), the images and text will justify the focus on these 3 lakes.

L164: methodology of obtaining hb is not clear
Under the new modelling approach, hb will not be estimated in this way, but will be dynamically calculated during the simulation.

L178: what breach scenarios?
Apologies, this should have referred simply to the 3 worst-case scenarios. Text will be revised.

L189: this value needs justification
Manning's n will not be required for the updated modelling with r.avaflow. Key model parameters (including the internal and basal friction angles of the solid material, the fluid friction number, and the coefficient of erosion) will be empirically defined.

L210-214: this approach seems too rough for detailed case study like this one
Agree – the approach to assessing ice/rock avalanche susceptibility will be revised, and will be based on detailed consideration of local geological and glaciological conditions.

L247: considering uncertainties behind a single-number result, I found a range of values highly desirable
See response to general comment (2)

L252 ms-1 when talking about velocities (please check throughout the manuscript)
Thank you – will be checked throughout.

L274-297: this is contradicting; on the one hand you expect 48 m deep breach of very flat moraine dam and on the other hand you find erosion unlikely?
Wording will be clarified. The lack of erosion potential is referring to flow path downstream of the lake, where channel slopes are relatively gentle. However, you are right that the 48 m deep breach is also rather unlikely given the very flat moraine dam – and this is mentioned later in the manuscript. Under the new approach to modelling with r.avaflow, the dam breach might be considerably less, and wording/values will be revised accordingly.

Table 1: please specify timings (see my general comments); Jialongco – are these values of the lake before or after the remediation?
Timings will be revised based on the new simulations, and reported consistently relative to the time of avalanche impact into the lake. For Jialongco, we'll now include an additional simulation for the reduced lake area/volume and altered dam geometry. This will allow us to assess the effect of the recent remedial measures on downstream flood magnitudes.

L300: consider moving to discussion (see my general comments)
See response to general comment (4). We will add quantitative detail here to the extent possible. This goes far beyond discussion in our view, even if some details will remain qualitative.

L319: this is not very well-argued (most of the glacial lakes are surrounded by glacierized slopes with >30°)
The slope is not really the main factor here, but rather the smooth, ramp-like topography of the glacier tongue and likely temperate bed. We will revise the text to place more emphasis on these characteristics, including evidence of past instabilities of this tongue.

L326-343: yes, large volume ice-rock avalanches are rare and in the seismically active regions, you can't rule out the possibility of hitting the lake – you can say this about most of the lakes in the region (and most of the high mountain lakes globally); I'm wondering

whether is there any site-specific implication for GLOF likelihood?

One key factor here is the size of the mountain headwalls immediately surrounding the lake (i.e., 3000 metre high slope of Shishapangma). Another factor is the potential peak ground acceleration linked to seismic activity (we will update the table to include this information), which varies across the Himalaya and is enhanced in such steep topography. We will also revise the text to include more detail on the structural geology of the surrounding slopes and glacial conditions (derived from the high resolution DEM). Of course, we will not get to a quantified likelihood level, but will be able to make a stronger site-specific qualitative assessment.

L346: Klimes et al. actually showed that landslides in moraines are not capable of producing any large GLOF from Lake Palcacocha

Apologies – this citation will be removed.

Tab. 4: what is freeboard to height ratio? Both existing studied lakes seem to have surface outflow (freeboard = 0m); catchment stream density / order seem odd for evaluating GLOF likelihood; you also report no evidence of historical instabilities, further questioning the likelihood of such events for triggering GLOFs

Reviewer refers to table 3 we believe. As has been done is many other studies, we considered the remaining freeboard (i.e., a hypothetical line across the crest of the dam), irrespective of the fact there is surface outflow. The logic being that the remaining freeboard area still offers some protection in the case of an overtopping wave. We will add a footnote to the table with this detail. Stream density/order is related to the potential for rainfall/snowmelt triggering of a GLOF (after Allen et al. 2016 paper on Kedarnath). All factors come from the GAPHAZ guidance document. We Agree that details on historical instabilities are needed, and we will add this evidence (and references) to a revised version of the table.

Tab. 4: again, estimating possible ice avalanche starting zones with precision to 1 m2 is not appropriate considering apparent uncertainties; better use a range of values

Agree the precision is unjustified. This table and section will be revised to focus on the largest scenarios affecting each lake, and will include an estimated ratio if ice to rock (again for a worst-case), as required for input to r.avaflow.

Fig. 6: if intensities are based on flow depth only, why not to use flow depth directly?

Fair comment – the thinking was that intensities are more generic, and allow authorities to compare with other approaches and estimates, including for other hazard processes.

L418-418: please comment on a difference between values estimated here and size of the future lake considered in Tab. 1?

This comparison between estimated lake size at 2100, and the maximum anticipated lakes size will be added here. We note that the estimate at 2100 does not account for ice calving feedbacks, so is expected to underestimate actual size, as commented in the discussion section.

L433: there is not much about management planning in the study

As noted in previous responses, we believe the results of this study have important implications for DRR planning, and are relevant for local authorities, even if the results cannot directly on their own be taken as a basis for hazard mapping or EWS design.

L435-436: the authors published several studies on GLOF from potential future lakes previously

Yes that is true, but this is the first time a study has undertaken any sort of GLOF modelling for an overdeepening/future lake to give downstream flow heights, velocities etc. Previous studies have identified overdeepenings and downstream infrastructure that are within a GIS routed flow

trajectory. However, to avoid a discussion around what is or is not considered GLOF modelling, we will remove the term "first time".

L446: the greatest immediate threat from 2 existing studied lakes
Sentence will be revised (assuming it remains true with the new simulations).

L450-454: this is not suggesting any lower limit, this is estimated potential loss for given scenario; please re-word this sentence
Sentence will be removed as we will no longer use the estimated maximum potential flood volume as input to the simulations.

L456-458: maybe the remediation is still in progress?
Yes, we agree, and hope this is true, but as indicated in the response to the general comment, it has not been possible to get information on this from authorities. We'll add a line that remediation may well be an ongoing process.

L465: EWS can help to save lives, but not the immovable property (which may already be there); if the value of potentially affected immovable property is >> than the cost of remedial works, then it makes sense also to remediate the lake(s)
Fair point and we were too dismissive of remedial measures in the initial draft. Both here and in the conclusions, we will revise the wording to make it clear that a comprehensive solution is recommended, which includes (but is not limited to) remedial works at the lake.

L508: no clear conclusion on GLOF likelihood is given
While we do not provide a quantitative assessment of likelihood, we provide clear conclusions on the relative likelihood of an outburst from the lakes, concluding that Jailongco (based on consideration of all susceptibility factors) has the highest likelihood of generating a large outburst event. In the revised manuscript, based on a more detailed assessment of local geology and glaciological conditions, we'll strengthen this aspect to the extent that is possible.

L515: details about the project (planned final stage) should be presented (maybe the plan is to drain the lake much more?)
As written in previous responses, information from local authorities working in this sensitive military-controlled area, has not been shared. We'll add a line to the conclusions that remediation may well be an ongoing process.

L519-519: this is general qualitative statement which is true for many lakes in the region (not very helpful for DRR authorities I guess)
This conclusion will be revised based on the new large-volume ice/rock avalanche scenario to be simulated. Reminding authorities that such very large/high impact events need to be considered in DRR planning is important in our view, and worth the line that is included here.

L529: why are they socially less desirable? And why environmentally less desirable (GLOF is a major disturbance to the valley ecosystem)?
In view of concerns raised by multiple reviewers we will remove the reference to social and environmental sustainability as this may be subjective (depending on how local people view glacial lakes). The wording will be revised to: "Hard engineering strategies, in isolation, do little to address underlying risk drivers of exposure and vulnerability to GLOFS and other geohazards, and are likely unsustainable in the face of ongoing environmental changes and lake growth"

---

## Author Comment (AC3)

**Fabian Walter**

**General Comment:**

In their submitted manuscript, Allen et al. propose a hazard assessment around two existing and one potentially forming proglacial lake in the Puiqu River basin, Himalaya. They document the glacial environment as well as its projected development under future climate conditions, potential outburst flood triggers and modeled discharges in vulnerable downstream communities. The study gives insights into different aspects of glacial lake outburst floods highlighting particularly interesting features of the investigated cases. However, it is difficult to grasp how relevant the scientific insights from this investigation are given that the authors' judgement is too often limited to qualitative assessment. For a scientific journal submission, I was expecting more substance in view of reproducibility and representativeness of the results

We thank the colleague for his critical feedback on the manuscript and apologise for the time taken to response. We designed this study to address what we see as a gap in common GLOF assessment scales (particularly in the developing world), fitting between large scale first-order approaches and comprehensive scenario based hazard mapping. We thereby aim to provide an illustrated case-study showing how potential new lakes can begin to be considered in DRR planning. Is a given lake really going to present an unprecedented threat to downstream areas as commonly implied (relative to existing lakes)? Does such a potential lake need to be considered in the design of response strategies such as EWS? Such questions are not answered by existing studies, which to our knowledge, have not gone beyond first-order assessment of future lakes.

The methodological approach was designed to be as simple as possible, with a view that methods could be attractive for upscaling to any number of potential future lakes. However, in view of the comments from all 3 reviewers, we accept that this was scientifically weak, as many assumptions were not well substantiated, and importantly, the susceptibility assessment and triggering processes were not directly linked to the outburst scenarios. Therefore, in the revised manuscript we will start with a more comprehensive assessment of lake susceptibility, to derive worst-case scenarios of ice-rock avalanche triggered GLOFs from all 3 lakes, including a new scenario for the artificially lowered lake volume of Jialongco. The use of r.avaflow for the modelling of the entire process chain will reduce the amount of qualitative assumptions regarding the link between the trigger event and outburst wave. We'd also note that we've also since been able to measure bathymetry for the two current lakes, which is important input for r.avaflow.

We must however acknowledge that many aspects of the lake susceptibility assessment will remain partially qualitative, as for some parameters like permafrost conditions, we simply lack high resolution modelling results for this area. Likewise we would note that there are few, if any, examples of quantitative likelihoods being applied in the hazard assessment of moraine dammed outburst floods, and related studies have typically applied qualitative high, medium and low likelihoods to large, medium and small magnitude events respectively. To better reflect the scope of our study, and acknowledge that the assessment of GLOF likelihood in our study is rather limited to a large worst-case scenario (and therefore can't be considered a full hazard assessment), we also propose to modify the title to *"Glacial lake outburst flood magnitudes under current and future conditions: implications for disaster risk reduction in a transboundary Himalayan basin"*.

**Major Comments:**

My main point of criticism is that the reader of this manuscript is left with little information on how to assess validity or accuracy of the findings. In its current state, the study appears more like a presentation of important facts and qualitative judgements, which are typical for technical reports. For a scientific paper I would have expected some critical assessment of the flood risk, e.g., a benchmarking of the presented methods against previous occurrences of

outburst floods. The authors cite accounts of previous outbursts in the area (Line 106). Could they be used for this?

Unfortunately there are very scarce details on the past GLOF events occurring in the upper basin (affecting village of Nyalam), beyond the fact that they occurred. The 2015 earthquake resulted in a lot of erosion around the village, so geomorphological evidence is also not clearly recognisable. However, further downstream towards the border with Nepal, an outburst from 2016 has been well documented (e.g. Cook et al. 2018) and we will attempt to compare our worst-case scenarios against this event. Likewise, we will also compare our results against those obtained in other model-based studies within the same basin (e.g., Shrestha et al. 2010, Zhang et al. 2021)

Cook, K. L., Andermann, C., Gimbert, F., Adhikari, B. R. and Hovius, N.: Glacial lake outburst floods as drivers of fluvial erosion in the Himalaya., Science, 362(6410), 53–57, doi:10.1126/science.aat4981, 2018.

Shrestha AB, Eriksson M, Mool P, Ghimire P, Mishra B, Khanal NR. 2010. Glacial lake outburst flood risk assessment of Sun Koshi basin, Nepal. Geomatics, Natural Hazards and Risk. Taylor & Francis 1(2): 157–169. DOI: 10.1080/19475701003668968.

Zhang, T.; Wang, W.; Gao, T.; An, B. Simulation and Assessment of Future Glacial Lake Outburst Floods in the Poiqu River Basin, Central Himalayas. Water 2021, 13, 1376. https://doi.org/10.3390/w13101376

The first part of the paper presents some motivation on why to study the chosen three lake basins. However, it is not possible to verify that this corresponds to a worst-case analysis. In this case, it would be necessary to show that no lake basins could produce more serious outburst floods. As the authors argue, this depends on moraine dam geometry, water volume and trigger potential. Under these aspects it cannot be argued that the presented set of lakes is representative for worst case scenarios.

Thanks for this comment. We struggled on this aspect as we did not want to reproduce results from our earlier study (Allen et al. 2019), that was primarily the basis for focussing on the two current lakes. In view of this, and comments from the other two reviewers, we will now add a physical description of the 3 lakes to the study area section, including key physical parameters and details on assessed hazard/danger levels provided in previous studies (for Jialongco and Galongco). This will also help better justify the focus on these lakes. We will add images of the lakes to Figure 1. Likewise, we'll add further description on why this particular potential future lake was identified as a priority for worst-case analyses in the study (large size, proximity to village, evidence of supraglacial lakes already forming etc).

At too many parts of the manuscript, the authors' qualitative judgement is presented as a scientific result. In particular, in the Section "GLOF likelihood", various factors influencing or triggering outburst floods are presented, but I could hardly find any objective arguments. It seems that the only one is the estimate of a dam-overtopping wave volume, which can be 10 times as high as the "incoming mass". Here and elsewhere in the manuscript, it has to be made clear that the conclusions are based on solid scientific grounds. Otherwise, a "low probability" could indicate one catastrophic event every 5 years as opposed to several ones per year. This is not what the authors imply. In a similar sense, it is not clear what the demanded "comprehensive and forward-looking approach to disaster risk reduction" is. To me, such an approach should always be taken and I see little connection to the present study or any finding, which made the suggested strategy particularly pertinent to the Poiqu River basin compared to any other place in the world.

Effort will be made throughout the manuscript to provide a more quantitative basis for the assessment findings. The section on GLOF likelihood (renamed to "lake susceptibility") will be redrafted, with more focus on the local geological and glaciological conditions that determine large ice/rock avalanche potential. Table 3 will be revised, and where possible, quantitative details will be provided with references cited. For example, the slope threshold-based avalanche calculations will be removed, and replaced with a more detailed assessment of local geological and glacial conditions. Again, we must acknowledge that many aspects of the lake susceptibility assessment will remain partially qualitative and subjective for those factors that lack quantitative data or can be inferred only. However, we will provide references in Table 3, that support the expert judgements being made.

On the final point, we are yet to see GLOF mitigation strategies anywhere in the developing world that can be considered forward-looking. There has now been several years' worth of first-order modelling studies showing that glacial lake area and number will increase in the future, yet in our interactions with authorities across several countries, we see no indication yet that these future threats are being considered in DRR planning. We fully agree that DRR planning ultimately requires more quantitative assessment of outburst probabilities etc. However, this remains the holy grail for the assessment of existing moraine dammed lakes, let alone future lakes for which conditions of both the lake and surroundings are highly uncertain. Particularly for DRR strategies such as an EWS, we would therefore argue that there is merit in our approach of simulating single worst-case scenarios, as these can be a first basis for planning response strategies that remain robust under an uncertain future. For example, by planning community awareness programmes around a worst-case of 70 minutes vs. 130 minute warning time, or ensuring critical infrastructure is located well away from a future GLOF path.

**Specific Comments:**

I suggest including a cartoon explaining different lake formation scenarios and specifically the hydrological base line. To be honest, I had to stare for some time at Figure 3 of Benn et al. (2012) to understand this concept. On the other hand, I never grasped the meaning of the "the lowest point where the glacier surface intersects the terminal moraine" (it seems that by definition, the glacier and the terminal moraine should not intersect). Similarly, when the future evolution of the lakes is described, longitudinal profiles would be extremely helpful. This would help the reader to understand Figures 7 and 8 and appreciate the shown information.

We thank the reviewer for their suggestions about how to improve the description of the concepts we employ to examine the likelihood of future lake formation on glacier RGI60-15.09475. To ensure that the approach we have followed is as clear as possible, we will improve the description of the lake formation scenarios in section 3.3 (Future lake development) and clearly refer to Figures 3 and 19 in Benn et al. (2012), as these two figures already provide a good visual representation of how supraglacial meltwater ponding can lead to glacial lake formation. We will be clear to distinguish between the initial stages of supraglacial meltwater ponding and lake formation (when the terminus of the glacier is still in contact with the terminal moraine) and the later stages of lake expansion (when, as the reviewer suggests, the terminus of the glacier would not be in contact with the terminal moraine).

We also thank the reviewer for their suggested improvements to Figures 7 and 8. Figure 7 already incorporates longitudinal profiles of glacier surface elevation, surface slope and surface elevation change, but we agree that Figure 8 would benefit from the addition of centreline profiles of glacier surface elevation under different scenarios of thinning in 2050, 2075 and 2100.

The flood model is a key ingredient to this investigation. Although I agree that too many technical and mathematical details are not appropriate for this study, I was wondering what the main parameters and boundary conditions of this model are. Apparently, the flood volume, some time scale of drainage initiation and dam geometry play a role and it would be interesting to hear how these parameters drive the model.

In view of the comments from all 3 reviewers, we accept that the previously used modelling approach was scientifically weak, as the triggering processes were not linked to the outburst scenario, leading to reliance on assumptions (e.g., concerning Potential Flood Volume) and empirical relationships to define parameters. Therefore, in the revised manuscript we will use r.avaflow to simulate worst-case ice-rock avalanche triggered GLOF process chains from all 3 lakes, including a new scenario for the artificially lowered lake volume of Jialongco. This has the advantage of the outburst process (and related parameters) being dynamically simulated based on the ice-rock avalanche triggering event and overtopping wave. Key model parameters (including the internal and basal friction angles of the solid material, the fluid friction number, and the coefficient of erosion) will be empirically defined, with references added.

Lines 45-47: "... most scientific attention has focused upon ..." I do not agree with this statement. In the jökulhlaup literature, ice-marginal and subglacial lake drainages have also

received a lot of attention. Whereas I cannot say which scenario has been most prominent, I would refrain from an absolute statement on scientific attention.
Good point – we will qualify this statement as *"In Asia, most scientific attention has focused upon…."*

Lines 175-176: "B_w and h_b are fully obtained" measured?
Under the revised modelling approach with r.avaflow, these values will be dynamically simulated.

Line 178: Reference for HEC-RAS is needed.
Section to be revised, based on modelling approach with r.avaflow.

Line 185: Reference for DEM's is needed.
Will be added.

Line 189: Reference for Manning roughness value is needed.
Manning's n will not be required for the updated modelling with r.avaflow. Key model parameters (including the internal and basal friction angles of the solid material, the fluid friction number, and the coefficient of erosion) will be empirically defined, with references provided.

3.2 Lake susceptibility assessment: Presenting the likelihood calculations seems appropriate here.
Table 3 will be extensively revised and to the extent possible, quantitative assessment of the various GLOF susceptibility factors will be provided. We note, as in the general comments, that some qualitative expert judgment is inevitable in this field, as outlined in International Guidelines to Glacier Hazard Assessment (https://www.gaphaz.org/files/Assessment_Glacier_Permafrost_Hazards_Mountain_Regions.pdf).

Line 231: "considering the factors outlined by …" these factors should be specified.
We will amend the text here to reiterate that the surface slope and velocity of the glacier are both suitably low to allow for substantial meltwater ponding, as it already stated on lines 220 and 221.

Line 257: I suggest a paragraph break here.
Noted.

Line 457: "recent removal of much of the frontal moraine …" this needs a reference.
I reference to Figure 9 will be made here.

Lines 466: What are "capacity building programs"?
Will be revised to "programs to strengthen local response capacities"

**Figures:**

Figure 1: The lakes at Cirenmaco and Jialongco are difficult to discern.
Noted – we will improve the colours and symbology

Figure 2: The font sizes are a bit small, but h_b defined in Panel A seems to disagree with Panel B.
Noted – figure will be replaced to reflect new modelling approach.

Figure 4A: Where is the lake? The blue outline or the light blue polygon? A different color scale for maximum flow height would help.
Noted – we will improve the colours and symbology

Figure 5: It is difficult to tell where the lakes are. I do not see any blue patches.
Noted – will try to improve contrast of the colours in the images.

Figure 6: The two images in Panels C need some annotation. What does the reader see in these images? Why is one so dark and the other one bright? What happened between the two?

Caption will be expanded for panel C. The significant expansion of infrastructure seen between 2015 and 2017 should be obvious from these panels, despite the differences in colouring from the two satellite images.

Figure 8: This is the future lake site, right? Where will the lake form? The colored outlines make little sense and are hard to sea. Which direction does the ice flow?
Red line is the maximum projected overdeepening, and the blue/purple colourings are extent to which the lake is projected to emerge under the different future scenarios. We will improve colouring and information in the caption. Direction of ice flow will be added, thanks!

Figure 9: Some arrows and annotation as well as scale bars are needed.
Noted – will be added.

---

## Author Response (AR1)

**Department of Geography**

Dr. Simon Allen
University of Zurich
Department of Geography
Winterthurerstrasse 190
CH-8057 Zurich

simon.allen@geo.uzh.ch

The Editor
Natural Hazards and Earth System Sciences

Zurich, 31 August 2022

**REVISED MANUSCRIPT FOR CONSIDERATION**

Dear Editor

On behalf of my co-authors, I am happy to provide a heavily revised version of our manuscript for further consideration:

*"Glacial lake outburst flood hazard under current and future conditions: worst-case scenarios in a transboundary Himalayan basin"*

Let me sincerely apologize at this point for the significant length of time it has taken us to prepare this revised version of our manuscript. I'm aware we have tested the patience of the journal and reviewers after already receiving several extensions, due to various health (COVID) and work-related challenges. As the editor will be aware, the revisions we were required to undertake were major, involving a fundamental change to the modeling methodology and tools used. This was not trivial, and has led to almost a complete rewrite of the methodology, results and discussion section.

Nonetheless, the fundamental aims and scope of the paper remain unchanged, and we are confident now that based on the extensive reviewer comments we have produced a robust and significant contribution to the field. In particular, we would argue that comparable studies have rarely considered worst-case scenarios involving very large avalanche-triggered GLOF process chains of the magnitude we assess here. Our key findings, regarding potential warning times and the limitations of remedial lake lowering, will be of high relevance to those engaged in DRR planning in the fragile Himalayan region.

Below you will find our point-by-point final response to all reviewer comments. We thank you for your further consideration.

Yours Sincerely,

Dr. Simon Allen
(on behalf of the author team)

**Reviewer 1**

**Major Comments:**

Allen et al. used a suite of well-established tools to assess possible trigger mechanisms of GLOFs, and to model hydro-mechanical characteristics and runouts of the flood waves. Unfortunately, the authors missed the opportunity to couple such triggers with the subsequent failure of moraine dams. Recent software such as r.avaflow have recently improved the understanding of GLOF processes, allowing for pseudo-probabilistic assessments of GLOF triggers and impacts under a range of adjustable and testable boundary conditions. However, given that the authors considered only three worst-case scenarios of moraine-dam failure, largely decoupled from the initial triggers, I see only limited progress in calculating the likelihood of observing a GLOF of a given magnitude, though this is repeatedly stated in the manuscript. Clearly, the authors present a thorough expert-based assessment of potential conditioning or triggering factors. Yet it remains unclear how this traditional, rather subjective procedure brings GLOF hazard and risk assessment forward with regards to the many previous case studies that we saw in the past years. Thus, the authors' goal to estimate "the core hazard dimensions of GLOF magnitude and likelihood (or probability)" (L245-246) only remain partly fulfilled in my opinion. If this manuscript is intended to improve early warning and risk management, then the reader (at best a person involved in such tasks) might expect a more objective use of the terms hazard or probability, that is a numeric value between zero and one. A probability or return period might provide a robust baseline for decision makers compared to the current distinction between a 'high' or 'low' level of susceptibility or impact. Fortunately, most of the data (including high-resolution DEMs and satellite images) for such an extended appraisal already come with this manuscript. I thus would like to encourage the authors to revise their manuscript accordingly, considering a much wider range of scenarios in order to proceed towards a more objective assessment of current and future GLOF hazard in their study region.

We thank the reviewer for this critical feedback on the manuscript, and sincerely apologise for the time taken to revise the paper. The comment from the reviewer gets right to the fundamental contribution our study is trying to make, and unfortunately this has perhaps not been adequately introduced or fulfilled in the initial manuscript. We are aware of the value of r.avaflow and other approaches for undertaking detailed, scenario-based hazard assessment, and are using exactly these models in other studies, where comprehensive hazard mapping is the desired outcome. However, what we are aiming to achieve in this particular paper is to address a gap in approaches/scales, fitting between large scale first-order approaches and scenario based hazard mapping. We also recognise that comparable studies have rarely considered worst-case scenarios involving very large avalanche triggered GLOF process chains. We thereby aim here to provide an illustrated case-study showing how worst-case scenarios, including from a potential new lake can begin to be considered in DRR planning. Is a potential new lake really going to present an unprecedented threat to downstream areas (relative to existing lakes)? Does such a potential lake need to be considered in the design of response strategies such as EWS? Such questions are not answered by existing studies, which to our knowledge, have not gone beyond first-order assessment of future lakes (e.g. GIS-based approaches).

On a practical level, we see this gap as an important niche to fill. There has now been several years' worth of first-order modelling studies showing that glacial lake area and number will increase in the future, yet in our interactions with authorities across several countries, we see no indication yet that these future threats are being considered in DRR planning. While it would be academically possible to generate detailed scenario-based hazard maps for a potential future lake, we consider a more practical intermediate step is needed to first demonstrate that a given future lake warrants such attention and further monitoring.

We agree that ultimately, DRR planning can be best informed by more quantitative assessment of outburst probabilities. However, this remains the holy grail for the assessment of existing lakes, let alone future lakes for which conditions of both the lake and surroundings are highly uncertain. Particularly for DRR strategies such as an EWS, we would therefore argue that there is merit in simulating single worst-case scenarios, as these can be a first basis for planning response strategies that remain robust under an uncertain future. **In fact, for EWS design, in our experience worst-case scenarios are the most important consideration, yet have often been neglected in comparable studies**. For example, based on our results,

response actions at the border with Nepal can be planned considering a minimum warning time of 30 minutes (our study), compared to a time of 120 minutes simulated for more gradual breach processes (Zhang et al. 2021). Likewise, our results help ensure critical infrastructure in Nyalam is located well away from a worst-case or future GLOF path.

The decision to use HEC-RAS for modelling downstream flood impacts was initially driven by a desire to keep the approach as simple as possible, with a view that methods could be attractive for upscaling to any number of potential future lakes. However, in view of the comments from all 3 reviewers, we accept that this was scientifically weak, as the triggering processes were not linked to the outburst scenario, leading to reliance on assumptions (e.g., concerning PFV) and empirical relationships. Therefore, in the revised manuscript we have used r.avaflow to simulate worst-case ice-rock avalanche triggered GLOFs process chains from all 3 lakes, including a new scenario for the artificially lowered lake volume of Jialongco. The selection of worst-case avalanche volumes has been comprehensively described, and based on a more nuanced assessment of triggering factors described in section 4.1.

In conclusion, we would summarise the novelty and contribution of the revised manuscript as being **a first study to demonstrate the effect of future lake development on downstream flood magnitudes (and also timing of peak discharge), with implications for DRR planning.** In addition, we are now able to **demonstrate the limited effect of lake lowering on downstream flood magnitudes and timing relating to worst-case events**.

Given the valid concerns of the reviewer, we also have modified the title to more accurately capture the scope and rationale of the paper:

*"Glacial lake outburst flood hazard under current and future conditions: first insights from worst-case scenarios in a transboundary Himalayan basin"*

The abstract has been further modified to make it clear the focus is on the timing and magnitude of simulated worst-case outburst events.

**Detailed comments:**

-L6: 'far reaching': in terms of run-out? Social consequences? Media coverage?
Revised to *"where societal impacts can extend far downstream"*

-L10: 'well-known dangerous': please consider avoiding subjective terms.
Revised to *"previously identified dangerous lakes"*. This is in line with the numerous studies cited in the text that assess these lakes as being dangerous.

- L26-27: 'Based on … capacity building programs': Hardly mentioned in the text, consider revising or deleting.
Revised to *"awareness raising programs"*. We feel it's an important point to make, even if not a central component of this study, because authorities (and funders) often overlook such important social dimensions of DRR.

- L39: 'rapidly in both size and number': how much? Consider using recently multi-temporal glacier lake inventories.
The more recent studies have been cited.

- L40: What is the frequency of GLOFs in the Himalaya? If I am not mistaken, most researchers deem GLOFs a low rate of occurrence.
We now report the GLOF frequency of 1.3 per year for the Himalaya since the 1980s (after Vey et al 2019).

- L47f: 'Such lakes can have volumes >100 m³', according to ref. [16]
Sentence revised to *"volumes larger than 100m³"* and the new reference added.

- L69: 'within the eastern Himalayan region': why there?
Have removed this part of the sentence – it was stating the obvious as the largest border area between Nepal and China is in the eastern Himalayan region.

- L75: 'are lacking': suggest to tone this a bit down, as it undermines previous appraisals
We don't intend to undermine the previous studies which have considered future lakes – many of which we were involved in or led – but these were studies aiming to identify district or regional-scale changes in GLOF risk. Other than a general indication of how hazard or risk could change, they don't provide information for specific lakes, that is required for DRR planning.

In view of the concern of the reviewer, wording of this sentence will be revised to *"yet practical examples on how to account for future lake development in local GLOF risk assessment and risk management have been rarely demonstrated".*

- L79-82: Is this statement valid for all glaciated high mountains? Or only the Himalayas?
To our knowledge the statement is valid globally. We know of no cases where local authorities have begun to consider future lakes in the their hazard and risk assessment. Hence, we leave as a general, globally applicable statement.

- L89-91: The last sentence could be deleted, if shortening is needed. The introduction ends well with a strong presentation of the research goals.
Agree, sentence removed.
* * *
- L93: 'considering potential GLOF impacts'
Agree, wording revised

- L97: 'mean annual air temperature'
Wording revised

- L99: are these precipitation rates (I think these are monthly?) the maximum or the average recorded in a given period?
Thanks for pointing this out. These are average monthly rainfall totals (not rates). Text will be corrected.

- L105: 'GLOF danger': replace with 'GLOF hazard'?
Disagree – "danger" was the term deliberately used in Allen et al. 2019 as the study went beyond hazard to consider also exposure of infrastructure (but fell short of a full risk study).

- L107: 'losses of up to US$4 million': is this in today's currency?
These were values of 2015 – we have clarified this in the text.

- L111-112: 'a large and persistent threat, considered as one of the most dangerous lakes in Tibet': suggest to tone this down again. Given that previous appraisals used different criteria and thresholds, we may wish to avoid confusion what is now the 'most dangerous lake' in a given region.
The point we want to make here is that irrespective of different methods and criteria used, multiple studies point to this lake as being one of the most dangerous/hazardous in the region. We have added a further paper to support this. Also we've removed the term "large" as this is subjective. Text revised to *"identified by multiple studies as being one of the most critical lakes in Tibet"*
* * *
- Figure 1: Please highlight the border between Nepal and China.
The main image is in fact clipped to the border. To better indicate this, we have added the label of 'border' and also indicated 'Nepal' to the Bhotekoshi river label below this point. We have also specified in the caption that the border passes through Zhangmu.

- L139: So estimating lake depth comes with zero uncertainty?

Over the past 2 field seasons we have been able to measure bathymetry of both Jialongco and Galongco lakes, and we now use this data as input to the GLOF process chain modelling (see new section 3.2), avoiding any need for empirically derived estimates.

- L150: To my knowledge GlabTop models the glacier thickness in a bedrock topography. Did the authors add the moraine on top of this bedrock depression? How high / wide is this moraine? If not, how susceptible is a bedrock depression to overtopping? Did the authors consider sediment input from supraglacial debris into the lake?

Reviewer is correct in their interpretation. In keeping with a philosophy of a worst-case simulation from the future lake, we do not consider additional height (and freeboard) from a moraine dam, the height of which can only be guessed. We take the bedrock topography as simulated from Glabtop. Likewise sediment deposition into the lake, thereby reducing the potential lake volume is not considered. These aspects/limitations have now been noted in section 3.2.

- L155: Please consistently use Fig. / Fig / Figure (or any other option) throughout the manuscript. Sometimes the abbreviation is written with a dot, sometimes not.

Corrected.

- L154-164: the PFV approach may underestimate the amount of water that is generated from mass flows entering the lake and causing a splash wave. Given that the authors consider ice avalanches and landslides entering the lakes, how useful is it to use the PFV approach?

The revised modelling approach using r.avaflow generates the flood volume based on the actual simulated avalanche scenario into the lake. See revised section 3.2.

- L156: The PFV concept further seems to confuse mean and maximum depth. Why should the drain empty completely if the depth of the breach is equal to the mean depth of the lake? Again, the assumption of a fixed breach depth might simplify (or even underestimate) the worst-case scenario of GLOF volumes.

This was a typo, and should have referred to maximum depth. In any case, the PFV approach has been replaced, and flood volumes are dynamically generated with r.avaflow. See revised section 3.2.

- L162-163: 'In comparison, … (Xu, 1988)': what is this comparison good for?

This paragraph has now been removed. See revised section 3.2.

- L165: 'the DEM accuracy is unknown': what is the accuracy of the 1-m DEM then?

No other high-resolution DEMs of Poiqu are publicly available to assess the vertical and horizontal accuracy in the Pleiades DEM. However, Berthier et al. (2014) computed vertical accuracy of Pleiades DEMs over the Agua Negra study site and reported mean vertical biases ranging from 0.99 to 1.33 m without GCPs. Similar accuracy level (0.3 m) was also reported by Zhou et al. (2015) from the comparison of a Pleiades-1 DEM with an airborne LiDAR DEM. Similarly, without ground control points (GCPs), the horizontal location accuracy of the images was estimated as 8.5 m (CE90, Circular Error at a confidence level of 90 %) for Pléiades-1A and 4.5 m for Pléiades-1B (Lebègue et al., 2013; Oh and Lee, 2014).

References

Berthier, E., Vincent, C., Magnusson, E. et al. (2014). Glacier topography and elevation changes derived from Pleiades sub-meter stereo images. The Cryosphere, 8, 2275-2291.

Zhou, Y., Parsons, B., Elliott, J.R., Barisin, I and Walker, R.T. (2015), Assessing the ability of Pleiades stereo imagery to determine height changes in earthquakes: A case study for the El Mayor-Cucapah epicentral area. Journal of Geophysical Research- Solid Earth, 120, 8793–8808.

Oh, J. and Lee. C. (2014), Automated bias-compensation of rational polynomial coefficients of high-resolution satellite imagery based on topographic maps. ISPRS Journal of Photogrammetry and Remote Sens., 100, 12–22.

Lebègue, L., Greslou, D., Blanchet, G., De Lussy, F., Fourest, S., Martin, V., Latry, C., Kubik, P., Delvit, J.-M., Dechoz, C., and Amberg, V. (2013). PLEIADES satellites image quality commissioning, Proc. SPIE 8866, Earth Observing Systems XVIII, 88660Z (23 September 2013), doi:10.1117/12.2023288, 2013.

- L171-172: Again no uncertainties in these equations? Propagating estimates through Equations 1-4 might have already generated a substantial amount error that remains not completely untouched in the remainder of the manuscript. Also, why did the authors not choose a physical dam break model such as BASEMENT?

The Froehlich equations are not used in the revised manuscript. The simulation of the entire process chain has been undertaken using r.avaflow.

- L180: What is the mechanism that causes overtopping? A splash wave? Or overflow by a (gradually) growing lake volume?

In the revised manuscript, the overtopping mechanism has been simulated using r.avaflow. Given that ice-rock avalanching is identified as the most feasible GLOF triggering mechanism for all 3 lakes, we expect a splash wave to be the primary mechanism. Gradual collapse of a lateral moraine wall could feasibly lead to gradual overflow in the case of Jialongco and is mentioned in the susceptibility assessment.

- L189: 'Uniformity of land cover and lack of vegetation': strong statements - please show this. Source for the chosen value of Manning's n?

Manning's n was not required for the updated modelling with r.avaflow. Key model parameters (including the internal and basal friction angles of the solid material, the fluid friction number, and the coefficient of erosion) have been empirically defined, with references added (see section 3.2).

- L201-2014: I found it somehow confusing to see first the paragraph on flood modelling, followed by a paragraph that describes potential causes for this flood. Again, it may sound a bit harsh, but considering these triggers decoupled from the flood routing models seems to represent not the state of the art in GLOF modelling. Determining the likelihood (again without a probability) for a rock or ice avalanche from a fixed angle of reach might be suitable for a large-scale assessment of GLOF susceptibility that pursue a rapid screening for sites that need more attention. Here, the authors have identified such sites, so there is (from my perspective) no need to rely still on the more traditional side of GLOF susceptibility appraisals. In other words, if there is a potential for rock to detach, what could be its size? How rapidly may this rock avalanche enter a lake? What would the displacement wave look like? It also remains unclear why the authors had chosen exactly those susceptibility factors in Table 3 (or 2). Suggest to add references to this table that showed if these susceptibility factors in deed had positive (+) relevance for triggering a GLOF.

Thanks for these comments. We agree, the pervious structure was not logical or in keeping with best practice. In the revised manuscript, the methodological description on lake susceptibility (3.1) comes before the section on GLOF modelling (and likewise in the results). The focus on a primary (and most likely) trigger of an avalanche and resulting GLOF process chain justifies the choice of modelling approach (r.avaflow). Use of topographic potential (angle of reach) for the rock/ice avalanche likelihood has been removed, and replaced with a more comprehensive assessment of local geological and glacial conditions to estimate possible upper limits to an initial avalanche volume (section 4.1). Characteristics of the avalanche flow and its interaction with the lake are simulated with r.avaflow. The susceptibility factors in Table 3 are taken from the GAPHAZ international guidance document (2017), which comprehensively outlines there relevance (with cited evidence) for GLOF triggering.

- L205: There seems to be a referencing issue: The item Table 3 must be Table 2.
Thanks –corrected.

- L222: 'cannot': 'may not'?
Wording revised.

- L217-231: It's good to know about historic changes in glacier velocity and elevation, but these insights seem to be decoupled from the future lake development. Please try to make this link clearer or consider deleting.

As suggested above, the text in section 3.3 has been improved to ensure that the relation between glacier dynamics and geometry, and their impact on future lake development, are more clearly set out. Contrary to the reviewers statement here, historic changes in glacier surface velocity and elevation are key to supraglacial meltwater ponding (e.g. Quincey et al., 2007; King et al., 2018) and initial lake development, which is now made clear to the reader.

- L241: Please elaborate more on what you mean with 'replicate'.

Will be revised to *"seen again"*

- L237-243: Do these scenarios also consider accelerated lake growth and glacier retreat, once the lake has formed?

They do not, which we are careful to acknowledge in the discussion of the results. We have added text that lake expansion rates will likely be much higher when processes such as ice front calving begin to occur in response to lake expansion. Unfortunately, inclusion of ablative processes such as calving are beyond the scope of the approaches we employ in this work.

- L246: 'probability': again, there is no probability involved in this study. 'exposure of buildings' has not been mentioned in the Methods. 'A full hazard … assessment … is beyond the scope of this study': why then calling this paper then a hazard assessment in the title?

We accept the reviewers point. The opening sentence has been revised to *"We focus below on results relating to the susceptibility of the lakes to produce an outburst event, and the potential magnitude of downstream impacts, as simulated under worst-case outburst scenarios".*
Furthermore, the results section previously titled "GLOF likelihood" has been revised to "lake susceptibility and susceptibility development" (consistent with the methodology section). This section now comes first, before the section on downstream impacts.
Please note also the revision to the paper title: *"Glacial lake outburst flood magnitudes under current and future conditions: implications for disaster risk reduction in a transboundary Himalayan basin".*

- L250: Why did the simulations stop the border? Isn't it one of the goals of this paper to show the transboundary hazard from GLOFs?

We chose to only simulate until the border as that was the extent of the high resolution DEM we created, and further extension would have been at significant financial cost and beyond the scope of this paper. We have noted this now. By simulating the arrival time and magnitude of the flood at the border, we believe this provides the intended basis for discussing transboundary hazard. Of course, detailed planning of disaster risk management strategies will require modelling further downstream. For this, our worst-case results can be taken as input to a new model run starting at the border.

- L266: Why are the flow depths of 25m more realistic? Calls for some sensitivity analysis.

The problems with the lower resolution DEMs were clear and could be seen with simple profiles taken along the stream (stream path flowing uphill for example with obvious steps/blockages in the topography). However, with the process-chain modelling using r.avaflow, involving much higher peak discharges, we now see pooling in at the stream juncture that is consistent with what has been observed for other large mass flows (e.g. Chamoli). This is now mentioned in section 4.2.

- L274-276: Again, very simplified assumption on sediment entrainment and aggradation.

Section has been revised.

- L285: 'significant': how much?

This section has been revised based on the modelling of the cascade with r.avalfow. While it is not trivial to isolate and identify the volume that is added following the overtopping into Jialongco, we have added hydrographs to Figure 5, giving discharge values immediately upstream and downstream of the impact into Jialongco.

- L287: 'high-impact low-probability': How high is this impact and how low is this probability?
As we are dealing with worst-case scenarios, we are assuming low (or very low) probabilities. As the reviewer no doubt knows, it's extremely difficult in the field of glacial hazards to assign quantitative probabilities, unless dealing with reoccurring events (ice dammed lakes for example). We now model the process cascade explicitly, i.e., overtopping of the GLOF from the future lake into Jialongco and the resulting combined downstream flow magnitudes in Nyalam and Zhangmu are given in Table 3 and Figure 3c.

- L288: 'requires more sophisticated modeling': unclear why this has not been considered in this specific case study that offers the data to do this sort of modeling.
In the revised manuscript we have now simulated this cascading process using r.avaflow. This has been made possible now that bathymetry data is available, which was not the case for the first version of the manuscript.

- L301: 'likelihood or probability': replace with 'possibility'?
Here and throughout, speak now only of likelihood, which can be qualitative (eg. high, med, low).

- L310: 'low': how low?
We don't believe given the scope of this study it is feasible to go beyond a qualitative probability level.

- L311: 'can be effectively discounted': why? And how efficient?
As above, we find it unfeasible to be more quantitative with such statements. As pointed out be other reviewers, the huge width of the dam make catastrophic self-destruction of the dam extremely unlikely.

- L316: 'large slope instabilities': In-SAR might help to detect those?
Noted, but outside the scope of this study. We have added in the discussion a point on the potential for In-SAR to be included in a monitoring/EWS strategy.

- L326-347: Most of this section could be redundant or offer much more potential for discussion, if the authors considered a suite of lake impacts in this study.
As noted in the response to the general comment, we prefer to keep the scope of this study on single worst-case scenarios, and believe there we provide sufficient justification for this. Using r.avaflow allow us now to determine the outburst volume and downstream flood magnitudes linked to an assessed worst-case avalanche volume, and the results have been rewritten accordingly.

- L343: 'making this a high magnitude, but very low likelihood process chain': how high is this magnitude and how low is the likelihood?
Section has been rewritten based on new simulations, which directly link the worst-case avalanche volumes with the lake outburst volume. The estimation of likelihoods however remain evidence-based qualitative estimates, and we refer to several relevant papers. Later we place the simulated peak discharges in the context of other GLOF events recorded globally (with reference to Carrivick and Tweed 2016).

- L344-345: 'active instabilities are clearly evident': so, why not show these?
This zone of instabilities was, and still is shown in Figure 2 (a). This is also clearly evident in the new photo included in Figure 9c.

- L349-355: Similar to the comments above, it remains unclear, how the authors define the likelihood (or magnitude) of a GLOF.

Section has been rewritten based on new simulations, which directly link the worst-case avalanche volume with the outburst volume. The estimation of likelihoods being in the range of low to very low, remains an evidence-based qualitative estimate. Later we place the simulated peak discharges in the context of other GLOF events recorded globally (with reference to Carrivick and Tweed 2016).

- L360: 'Con.', 'Trig.', 'Mag.': please write out fully. 'Catchment drainage density': why is this important. 'TP = 3280': Unclear what this means, please explain. 'Steep moraine slopes >500 m high': really? A 500 m high moraine dam?

Con., Trig., and Mag., are now spelled out in the caption. The reader is referred to the GAPHAZ guidance document for full discussion and explanation of why the different factors are important, and the evidence this is based on. We cannot repeat here for all factors. The moraine slope >500m (revised now to 400m) refers to the lateral moraine walls and adjacent talus slopes. We have revise the wording.

- L372: How complete is OSM in this region?

As noted in the text, both OSM and latest google earth imagery were used to ensure all exposed buildings were identified.

- L387: 'While we did not simulate beyond the border owing to the limited coverage … ': needs to come earlier.

Noted, and we have moved earlier.

- L435: 'applied these approaches for the first time': really?

To our knowledge, yes, this is the first time a study has undertaken any sort of GLOF modelling for an overdeepening/future lake to give downstream flow heights, velocities etc. Previous studies have identified overdeepenings and downstream infrastructure that are within a GIS routed flow trajectory. However, to avoid a discussion around what is or is not considered GLOF modelling, we will remove the term "first time".

- L436: What is 'complex' in this topography?

Revised to *"steep, mountain topography"*

- L444-447, and again L455ff: The artificial drainage of Jialongco somehow undermines parts of this paper, given that the simulations use the lake volume before the drainage, right? It would have been good to show how simulated flood magnitudes (flow depths, extent of the inundation, etc.) change because of the reduced flood volume, and if this in turn changes GLOF risk. In essence, if the probability of a given flood magnitude decreases ( = hazard), then risk must also decrease, assuming constant values of exposure and vulnerability. It thus remains unclear why this insight has been held out from the introduction and all subsequent analysis.

We agree, and learned only of the artificial drainage late in the process of working on the first draft of this paper (highlighting the disconnect between local authorities and scientists in this region). For the revised manuscript, we now include two simulations for Jialongco – a) for the original natural state of the lake (with bathymetry from 2019 and DEM from 2018) and b) for the lowered state (with bathymetry and DEM from 2021). Results from both simulations are included in Table 2, and Figure 3.

- L458: 'had only a minimal effect on the overall lake size': Unclear conclusion with regards to Fig. 9.

Now that we have measured bathymetry for this lake both before and after lake lowering, we provide a calculation of how much the volume has been reduced (Table 3). The sentence has been revised accordingly.

- L463: 'would reduce the threat to these buildings': contradicts to what is written some sentences above?

The sentence has been revised according to the results from the new scenario with the reduced lake volume that shows the lowering had only a minimal effect on the on the potential downstream flood magnitudes resulting from a catastrophic ice avalanche into the lake. However, it is expected that the lake lowering would have had an impact on the reducing the risk from much smaller events.

- L470-483: largely repetitive from the Results, consider substantial shortening or deleting.
Repetition has been removed.

- L495-499: Content of these sentences unclear, even if formulated 'in other words'. Please rephrase or shorten.
The sentence has been revised and expanded based on the new modelling results.

- L501: When talking about early warning, why not using the calculated flood arrival times to provide a solid basis for discussion? What do the authors suggest to implement in such an early warning system? How can people response to the warning?
This part of the discussion has been rewritten as suggested, based on the flood arrival times modelled with r.avaflow. We highlight that EWS on its own will offer minimal protection against a very large outburst scenario – one of the key messages of the revised paper.

- L503-504: 'under the philosophy of preparing for the worst, while hoping for the best': please avoid jargon.
Agree and revised.

- L504: 'complex transboundary regions': what is complex here and how can this study help to understand this complexity?
Wording of "complex" has been removed. The sentence is intended to simply highlight why warning times are so critical in a transboundary region, because we know from past experiences that communication between national authorities can lead to delays in alerting communities.

- L523-525: Not sure whether the increase in GLOF risk has been quantified in this study?
Fair point. We have revised to *"increased GLOF exposure"* which is justified based on Figure 6.

- L528-529: 'Hard engineering strategies that address only the hazard source are a socially and environmentally less desirable option': Not sure whether I can agree with this statement: If Hazard = 0
We want to say here that hazard will never = 0 in such a context, unless you drain every lake completely, and do so for every new lake that develops. In addition, even if you arguably could reduce GLOF hazard to zero, the community remains vulnerable and exposed to other geohazards, including Chamoli-style mass movements. Hence, a more comprehensive approach, involving EWS, affective land use zoning, and related social interventions, are far more desirable in almost all contexts in our view. This does not mean that remedial measures do not have a place within such a comprehensive approach. In view of the reviewers concern, we will revise the wording to: *"Relying only on hard engineering strategies at the lake source will prove insufficient, as such strategies do not address underlying exposure and vulnerability to GLOFS and other geohazards, and are demonstrated to be ineffective in the face of worst-case, catastrophic outburst events"*

**Reviewer 2**

**General Comments:**

1) The overall framework and structure – in the current version of this study, the authors first do the 'worst case' modelling and then search for possible GLOF triggers to justify modelled results (which is actually done not very convincingly when admitting that modelled GLOFs would need very unlikely occurrence of high magnitude (X0 Mm3) ice-rock avalanche into the lake as a trigger); logical framework would start with: (i) search for possible / likely GLOF triggers for existing lakes, (ii) feeding them into definition of outburst parameters and scenarios, and (iii) leading to GLOF modelling + (iv) future lake and GLOF. I suggest to consider re-structuring the manuscript accordingly

We thank the reviewer for his careful and comprehensive comments, and apologise for the time taken to respond and revise. We agree that the logic of the paper structure was not ideal or in line with best practice. Both the methods and results have been heavily restructured to first address lake susceptibility (incl. triggering), before proceeding to GLOF modelling, and finally consideration of the future lake. The fundamental methodological change we have made, based on comments from all reviewers, is to now use r.avaflow to simulate the GLOF process chains, thereby directly linking the triggering processes (large ice or ice/rock avalanche) to the outburst events.

2) Uncertainties in input data: as the future is uncertain, I'm quite reluctant to using any single value 'worst case scenario' concept and I call for using a range of values (and scenarios) instead. Below I comment on (some of the) major sources of uncertainties which are cumulating throughout the process and are not properly treated:

We understand the reviewers concern, and ourselves are involved in several studies where the goal necessitates a full range of scenarios are modelled. However, in this paper we are aiming to address what we see as a gap in approaches/scales, fitting between large scale first-order approaches and scenario-based hazard mapping, considering both current and potential future threats, and implications for DRR planning. Is a given lake really going to present an unprecedented threat to downstream areas (relative to existing lakes)? Does such a potential lake need to be considered in the design of response strategies such as EWS? Such questions are not answered by existing studies, which to our knowledge, have not gone beyond first-order assessment of future lakes (e.g. GIS-based approaches). Likewise, we believe these questions can be answered without going to the level of detailed scenario-based hazard modelling. On a practical level, we see this gap as an important niche to fill. There has now been several years' worth of first-order modelling studies showing that glacial lake area and number will increase in the future, yet in our interactions with authorities across several countries, we see no indication yet that these future threats are being considered in DRR planning. While it would be academically possible to generate detailed scenario-based hazard maps for a potential future lake, we consider a more practical intermediate step is needed to first demonstrate that a given future lake warrants such attention and further monitoring.

We therefore argue that there is merit in simulating single worst-case scenarios, as these can be a first basis for planning response strategies that remain robust under an uncertain future. **In fact, for EWS design, in our experience worst-case scenarios are the most important consideration, yet have often been neglected in comparable studies**. For example, based on our results, response actions at the border with Nepal can be planned considering a minimum warning time of 30 minutes, compared to a time of 120 minutes simulated for more gradual breach processes (Zhang et al. 2021). We therefore maintain the focus in the study on worst-case scenarios, but have revised the way in which these scenarios are developed. The revised approach focuses on defining an upper limit to an expected ice/rock avalanche starting volume, and uses r.avaflow to simulate the resulting GLOF cascade.

a) the essential value at the very beginning is the estimation of breach depth (in this study referred as breach height hb). The authors provide neither details on how this
value is estimated nor what the uncertainty of this estimation is; another issue is
whether flat (<7° (rough Google-Earth-based measurement)) and pretty wide (> 450
m) moraine dam (e.g. Galong co) could ever be breached; and if it is breached, the
crucial question is how deep (longitudinal profile of the breach is typically far from flat –
I mean, if you have a vertical difference between the lake level and the toe of moraine
dam 40 m (this is how you define breach depth, right?), lake level decrease in case of
breach will be less than that (it is not going to be breached to 0° slope), depending on
longitudinal width of dam body; this is actually seen in Fig. 2b: if you define breach
depth in this way, you should not use the same value in calculating released volume,
because it differs to the lake level decrease (and in turn it leads to substantial overestimation of
released volume))

We agree that these parameters and assumptions on breach depth were not well substantiated. The
revised modelling approach using r.avaflow has reduced the need for these assumptions around the
dam breaching and/or overtopping mechanisms. The erodible area of the dam is now based on
careful consideration of the dam characteristics including any possible ground ice (relevant for
Galongco only). In addition, we've since obtained bathymetry measurements from both Jialongco
(before and after lowering) and Galongco that have been used for the modelling.

b) in the next step, the authors use this pretty uncertain value to estimate released
volume (which is not correct in my opinion, see above) and breach parameters, using
Froehlich (1995) empirical relationships; but it is important to realize that: (i) Froehlich
(1995) is based on compiled information of man-made earthen dam failures, not
natural dams; (ii) failure mechanism of most of these cases in the database was piping,
not overtopping; (iii) released volumes in the dataset was mainly <1Mm3; and >100
Mm3 in only two cases (Oros, Teton); with expected released volumes 25, 70 and 262
Mm3, you are extrapolating far beyond observed data of Froehlich (1995) and the
uncertainty is unknown (Froehlich, 2004 should be checked).

Based on this, and other comments, we no longer use the Froehlich approach. The GLOF process
chains has been simulated using r.avaflow, with breach parameters dynamically calculated.
Important model parameters (e.g. internal and basal friction angles of the solid material, the fluid
friction number, and the coefficient of erosion) have been empirically defined, with references
provided.

3) Timing - Using Eq. 4, calculated time for breach formation of Galong co is 153 min, but
you expect peak discharge in Nyalam in 82 min -> please explain what times are you
referring to (82 min from breach initiation, from peak discharge at the dam (when from
breach initiation?) or from development of the breach?); being as clear as possible is
especially important when talking about EWSs, presenting hydrographs at the dam would
be beneficial.

A new figure (Fig 4.a) has been included giving the hydrographs at the dam – thank you for this
suggestion! For clarity, all timing is relative to the start of the process chain (initiation of the ice/rock
avalanche).

4) GLOF likelihood – this section gives some largely general statements and qualitative RS based
observations and looks more like a discussion rather than result to me; Tab. 3
summarises first order GLOF susceptibility factors, but this study is not a first order
assessment – it is a detailed study of two existing and one potential future lake; what is

shown in Tab. 3 is perhaps true for most of the lakes in the region (warming climate, steep slopes and crevassed glaciers upstream, …) and leaves the question of GLOF likelihood open; the use of >30° threshold for initiation of mass movements seems too simplifying and not really helpful for the scale you are working on

In line with other reviewer comments, this section has been renamed to "4.1 lake susceptibility and scenario development" recognising that it is outside the scope of this study, and not feasible, to provide quantitative likelihoods linked to specific scenarios. Table 3 has been revised, and where possible, quantitative details have been provided. The slope threshold-based avalanche calculations have been removed, and replaced in the text with a more detailed assessment of local geological and glacial conditions. A model-based estimate of ground temperature (Obu et al. 2019) is now included to give an improved indication of likely permafrost conditions. It's true that some statements are general and could be applied to other lakes and slopes in the region. Nonetheless, we prefer to keep the table in full, even if some entries are general, because this is the first study to directly utilise the comprehensive "check-list" table of susceptibility factors coming out of the GAPHAZ international guidance document, so an important opportunity to demonstrate both the applicability and limitations of the GAPHAZ approach.

We would note that there are few, if any, examples of quantitative likelihoods being applied in the hazard assessment of moraine dammed outburst floods, and related scenario-based studies have typically applied qualitative high, medium and low likelihoods to large, medium and small magnitude events respectively.

5) Practical implications – the authors mention the importance of such studies for local authorities, which is in principal true and also a rationale of many similar studies. My experience is that practical utilization, however, often lacks behind. As documented by the authors, local authorities meanwhile started remedial works by themselves, meaning that they have some kind of GLOF hazard assessments and management procedures in hands. I expect these documents may not be publicly available, but attempting getting in touch with authorities in charge of these measures would be highly appreciated (and could also help to bridge the gap between what scientists and authorities are doing).

This is a great comment on the practical implications of this project. We have struggled to make progress here throughout the 3 years of the project, and despite several repeat visits in the field, we've been unable to enter into exchange with the local authorities and learn details from them about what is planned in terms of further DRR measures. In Tibet, and close to a military controlled border region, exchange with authorities is even more challenging than in other regions we have worked. It seems almost everything is explained as being "for military purposes". Hence, we are not sure to what extent these measures implemented so far are underpinned by comprehensive GLOF modelling. A desire to bridge this gap between scientists and authorities is exactly why we focussed this study on the methodological space between first-order assessment approaches, and detailed scenario-based hazard assessment. In our view, it is within this space that we can most reasonably demonstrate the importance of considering worst-case scenarios and future lakes as part of DRR planning. Going to the level of comprehensive scenario-based GLOF modelling and hazard mapping only makes sense in our view once there is buy-in and exchange with local authorities, particularly with regards to the threat of a future lake. Otherwise it remains an academic exercise only.

**Specific Comments:**

L19-20: please comment on what can be done to reach this ambitious aim (not a part of the study)

We agree that our study rather does not feed directly into "decision making" and have removed reference to this in the abstract. Nonetheless, as described in the responses above, we do believe our study provides fundamental insights for future-proofing DRR planning. For example, highlighting the minimal warning times that will occur under worst-case outburst scenarios. Anecdotally we know there are plans from international donors to fund the design and implementation of a transboundary EWS in this basin, so such messages are important to feed into the scoping of such projects.

L40: high magnitude
As per comments from other reviewers, we have avoided subjective wording here, and rather refer to the reported GLOF frequency of 1.3 per year for the Himalaya since the 1980s (after Vey et al 2019).

L52: I would not call 17 GLOFs overt the Tibet since 1935 'particularly common'
Agree – sloppy wording. We'll revise factually to state "At least 17 GLOF disasters (causing loss of life or infrastructure) have been documented in Tibet since 1935, mostly….."

L68: these numbers are confusing; you mentioned 3-fold increase, does it mean that future doubling in border areas of China – Nepal is thus below average?
Agree – taken out of context of the original paper these numbers are confusing and cannot be directly compared. To avoid confusion we have removed the reference to a 3-fold increase in risk, and focus just on the doubling of transboundary lakes, which is the most relevant information for the present study.

L82-85: not sure this is met
We agree that our study rather does not feed directly into "decision making" and we will remove reference to this. Nonetheless, as described in the responses above, we do believe our study provides fundamental insights for future-proofing disaster response planning.

L92: please consider adding description of 2(3) studied lakes in this section
Thanks for this suggestion. We considered to add a physical description of the 3 lakes here, but, we feel it is better placed in section 4.1 as part of the assessment of lake susceptibility.

Fig. 1: please consider adding topography info; there are many dangerous lakes in the region – the authors are asked to justify why they focus on these two existing and one potential future lake (while there are other lakes forming currently)
At this scale, topographic information is difficult to view, and we believe sufficient topographic information comes later in the results. The justification for the focus on Jialongco and Galongco is now expanded with reference to our early study (Allen et al. 2019), and a new study from Zhang et al (2021). The description on how and why the particular potential future lake was identified as a priority for worst-case analyses in the study is provided in section 3.1 (large size, proximity to village, evidence of supraglacial lakes already forming etc).

L164: methodology of obtaining hb is not clear
Under the new modelling approach, hb is no longer estimated in this way, but has been dynamically calculated as part of the r.avaflow simulation.

L178: what breach scenarios?

Apologies, this should have referred simply to the 3 worst-case scenarios. Text has been revised.

L189: this value needs justification
Manning's n was not required for the updated modelling with r.avaflow. Key model parameters (including the internal and basal friction angles of the solid material, the fluid friction number, and the coefficient of erosion) have been empirically defined, with references added (see section 3.2).

L210-214: this approach seems too rough for detailed case study like this one
Agree – the approach to assessing ice/rock avalanche susceptibility has been revised, and is based on detailed consideration of local geological and glaciological conditions. See section 4.1 for this assessment.

L247: considering uncertainties behind a single-number result, I found a range of values highly desirable
See response to general comment (2)

L252 ms-1 when talking about velocities (please check throughout the manuscript)
Thank you – has been revised to 'discharge' throughout.

L274-297: this is contradicting; on the one hand you expect 48 m deep breach of very flat moraine dam and on the other hand you find erosion unlikely?
Wording has been clarified. The lack of erosion potential is referring to within the river channels downstream of the lake, where channel slopes are relatively gentle. Under the new approach to modelling with r.avaflow, the moraine dams are set as entrainment zones where the erosion depths are dynamically simulated.

Table 1: please specify timings (see my general comments); Jialongco – are these values of the lake before or after the remediation?
Timings have been revised based on the new simulations, and are now reported consistently relative to initiation of the avalanche.  For Jialongco, we now include an additional simulation for the reduced lake area/volume and altered dam geometry (Jialongco post-lowering JC-L).

L300: consider moving to discussion (see my general comments)
See response to general comment (4). This section, now called "Lake susceptibility and scenario development' has been significantly revised. We have added quantitative detail to the extent possible. This goes far beyond discussion in our view, even if some details will remain qualitative.

L319: this is not very well-argued (most of the glacial lakes are surrounded by glacierized slopes with >30°)
The slope is not the only factor here, but rather the classical ramp-like topography of the glacier tongue, heavy crevassing, and likely temperate bed. We will revise the text to place more emphasis on these characteristics, including evidence of past instabilities of this tongue.

L326-343: yes, large volume ice-rock avalanches are rare and in the seismically active regions, you can't rule out the possibility of hitting the lake – you can say this about most of the lakes in the region (and most of the high mountain lakes globally); I'm wondering whether is there any site-specific implication for GLOF likelihood?
One key factor here is the size of the mountain headwalls immediately surrounding the lake (i.e., 3000 metre high slope of  Shishapangma). Another factor is the potential peak ground acceleration linked to seismic activity (we have updated table 2 to include this information), which varies across the Himalaya and is enhanced in such steep topography. We have also revised the text to include more detail on the structural geology of the surrounding slopes and permafrost conditions. Of

course, we will not get to a quantified likelihood level, but this revised text provides stronger basis for our qualitative assessment.

L346: Klimes et al. actually showed that landslides in moraines are not capable of producing any large GLOF from Lake Palcacocha
Apologies – this citation has been removed.

Tab. 4: what is freeboard to height ratio? Both existing studied lakes seem to have surface outflow (freeboard = 0m); catchment stream density / order seem odd for evaluating GLOF likelihood; you also report no evidence of historical instabilities, further questioning the likelihood of such events for triggering GLOFs
This was an error, and should be simply "freeboard". As has been done is many other studies, we measured this as the remaining freeboard (i.e., from a hypothetical line across the crest of the dam), irrespective of the fact there is surface outflow. The logic being that the remaining freeboard area still offers some protection in the case of an overtopping wave. We have added explanation to the table with this detail. Stream density/order is related to the potential for rainfall/snowmelt triggering of a GLOF (after Allen et al. 2016 paper on Kedarnath). All factors come from the GAPHAZ guidance document where their relevance, and supporting evidence is well cited. We agree that details on historical instabilities ere needed, and have add this evidence (and references) to a revised version of the table.

Tab. 4: again, estimating possible ice avalanche starting zones with precision to 1 m2 is not appropriate considering apparent uncertainties; better use a range of values
Agree the precision is unjustified. This table and section has been revised to focus on the worst-case scenarios affecting each lake (see Section 3.1) , and includes an estimated ratio if ice to rock, as required for input to r.avaflow.

Fig. 6: if intensities are based on flow depth only, why not to use flow depth directly?
Fair comment – our rationale is that intensities are more generic, and allows readers and authorities to compare with other approaches and estimates, including for other hazard processes. Flow depths are giving in Figures 3 and Table 3.

L418-418: please comment on a difference between values estimated here and size of the future lake considered in Tab. 1?
The maximum anticipated lake size anticipated from Glabtop (1.54 km$^2$) is now added to the text and included again in the caption.  This compares to estimated size of 1.33 km$^2$ by 2100 under scenario 2 in figure 8. We note in the discussion section that the estimate at 2100 does not account for ice calving feedbacks, so is expected to underestimate actual size.

L433: there is not much about management planning in the study
Wording revised.

L435-436: the authors published several studies on GLOF from potential future lakes previously
Yes that is true, but this is the first time a study has undertaken any sort of dynamic GLOF modelling for an overdeepening/future lake to give downstream flow heights, velocities etc. Previous studies have identified overdeepenings and downstream infrastructure that are within a GIS routed flow trajectory. However, to avoid a discussion around what is or is not considered GLOF modelling, we have removed the term "first time".

L446: the greatest immediate threat from 2 existing studied lakes

We are confident, given the justification that has been added for selecting these 2 lakes, that they are the 2 greatest immediate threats to Nyalam.

L450-454: this is not suggesting any lower limit, this is estimated potential loss for given scenario; please re-word this sentence
Sentence has been revised.

L456-458: maybe the remediation is still in progress?
As indicated in the response to the general comment, it has not been possible to get information on this from authorities. Since the first submission of the manuscript, we had another field visit to the region and the engineering works are now in a final state (see new image in Figure 9).

L465: EWS can help to save lives, but not the immovable property (which may already be there); if the value of potentially affected immovable property is >> than the cost of remedial works, then it makes sense also to remediate the lake(s)
Fair point and we were too dismissive of remedial measures in the initial draft, although with the inclusion now of the lowered lake scenario it is clear that this has little effect on a very large GLOF event. Both in the discussion and in the conclusions, wording has been revised to make it clear that a comprehensive solution is recommended, which includes (but is not limited to) remedial works at the lake.

L508: no clear conclusion on GLOF likelihood is given
While we do not provide a quantitative assessment of likelihood, we provide clear conclusions on the relative likelihood of an outburst from the lakes, concluding that Jialongco (based on consideration of all susceptibility factors) has the highest likelihood of generating a large outburst event.

L515: details about the project (planned final stage) should be presented (maybe the plan is to drain the lake much more?)
Since the first submission of the manuscript, we had another field visit to the region and the engineering works are now in a final state (see new image in Figure 9), and we have been able to quantify the level of lowering achieved, and the effect on a worst-case GLOF event.

L519-519: this is general qualitative statement which is true for many lakes in the region (not very helpful for DRR authorities I guess)
This conclusions have been revised based on the new results.

L529: why are they socially less desirable? And why environmentally less desirable (GLOF is a major disturbance to the valley ecosystem)?
In view of concerns raised by multiple reviewers we will remove the reference to social and environmental sustainability as this may be subjective (depending on how local people view glacial lakes). The wording has been revised to: "Relying only on hard engineering strategies at the lake source will prove insufficient, as such strategies do not address underlying exposure and vulnerability to GLOFS and other geohazards, and are demonstrated to be ineffective in the face of worst-case, catastrophic outburst events"

**Fabian Walter**

**General Comment:**

In their submitted manuscript, Allen et al. propose a hazard assessment around two existing and one potentially forming proglacial lake in the Puiqu River basin, Himalaya. They document the glacial environment as well as its projected development under future climate conditions, potential outburst flood triggers and modeled discharges in vulnerable downstream communities. The study gives insights into different aspects of glacial lake outburst floods highlighting particularly interesting features of the investigated cases. However, it is difficult to grasp how relevant the scientific insights from this investigation are given that the authors' judgement is too often limited to qualitative assessment. For a scientific journal submission, I was expecting more substance in view of reproducibility and representativeness of the results

We thank the colleague for his critical feedback on the manuscript and apologise for the time taken to respond and revise. We designed this study to address what we see as a gap in common GLOF assessment scales (particularly in the developing world), fitting between large scale first-order approaches and comprehensive scenario-based hazard mapping. We also recognise that comparable studies have rarely considered worst-case scenarios involving very large avalanche triggered GLOF process chains. We thereby aim to provide an illustrated case-study showing how worst-case scenarios, including from a potential new lake can begin to be considered in DRR planning. Is a potential future lake really going to present an unprecedented threat to downstream areas as is commonly implied (relative to existing lakes)? Does such a potential lake need to be considered in the design of response strategies such as EWS? Such questions are not answered by existing studies, which to our knowledge, have not gone beyond first-order assessment of future lakes.

The methodological approach was designed to be as simple as possible, with a view that methods could be attractive for upscaling to any number of current or potential future lakes. However, in view of the comments from all 3 reviewers, we accept that this was scientifically weak, as many assumptions were not well substantiated, and importantly, the susceptibility assessment and triggering processes were not directly linked to the outburst scenarios. Therefore, in the revised manuscript we now start with a more systematic assessment of lake susceptibility, to derive worst-case scenarios of ice-rock avalanche triggered GLOFs from all 3 lakes, including a new scenario for the artificially lowered lake volume of Jialongco. The use of r.avaflow for the modelling of the entire process chain reduces the amount of qualitative assumptions regarding the link between the trigger event and outburst flood, as the entire process chain is dynamically simulated. We'd also note that we've since been able to measure bathymetry for the two current lakes, which is important input that was previously preventing us from using r.avaflow.

We must however acknowledge that many aspects of the lake susceptibility assessment remain partially qualitative, as for some parameters like permafrost conditions, we simply lack high resolution modelling results for this area. Likewise we would note that there are few, if any, examples of quantitative likelihoods being applied in the hazard assessment of moraine dammed outburst floods, and related studies have typically applied qualitative high, medium and low likelihoods to large, medium and small magnitude events respectively. To better reflect the scope of our study, and acknowledge that the assessment in our study is rather limited to a large worst-case scenario (and therefore can't be considered a full hazard assessment), we have modified the title to *"Glacial lake outburst flood hazard under current and future conditions: first insights from worst-case scenarios in a transboundary Himalayan basin"*.

The abstract has been further modified to make it clear the focus is on the timing and magnitude of simulated worst-case outburst events

**Major Comments:**

My main point of criticism is that the reader of this manuscript is left with little information on how to assess validity or accuracy of the findings. In its current state, the study appears more like a presentation of important facts and qualitative judgements, which are typical for

technical reports. For a scientific paper I would have expected some critical assessment of the flood risk, e.g., a benchmarking of the presented methods against previous occurrences of outburst floods. The authors cite accounts of previous outbursts in the area (Line 106). Could they be used for this?

Unfortunately there are very scarce details on the past GLOF events occurring in the upper basin (affecting village of Nyalam), beyond the fact that they occurred (and even that information is dubious). The 2015 earthquake resulted in a lot of erosion around the village, so geomorphological evidence from past events is also not clearly recognisable. However, further downstream towards the border with Nepal, an outburst from 2016 has been well documented (e.g. Cook et al. 2018) and we do compare in the discussion our worst-case scenarios against this event. Likewise, we also compare our results against those obtained in other model-based studies within the same basin (e.g., Shrestha et al. 2010, Zhang et al. 2021)

Cook, K. L., Andermann, C., Gimbert, F., Adhikari, B. R. and Hovius, N.: Glacial lake outburst floods as drivers of fluvial erosion in the Himalaya., Science, 362(6410), 53–57, doi:10.1126/science.aat4981, 2018.

Shrestha AB, Eriksson M, Mool P, Ghimire P, Mishra B, Khanal NR. 2010. Glacial lake outburst flood risk assessment of Sun Koshi basin, Nepal. Geomatics, Natural Hazards and Risk. Taylor & Francis 1(2): 157–169. DOI: 10.1080/19475701003668968.

Zhang, T.; Wang, W.; Gao, T.; An, B. Simulation and Assessment of Future Glacial Lake Outburst Floods in the Poiqu River Basin, Central Himalayas. Water 2021, 13, 1376. https://doi.org/10.3390/w13101376

The first part of the paper presents some motivation on why to study the chosen three lake basins. However, it is not possible to verify that this corresponds to a worst-case analysis. In this case, it would be necessary to show that no lake basins could produce more serious outburst floods. As the authors argue, this depends on moraine dam geometry, water volume and trigger potential. Under these aspects it cannot be argued that the presented set of lakes is representative for worst case scenarios.

Thanks for this comment. We struggled on this aspect as we did not want to reproduce results from our earlier study (Allen et al. 2019), that was primarily the basis for focussing on the two current lakes. In view of this, and comments from the other two reviewers, we now refer explicitly to the danger levels that were assessed for these lakes in this previous study. Details on dam geometry and other characteristics of the lakes come in section 4.1. The description on how and why the particular potential future lake was identified as a priority for worst-case analyses in the study is provided in section 3.1 (large size, proximity to village, evidence of supraglacial lakes already forming etc).

At too many parts of the manuscript, the authors' qualitative judgement is presented as a scientific result. In particular, in the Section "GLOF likelihood", various factors influencing or triggering outburst floods are presented, but I could hardly find any objective arguments. It seems that the only one is the estimate of a dam-overtopping wave volume, which can be 10 times as high as the "incoming mass". Here and elsewhere in the manuscript, it has to be made clear that the conclusions are based on solid scientific grounds. Otherwise, a "low probability" could indicate one catastrophic event every 5 years as opposed to several ones per year. This is not what the authors imply. In a similar sense, it is not clear what the demanded "comprehensive and forward-looking approach to disaster risk reduction" is. To me, such an approach should always be taken and I see little connection to the present study or any finding, which made the suggested strategy particularly pertinent to the Poiqu River basin compared to any other place in the world.

Effort has been made throughout the revised manuscript to provide a more quantitative basis for the assessment findings. The section on GLOF likelihood (renamed to "lake susceptibility") has been redrafted, with more focus on the local geological and glaciological conditions that determine large ice/rock avalanche potential. Table 3 (now 2) has been revised, and where possible, quantitative details provided with references cited. For example, the slope threshold-based avalanche calculations have been removed, and replaced with a more detailed assessment of local geological and glacial conditions. Again, we must acknowledge that many aspects of the lake susceptibility assessment remain partially qualitative and subjective for those factors that lack quantitative data or can be inferred only. However, we will provide further references, that support the expert judgements being made.

On the final point, we are yet to see GLOF risk reduction strategies anywhere in the developing world that can be considered forward-looking. There has now been several years' worth of first-order modelling studies showing that glacial lake area and number will increase in the future, yet in our interactions with authorities across several countries, we see no indication yet that these future threats are being considered in DRR planning. We fully agree that DRR planning ultimately requires more quantitative assessment of outburst probabilities etc. However, this remains the holy grail for the assessment of existing moraine dammed lakes, let alone future lakes for which conditions of both the lake and surroundings are highly uncertain. Particularly for DRR strategies such as an EWS, we would therefore argue that there is merit in our approach of simulating single worst-case scenarios, as these can be a first basis for planning response strategies that remain robust under an uncertain future. For example, by planning response actions at the border with Nepal around a warning time of 120 minutes (Zhang et al. 2021) vs. 30 minutes (our study), or ensuring critical infrastructure in Nyalam is located well away from a worst-case or future GLOF path.

**Specific Comments:**

I suggest including a cartoon explaining different lake formation scenarios and specifically the hydrological base line. To be honest, I had to stare for some time at Figure 3 of Benn et al. (2012) to understand this concept. On the other hand, I never grasped the meaning of the "the lowest point where the glacier surface intersects the terminal moraine" (it seems that by definition, the glacier and the terminal moraine should not intersect). Similarly, when the future evolution of the lakes is described, longitudinal profiles would be extremely helpful. This would help the reader to understand Figures 7 and 8 and appreciate the shown information.

We thank the reviewer for their suggestions about how to improve the description of the concepts we employ to examine the likelihood of future lake formation on glacier RGI60-15.09475. To ensure that the approach we have followed is as clear as possible, we have improved the description of the lake formation scenarios in section 3.3 (Future lake development) and clearly refer to Figures 3 and 19 in Benn et al. (2012), as these two figures already provide a good visual representation of how supraglacial meltwater ponding can lead to glacial lake formation. We are clear to distinguish between the initial stages of supraglacial meltwater ponding and lake formation (when the terminus of the glacier is still in contact with the terminal moraine) and the later stages of lake expansion (when, as the reviewer suggests, the terminus of the glacier would not be in contact with the terminal moraine).

We also thank the reviewer for their suggested improvements to Figures 7 and 8. Figure 7 already incorporates longitudinal profiles of glacier surface elevation, surface slope and surface elevation change, but we agree that Figure 8 benefits now from the addition of centreline profiles of glacier surface elevation under different scenarios of thinning in 2050, 2075 and 2100.

The flood model is a key ingredient to this investigation. Although I agree that too many technical and mathematical details are not appropriate for this study, I was wondering what the main parameters and boundary conditions of this model are. Apparently, the flood volume, some time scale of drainage initiation and dam geometry play a role and it would be interesting to hear how these parameters drive the model.

In view of the comments from all 3 reviewers, we accept that the previously used modelling approach was scientifically weak, as the triggering processes were not linked to the outburst scenario, leading to reliance on assumptions (e.g., concerning Potential Flood Volume) and empirical relationships to define parameters. Therefore, in the revised manuscript we use r.avaflow to simulate worst-case ice-rock avalanche triggered GLOF process chains from all 3 lakes, including a new scenario for the artificially lowered lake volume of Jialongco. This has the advantage of the outburst process (and related parameters) being dynamically simulated based on the ice-rock avalanche triggering event. Key model parameters (including the internal and basal friction angles of the solid material, the fluid friction number, and the coefficient of erosion) remain empirically defined, with references added.

Lines 45-47: "… most scientific attention has focused upon …" I do not agree with this

statement. In the jökulhlaup literature, ice-marginal and subglacial lake drainages have also received a lot of attention. Whereas I cannot say which scenario has been most prominent, I would refrain from an absolute statement on scientific attention.

Good point – we now qualify this statement as *"In Asia, most scientific attention has focused upon…."*

Lines 175-176: "B_w and h_b are fully obtained" measured?

Under the revised modelling approach with r.avaflow, these values have been dynamically simulated.

Line 178: Reference for HEC-RAS is needed.

Section has been revised, based on modelling approach with r.avaflow.

Line 185: Reference for DEM's is needed.

Section revised.

Line 189: Reference for Manning roughness value is needed.

Manning's n was not required for the updated modelling with r.avaflow. Key model parameters (including the internal and basal friction angles of the solid material, the fluid friction number, and the coefficient of erosion) have been empirically defined, with references added (see section 3.2).

3.2 Lake susceptibility assessment: Presenting the likelihood calculations seems appropriate here.

Table 3 has been  revised and to the extent possible, more quantitative assessment of the various GLOF susceptibility factors has been provided. We note, as in the general comments, that some qualitative expert judgment is inevitable in this field, particularly for likelihood estimates, as outlined in International Guidelines to Glacier Hazard Assessment (https://www.gaphaz.org/files/Assessment_Glacier_Permafrost_Hazards_Mountain_Regions.pdf).

Line 231: "considering the factors outlined by …" these factors should be specified.

Text has been amended here to reiterate that the surface slope and velocity of the glacier are both suitably low to allow for substantial meltwater ponding, as was already stated on lines 220 and 221 of original draft.

Line 257: I suggest a paragraph break here.

Section has been heavily revised

Line 457: "recent removal of much of the frontal moraine …" this needs a reference.

I reference to Figure 9 is now provided.

Lines 466: What are "capacity building programs"?

Has been revised to "programs to strengthen local response capacities"

**Figures:**

Figure 1: The lakes at Cirenmaco and Jialongco are difficult to discern.

Its not possible at this scale to see the outline of these lakes but the location shown by the red points is clear. This overview figure is simply to give the reader an impression of the region and the location of the different lakes mentioned in the text.

Figure 2: The font sizes are a bit small, but h_b defined in Panel A seems to disagree with Panel B.

Figure removed.

Figure 4A: Where is the lake? The blue outline or the light blue polygon? A different color scale for maximum flow height would help.

Figure revised (now Figure 5).

Figure 5: It is difficult to tell where the lakes are. I do not see any blue patches.

Figure revised (now Figure 2).

Figure 6: The two images in Panels C need some annotation. What does the reader see in these images? Why is one so dark and the other one bright? What happened between the two?
Annotation has been added and caption has been expanded for panel C. The significant expansion of infrastructure seen between 2015 and 2017 should be obvious from these panels, despite the differences in colouring from the two satellite images.

Figure 8: This is the future lake site, right? Where will the lake form? The colored outlines make little sense and are hard to sea. Which direction does the ice flow?
Red line is the maximum projected overdeepening, and the blue/purple colourings are extent to which the lake is projected to emerge under the different future scenarios. We have improved the colouring and information in the caption. Direction of ice flow has been added.

Figure 9: Some arrows and annotation as well as scale bars are needed.
Caption has been expanded and an additional image from 2021 has been added where the engineering works are much more visible.

---

## Author Response (AR2)

**Department of Geography**

Dr. Simon Allen
University of Zurich
Department of Geography
Winterthurerstrasse 190
CH-8057 Zurich

simon.allen@geo.uzh.ch

The Editor
Natural Hazards and Earth System Sciences

Zurich, 30 September 2022

**REVISED MANUSCRIPT FOR CONSIDERATION**

Dear Dr. Bühler

On behalf of my co-authors, let me thank you and the reviewer (Adam Emmer) for your prompt and positive handling of our revised manuscript:

*"Glacial lake outburst flood hazard under current and future conditions: worst-case scenarios in a transboundary Himalayan basin"*

We have undertaken the requested minor revisions. Below you will find our point-by-point final response to these comments, including on the issue of the likelihood and implications of the very high magnitude events we model here. On this point we have added further text to the discussion.

Please note, we have also edited the references to align to the journal style.

We believe this paper will generate lively discussion in the research community regarding large worst-case process chains involving glacial lakes.

Yours Sincerely,

Dr. Simon Allen

**Referee #2: Adam Emmer**

*I have reviewed the previous version of this study in which I criticized the worst-case scenario approach. I thank the authors for substantial revisions they made and for addressing many of my comments. I have no concerns regarding the technical aspects of modelling which is sound; however, I'm still struggling with the worst-case scenario approach (scenario development) and find the modelled results detached from what has been observed in HMA in human-relevant temporal context. Strikingly, Veh et al. (2020; https://www.pnas.org/doi/10.1073/pnas.1914898117) estimated 100-y mean GLOF discharge 15,600 m3/s (compare to yours > 500,000 m3/s (!)) roughly corresponding to the peak discharge of the largest reported GLOF in the region (15,920 m3/s from Zhangzhanbo Lake). Indeed, extremely large mass movements occur in mountains (e.g. Chamoli with comparable volume to your scenarios), but should that be taken as a 'golden standard' for executing GLOF / process chain studies? In the model world, should we let 20Mm3 collapse to all GLOF-susceptible lakes in the HMA?*

*Considering indisputable interest for and potential utilization of this study by disaster risk reduction practitioners, I wonder what would be the return period of > 500,000 m3/s flood that is presented here? It is mentioned in discussion that it is probably > 200 years. Yes, perhaps much more. Is it still relevant for anyone or is the probability of such event far below the threshold of acceptable risk? Moreover, the authors basically conclude that hazard mitigation measures implemented at Jialongco are useless (and even counterproductive). Such statements might demotivate anyway difficult-to-implement (and expensive) hazard mitigation works and should be communicated carefully and put into broader context (for instance, their effectiveness for more likely lower magnitude triggers when it can prevent dam breaching for instance). For these particular reasons, I always call for (and Ashim knows that well from our previous collaboration 😊) considering a range of scenarios (ideally associated with probabilities or return periods) instead of the worst-case scenario.*

*I don't expect you to revise your study in line with my comments above, but I would be happy if you would consider it in your future studies.*

We sincerely thank Adam Emmer for his valuable insights, as always. We absolutely agree that communication around the likelihood and implications of a worst-case scenario for disaster risk management are critical. In fact, one of the main motivations with this work is to initiate discussion in the research community around worst-case scenarios, which until now, are basically neglected (with only small, medium and large scenarios typically modeled). We are definitely not saying that such scenarios need to be the "golden standard" for GLOF process chain studies, but merely that they need to be at least considered. This point was already made in the GAPHAZ international guidance from 2017, but, ironically without actually providing any guidance on how to establish and model such worst-case events. We believe our study makes a valuable contribution to further this discussion.

**The reviewer is absolutely right to remind that ultimately this discussion on the inclusion of, and implications of worst-case scenarios comes down to risk tolerance levels of local communities and their decision-makers. With this in mind we have moved some text, and expanded to create a final prominent paragraph in the discussion that focusses on this issue and we hope addresses the reviewers concern.**

Note we have also included a reference to Veh et al. (2020), although of course, as we know from across the climate change risk management literature, we are starting to see and need to be preparing for events that may far exceed historical precedence.

**Tab. 3: please consider showing dam breach parameters and considered released volumes**

We have added dam breach width, which can be easily extracted from the model results. Release volumes of the initial avalanche are given in Table 1.

**Fig. 4: please elaborate on why modelled discharge from lowered Jialongco is higher than modelled discharge from not remediated Jialongco**

This is due to the fact that not only the lake level was lowered, but also the moraine dam was lowered and armoured. As a consequence, dam erosion was set to zero in the simulation, and the lowered freeboard results in a larger overtopping volume. We have elaborated the explanation of this in section 4.2.

**Discussion section – please consider splitting into sub-sections**

We have added two main sub-section titles.